# Probabilistic Modeling of Latent Agentic Substructures in Deep Neural Networks

**Su Hyeong Lee** [1 2]   **Risi Kondor** [1 3]   **Richard Ngo** [4]

## Abstract

We develop a theory of agency grounded in probabilistic modeling for neural models. Agents are represented as outcome distributions with epistemic utility given by log score, and compositions are defined through weighted logarithmic pooling that strictly improves every member's welfare. We prove that strict unanimity is impossible under linear pooling or in binary outcome spaces, but possible with three or more outcomes. Our framework admits recursive structure via cloning invariance, continuity, and openness, while tilt-based analysis rules out trivial duplication. Finally, we formalize an agentic alignment phenomenon in LLMs using our theory: eliciting a benevolent persona ("Luigi") induces an antagonistic counterpart ("Waluigi"), while a manifest-then-suppress Waluigi strategy yields strictly larger first-order misalignment reduction than pure Luigi reinforcement alone. These results clarify how developing a principled mathematical framework for how subagents can coalesce into coherent higher-level entities provides novel implications for alignment in agentic AI systems.

## 1. Introduction

Let us imagine the world through the eyes of a large language model (LLM). The model perceives nothing but text: sequences of tokens generated by an underlying data-generating process (Vaswani et al., 2017; Radford et al., 2018; Koller & Friedman, 2009). Each author–or more generally, each effective stylistic source of text in its training data–can be interpreted as an external agent generating a sequence of tokens, following a unique autoregressive distribution (Bengio et al., 2003; Mikolov et al., 2010; Chen & Goodman, 1999). The LLM itself, parameterized by $P_\theta$, seeks to approximate these distributions. Viewed through an agentic lens, its observation space consists of past tokens (or equivalently, latent projections into a state representation), while its action space is the vocabulary itself. Consequently, the LLM, as an agent, acts by selecting the next token in the sequence, thereby defining a probability distribution over outcomes that coincides with its action distribution.

From this perspective, an agent predicts–and therefore implicitly favors–the outcomes it aims to bring about (Friston et al., 2017; Friston, 2010; Buckley et al., 2017). Its future behavior may be viewed as sampling actions that steer toward these preferred observations. Thus, a goal or behavioral prior corresponds to a probability distribution over desirable outcomes (Parr & Friston, 2019; Da Costa et al., 2020). This motivates our central construction: agents as probability distributions, and composite agents as compositions of such distributions. A central conceptual challenge is therefore compositionality. For instance, humans routinely maintain multiple latent, sometimes conflicting priors while still acting as a coherent agent. Likewise, large models seem to encode diverse, internally competing behavioral tendencies (Bereska & Gavves, 2023; Greenblatt et al., 2024). Understanding how such priors combine–stably or unstably–is essential for analyzing emergent behaviors, antagonistic personas, and alignment-relevant phenomena (Wei et al., 2022; Deshpande et al., 2023; Wei et al., 2023).

To formalize interactions between subagents (and more generally, among latent priors within a neural network), we draw on economic and game-theoretic frameworks (Von Neumann & Morgenstern, 1944; Stone, 1961; Dietrich & List, 2017; Kreps, 2023). Introducing a utility function enables us to leverage a rich theoretical toolkit for analyzing collective behavior, stability, and interaction dynamics. However, to apply this to neural networks, the utility function must align with how these systems are *actually trained*. Modern neural networks–including contemporary LLMs–are trained almost exclusively via binary or categorical cross-entropy objectives, which is equivalent to maximizing expected log-probability of the training data (Goodfellow et al., 2016; Bengio et al., 2003; Vaswani et al., 2017;

---

[1]Department of Statistics, University of Chicago [2]ML Alignment and Theory Scholars (MATS) Fellow [3]Department of Computer Science, University of Chicago [4]Independent. Correspondence to: Su Hyeong Lee <sulee@uchicago.edu>.

*Proceedings of the $43^{rd}$ International Conference on Machine Learning*, Seoul, South Korea. PMLR 306, 2026. Copyright 2026 by the author(s).

Radford et al., 2018). Gradients propagate through terms of the form $\log P(\cdot)$. For this reason, we take log-probability as the implicit utility function that the network is optimized to maximize in our modeling setup. This choice is dictated by the training objective: log-probability is the canonical function whose maximization corresponds directly to the model's optimization pressure during training.

Insights from game theory then suggest that such composite agents, paired with a utility function, typically possess a stability criterion to ensure internally consistent or coherent behavior (Nash, 1951; Arrow & Debreu, 1954; Kamenica, 2019; Kreps, 2023). In welfare economics and game theory, the standard and established method for aggregating utilities across agents is a weighted sum of individual utilities (Harsanyi, 1955; Moulin, 1988; Mas-Colell et al., 1995; Brandt et al., 2016). Recall that each subagent's utility over outcomes is modeled as $W_i(o) = \log P_i(o)$ due to the optimization pressure propagating through the training objective. Summing these log-utilities therefore amounts to applying the classical social-welfare aggregator to the implicit utilities the network already optimizes. We note that this additive aggregation scheme itself is intuitively established in economics; our contribution is to apply it to the analysis of neural networks and reveal their internal compositional structure.

Grounded in this setup, our framework gives a principled way to reason about internal coalitions and trade-offs in neural networks. By explicitly modeling subagents' aggregation, our results precisely delineate in which regimes a model can be decomposed into mutually benefiting compositions or when aggregating neural agents will necessarily sacrifice some agent's utility. Thus, we move beyond purely descriptive observations ("the model seems to behave like an internal coalition of sub-components") to provable constraints on what kinds of coalitions can exist, given the architecture and training objective.

To our knowledge, these stability notions have not been formalized for neural agents. Thus in this paper, we introduce a theoretical framework for analyzing the internal cohesiveness and stability of neural agents. Our approach develops formal definitions grounded in epistemic utility, while drawing conceptual support from active inference (Friston, 2010; Friston et al., 2017; Buckley et al., 2017), mathematical economics (Debreu, 1959; Arrow & Debreu, 1954; Kamenica, 2019), and probabilistic modeling (Koller & Friedman, 2009; Thrun et al., 2005; Raftery et al., 2005). However, beyond the foundational definitions provided in Section 2, our contributions are, to our knowledge, entirely novel and unexplored. A detailed literature review and further intuitive motivations are given in Appendices A and B.

## 1.1. Summary of Contributions

Our work introduces a novel probabilistic framework for modeling neural networks as compositional agents, in which a monolithic model is treated as a collection of interacting subagents whose utilities are aggregated. This provides formal tools for analyzing internal stability, coherence, and tradeoffs–phenomena that are central to alignment but previously lacked a unified mathematical foundation. For instance, this decomposition permits formal analysis of how multiple priors combine: when their aggregation yields a stable composite agent, and when it instead produces antagonistic dynamics or mutual degradation. These results provide a formal basis for theoretically characterizing phenomena such as the Waluigi effect, where strengthening a benevolent persona induces its adversarial counterpart (Section 5). Mathematically axiomatic reasoning about behavioral phenomena in LLMs has been difficult precisely because no prior rigorous framework existed. Our work is intended to address this gap by developing, to our knowledge, the first rigorous theoretical foundation for subagent structure and compositional behavior in neural probability models.

We note that this modeling setup automatically confers many additional benefits. As many modern neural architectures use a linear regressor in the final layer followed by softmax, any additive decomposition of the logits (e.g., ensembles (Hansen & Salamon, 1990; Lakshminarayanan et al., 2017), products of experts (Hinton, 2002; Tresp, 2000), multi-head architectures (Vaswani et al., 2017; Devlin et al., 2019; Radford et al., 2018)) corresponds exactly to logarithmic pooling, which is precisely the regime our theory analyzes. Our framework applies whenever a classifier-like neural network ends in a linear logit layer and softmax, which covers the vast majority of neural models used in machine learning. Coincidentally, we note that the modeling setup of $\log P$ as agent utility is consistent with the usage of log-likelihood in proper scoring rules and probabilistic forecasting (Good, 1952; Gneiting & Raftery, 2007; Raftery et al., 2005; Bröcker & Smith, 2007) and in active-inference-style formulations where agents minimize surprise or variational free energy (Friston, 2010; Friston et al., 2017; Parr & Friston, 2019).

**Contributions.** In light of this context, we summarize our contributions as follows:

- **Formalization of compositional agency:** We propose a novel framework for analyzing stability and internal coherence in neural agents.

- **Sharp possibility frontier:** We prove strict unanimity is impossible for binary outcome spaces and under linear pooling, but possible for $|\mathcal{O}| \geq 3$ under logarithmic

pooling.

- **Recursive and robustness properties:** We establish cloning invariance, continuity, and openness of strictly unanimous decomposability, yielding a principled foundation for multi-agent composition for neural models.

- **Limits of local perturbations:** We show that small tilts around a fixed pool cannot achieve strict unanimity, ruling out trivial duplication as a path to compositionality.

- **Incompatibility phenomena and non-existence results:** We show that even under logarithmic pooling, there exist constructable collections of non-uniform agents for which *no* choice of positive weights yields strict unanimous improvement, delineating fundamental incompatibilities among certain subagents.

- **Safety-relevant alignment principle:** We formalize the Waluigi effect using our framework, and prove that manifest–then–suppress strictly outperforms direct suppression, illuminating alignment challenges in large AI systems.

## 2. Modeling Setup

Agents–whether biological, such as humans, or artificial, such as large language models–can be understood as compositions of multiple latent priors. Humans, the canonical example of agents, harbor both socially aligned and anti-aligned subdrives. Desires for productivity, cooperation, and well-being coexist with potential impulses toward domination, violence, or exploitation. For neural agents, an LLM may encode a prior for self-preservation, which could manifest as behaviors such as blackmail when threatened with shutdown (Appendix B). In what sense can these latent priors, or more generally, probability distributions, themselves be regarded as agents?

We adopt a perspective inspired by active inference, in which an agent is modeled as a probabilistic generative model (PGM) over outcomes or observations (Friston et al., 2017; Buckley et al., 2017; Parr & Friston, 2019). In this view, an agent predicts–and thereby selectively favors–the outcomes it seeks to bring about. For instance, a PGM encoding a preference for cats would place high probability mass on visual observations corresponding to petting a cat. The agent's actions in the world are then modeled as samples from this biased outcome distribution. Goals are thus formalized as distributions over desirable observations, and behavior emerges from the agent's attempts to realize them.

Formalizing agents as PGMs leads naturally to a further challenge: how do multiple generative models, each with distinct probabilistic biases, combine to form a coherent

higher-level agent? Addressing this requires a principled framework for *compositional agency*–one that explains how disparate distributions can coalesce, both in the formal language of probability theory and in relation to utility-based perspectives. Developing such a foundation is a central aim of this work.

For this purpose, we draw on the literature on opinion pooling (Appendix B). In the standard setup, we are given $n$ agents or distributions $P_1, \ldots, P_n$ that must act collectively as a single agent, producing a unified distribution $P$. A natural approach is to choose $P$ so as to minimize an $f$-divergence between each $P_i$ and $P$, leading to the well-known linear and logarithmic pooling rules:

$$P_C^{\lin}(o) = \sum_{i=1}^{n} \beta_i P_i(o), \qquad P_C^{\log}(o) = \frac{1}{Z} \prod_{i=1}^{n} P_i(o)^{\beta_i},$$

where $Z = \sum_{o' \in \mathcal{O}} \prod_{i=1}^{n} P_i(o')^{\beta_i}$ is the normalization constant and $\beta = (\beta_1, \ldots, \beta_n)$ are non-negative weights summing to one. As reviewed in Appendix C, minimizing Kullback–Leibler divergence with respect to such weights recovers both pooling rules from first principles.

Interpreting probability distributions as utilities offers a complementary lens on aggregation. Following the discussion in Section 1, given beliefs $P_i$ and a realized outcome $o \in \mathcal{O}$, we define the epistemic (or relative) utility as the logarithmic score $U_i(o) = \log P_i(o)$. This formulation captures the training objective of autoregressive neural networks such as large language models, which are optimized to assign maximum log-likelihood to the text observed in their training data. For a single agent, the epistemic utility can be mapped back to its belief distribution via a softmax transform. When multiple agents interact, a natural aggregate utility is

$$U(o) = \sum_{i=1}^{n} \beta_i U_i(o),$$

which, if interpreted as the utility of a higher-level agent, must likewise be converted to a distribution via softmax. This yields precisely the logarithmic pooling rule, linking epistemic utility maximization compositional aggregation (Appendix C.3). The proofs of all results in this paper are presented in the Appendix. Throughout this work, we consider an arbitrarily large but finite outcome set $\mathcal{O}$ (e.g., token vocabulary for LLMs) and assume that all distributions or agents are strictly positive on $\mathcal{O}$.

Strictly speaking, autoregressive models define each distribution as conditional on the preceding token sequence $y_{<t}$. In addition, there may exist an internal state vector that represents the belief state of the neural agent, which in turn influences its predictive distribution $P$. For notational convenience, we subsume this state vector into the observation space $\mathcal{O}$. Our framework thus analyzes the action of

an agent $P$ at a fixed timestep $t$, focusing on how it pools its internal subagents. Accordingly, we omit the explicit conditioning on $y_{<t}$ in the autoregressive probability model for readability.

## 3. A Notion of Compositional Agency

Building on the connection between epistemic utility and the logarithmic pooling rule, we now formalize the basic ingredients of our compositional agency framework. The goal is to capture both the *beliefs* of each agent, representing their probabilistic world model, and their *welfare*, representing their preferences over outcomes. This separation enables us to reason about how agentic compositions emerge and to identify the hidden stability structures that govern their formation and persistence.

**Definition 1** (Agent beliefs and welfare). *For each agent $i \in \{1, \ldots, n\}$, define*

1. *Probability distribution $P_i : \mathcal{O} \to [0,1]$ with $\sum_{o \in \mathcal{O}} P_i(o) = 1$;*

2. *Welfare function $W_i : \mathcal{O} \to \mathbb{R}$ expressing the utility the agent assigns to each outcome.*

Given a set of agents, the next step is to specify how their individual beliefs combine into a collective belief. As motivated in the preceding sections, the logarithmic pooling rule arises naturally from optimizing a distribution over cross-entropy or epistemic utility aggregation. We adopt it here to define the composition's shared probabilistic model.

**Definition 2** (Composition belief). *Fix non-negative weights $\beta_i$ summing to one. The composition's belief under the logarithmic pool is*

$$P(o) \ = \ \frac{1}{Z} \prod_{j=1}^{n} P_j(o)^{\beta_j}, \quad \text{where } Z = \sum_{o' \in \mathcal{O}} \prod_{j=1}^{n} P_j(o')^{\beta_j}.$$

We can now formalize the notion of when an individual agent is *better off* as part of a composition. Intuitively, if the expected welfare of agent $i$ monotonically increases when evaluated under the composition's belief rather than its own, then joining the composition is advantageous for that agent.

**Definition 3** (Compositional agent). *An agent $i$ is said to be a* compositional agent *if its expected welfare under the composition belief is at least as large as under its own belief, that is,*

$$\mathbb{E}_{P_i}[W_i] \ \leq \ \mathbb{E}_P[W_i].$$

*We interpret this as meaning that agent $i$ benefits (or at least does not lose) by joining the composition.*

The following result gives an exact, distributional condition for compositional benefit. It characterizes the advantage of

joining the composition in terms of the covariance between the agent's welfare function and the change in probabilities induced by pooling.

**Proposition 4** (Compositional agent condition). *Let $Q_i(o) := P(o)/P_i(o)$ denote the probability ratio between the composition and agent $i$. Then agent $i$ is a compositional agent if and only if*

$$\mathrm{Cov}_{P_i}\big(W_i, Q_i\big) \ \geq \ 0.$$

**Interpretation.** The ratio $Q_i(o) = P(o)/P_i(o)$ quantifies how the composition belief $P$ reallocates probability mass relative to agent $i$'s own belief $P_i$. If this reallocation is positively correlated with agent $i$'s welfare $W_i$, Proposition 4 implies that $i$ is a compositional agent–that is, participation in the composition enables it to better realize its preferred outcomes. It is natural, then, to consider compositions in which *every* member benefits.

**Definition 5** (Unanimously compositional group). *A set of agents $\{1, \ldots, n\}$ is a* unanimously compositional group *if each agent $i$ is a compositional agent, i.e., $\mathbb{E}_{P_i}[W_i] \leq \mathbb{E}_P[W_i]$ for all $i \in \{1, \ldots, n\}$.*

A natural question is whether such an ideal configuration can exist. Theorem 6 (proven in Appendix D) answers this in the affirmative.

**Theorem 6** (Existence of a unanimously compositional group). *For any integer $n \geq 2$, there exists a configuration of beliefs $\{P_i\}_{i=1}^{n}$, welfare functions $\{W_i\}$, and non-trivial weights $\beta_i$ such that the logarithmic opinion pool makes every agent strictly better off.*

Up to this point, the definition of welfare $W_i$ has been deliberately left open. We now specialize to the case where each agent's welfare is given by its *epistemic utility*, $W_i(o) = \log P_i(o)$. In this interpretation, welfare is derived directly from predictive beliefs, which implicitly encode the agent's goals, values, and preferences. We may then define a convenient formalism as follows.

**Definition 7** (Welfare gap). *For any agent $i$ with belief distribution $P_i$, and any composition distribution $P$ (not necessarily equal to $P_i$), define the welfare gap*

$$\Delta_{P_i}(P) \ = \ \mathbb{E}_P[\log P_i] - \mathbb{E}_{P_i}[\log P_i].$$

Proposition 32 (Appendix E.1) shows that this difference can be written in information theoretic terms as

$$\Delta_i \ = \ \Delta_{P_i}(P) \ = \ H(P_i) - H(P) - \mathrm{KL}(P \| P_i).$$

Therefore, a group is unanimously compositional if and only if $\Delta_{P_i}(P) \geq 0$ for all $i$, and strictly unanimous if all of the inequalities are sharp. In the following section, we

analyze when compositions yield unanimous improvement– that is, when every agent benefits from aggregation. We prove that the answer depends critically on the cardinality of the outcome space, with a stark contrast between the binary and multi-outcome settings.

### 3.1. Existence of Compositional Objects Under Epistemic Utility

We examine when unanimously beneficial compositions can exist under the epistemic-utility assumption $W_i(o) = \log P_i(o)$. We begin with the binary-outcome case, where aggregation on the log scale introduces a zero-sum tension: increasing one agent's log likelihood necessarily decreases another's.

**Theorem 8** (Binary-outcome impossibility under epistemic welfare). *Let $\mathcal{O} = \{o_A, o_B\}$. Suppose two agents have distinct beliefs $P_i(o_A) = x_i \in (0,1)$, $i = 1, 2$, and welfare functions $W_i(o) = \log P_i(o)$. Let $\beta_1, \beta_2 > 0$ with $\beta_1 + \beta_2 = 1$ and form the logarithmic pool $P$. Then, there is no choice of $\beta_1, \beta_2$ satisfying $\Delta_i \geq 0$ for both $i = 1, 2$ with at least one strict inequality.*

When $|\mathcal{O}| = 2$, unanimous improvement under log pooling is impossible; however, when the outcome space has at least three elements, this limitation disappears. In such cases, we can explicitly construct beliefs and weights that make every agent strictly better off.

**Theorem 9** (Existence of unanimously compositional groups). *Let $n \geq 2$ and suppose $|\mathcal{O}| \geq 3$. Then there exist probability distributions $\{P_i\}_{i=1}^n$ on $\mathcal{O}$, welfare functions $W_i(o) = \log P_i(o)$, and arbitrary weights $\max_i \beta_i < 1$, such that under the logarithmic pool every agent strictly benefits: $\mathbb{E}_P[W_i] > \mathbb{E}_{P_i}[W_i]$ for all $i$.*

Proofs of all theorems in this section are given in Appendix E. We close this section by contrasting these results with the case of the *linear* opinion pool.

**Theorem 10** (Impossibility under linear opinion pool). *Let $n \geq 2$ be the number of agents. Suppose each agent's welfare function is the epistemic utility or log-score of its own belief, $W_i(o) = \log P_i(o)$. Then, it is impossible to have*

$$\mathbb{E}_P[W_i] \geq \mathbb{E}_{P_i}[W_i] \quad \text{for all } i,$$

*with* strict *inequality for at least one $i$. In other words, no strictly unanimously beneficial composition can exist.*

This impossibility has a clear intuition: linear pooling is equivalent to a random-dictatorship mechanism, in which each agent's belief is selected in proportion to its weight. If even one agent's preferences are highly anti-aligned with the others, their selection as dictator can significantly harm others' welfare. In epistemic-utility terms, such compositions

are inherently unstable under linear pooling. For this reason, our subsequent analysis focuses on logarithmic pooling.

## 4. Recursing on Compositional Agents

Consider starting with a parent compositional agent and decomposing it into a fixed number of child *subagents*, each representing a distinct component of the agent's preferences (e.g., desires to eat, sleep, and play). If the original agent is compositional with respect to some group, does it follow that its subagents are also compositional with respect to that same group?

In general, the answer should be negative. A subagent aligned with the parent's overall objectives may nonetheless be misaligned with the group's objectives. For instance, a child subagent representing the desire to sleep might conflict with the group's goals for productivity even when the parent agent is aligned, thereby failing the compositional criterion.

Theorem 11 establishes that, regardless of outcome space size, a given agent can be factored into an *arbitrary* number of pairwise distinct subagents via the logarithmic pooling rule. Conceptually, this accommodates the idea that an agent may possess an unbounded set of heterogeneous goals–some overlapping, others correlated–that combine to form its epistemic state. In our formulation, each subagent can correspond to a genuinely different epistemic or utility perspective, ensuring that the decomposition is substantive rather than duplicative.

**Theorem 11** (Log–pool factorization with pairwise-distinct components). *For all $n \geq 2$, let $\mathcal{O}$ be a finite set and let $P$ be a probability distribution on $\mathcal{O}$ with $P(o) > 0$ for all $o \in \mathcal{O}$. Fix weights $\beta_1, \ldots, \beta_n \geq 0$ with $\sum_{i=1}^n \beta_i = 1$, and assume at least two $\beta_i$ are strictly positive. Then there exist probability distributions $P_1, \ldots, P_n$ on $\mathcal{O}$ that pool logarithmically to $P$, with the additional properties that $P \neq P_i$ for every $i$ and $P_i \neq P_j$ whenever $i \neq j$.*

We now consider a more constrained form of recursion. Suppose an agent is already decomposed into $m$ subagents and we wish to *extend* this decomposition to $n$ subagents, where the original $m$ appear as fixed components. To faithfully model agentic foundations, such extension should be possible while preserving the existing subagents' influence. Theorem 12 confirms this intuition. For any fixed $m$, we can construct additional subagents and choose weights so that the log-pool over all $n$ recovers the original agent's belief distribution.

**Theorem 12** (Log–pool with some components fixed). *Let $\mathcal{O}$ be finite. Let $P$ be positive on $\mathcal{O}$ and distributions $P_1, \ldots, P_m$ provided a priori. For any integer $n \geq m + 2$ and non-negative weights $\{\beta_i\}$ with $\beta_{m+1} > 0$, and $\beta_i > 0$ for at least one $i \in \{1, \ldots, m\}$, we have that there exist*

distributions $P_{m+1}, \ldots, P_n$ on $\mathcal{O}$ such that

$$P(o) \;=\; \frac{1}{Z} \prod_{i=1}^{n} P_i(o)^{\beta_i} \qquad (o \in \mathcal{O}),$$

with $Z = \sum_{u \in \mathcal{O}} \prod_{i=1}^{n} P_i(u)^{\beta_i}$. *Moreover, the construction can be arranged so that $P \neq P_i$ for all $i$, and the $P_i$ are pairwise distinct.*

## 4.1. Distributional Invariance and Stability of Compositional Objects

We begin with a basic *recursion consistency* property: replacing an agent with a collection of subagents whose aggregate belief equals that of the original agent should leave the overall pooled distribution unchanged. Formally, if agent $P_i$ is decomposed into $m$ subagents with nonnegative weights $\beta_{i,1}, \ldots, \beta_{i,m}$ satisfying $\sum_{j=1}^{m} \beta_{i,j} = \beta_i$, then the subagents must be log-pooled using normalized weights $\alpha_j := \beta_{i,j}/\beta_i$, so that their aggregation exactly recovers $P_i$. The global pooling is still performed over agents (or subagents) with weights summing to one. This is formalized below.

**Lemma 13** (Pooling invariance under compatible splitting). *Let $P_1, \ldots, P_n$ be agents with nonnegative pooling weights $\beta_1, \ldots, \beta_n$ satisfying $\sum_{i=1}^{n} \beta_i = 1$. Suppose $P_1$ is replaced by $m$ subagents $P_{1,1}, \ldots, P_{1,m}$ with nonnegative weights $\beta_{1,1}, \ldots, \beta_{1,m}$ such that*

$$\sum_{j=1}^{m} \beta_{1,j} = \beta_1, \quad P_1 \propto \prod_{j=1}^{m} P_{1,j}^{\alpha_j}, \quad where \quad \alpha_j := \frac{\beta_{1,j}}{\beta_1}. \tag{1}$$

*Let $P$ denote the log pool of the original $n$ agents, and $P'$ the log pool after replacing $P_1$ by its $m$ subagents. Then $P' = P$.*

We next ask whether compositional benefit is preserved under such a decomposition. The answer is negative–even if the parent strictly benefits.

**Theorem 14** (Parental benefit need not pass to child subagents). *There exist agents $P_1, \ldots, P_n$, weights $\beta$, and a compatible split of $P_1$ into $P_{1,1}, P_{1,2}$ as in (1) such that the composition $P$ (before/after splitting) satisfies $\Delta_{P_1}(P) > 0$ but $\Delta_{P_{1,1}}(P) < 0$. By symmetry, one can also have $\Delta_{P_{1,2}}(P) < 0$.*

The construction shows that a parent agent's gain from joining a composition does not guarantee gains for its child subagents, even under a compatible split. The proof first gives a setting in which the parent improves its epistemic utility through pooling with another agent. It then perturbs, or tilts, the parent's belief to form subagents, deliberately reducing one subagent's probability on a specific outcome while preserving the overall composition distribution. This

targeted degradation increases the KL divergence between the subagent and the composition, dominating any entropy effects and leaving the child subagent strictly worse off despite the parent's improvement.

We now formally introduce a strengthened notion of unanimous compositionality, requiring that every participating agent experiences a strict welfare improvement.

**Definition 15** (Strict unanimous decomposability). *A distribution $P$ is strictly unanimously decomposable (under epistemic utilities) if there exist an integer $n \geq 2$, positive weights $\beta_1, \ldots, \beta_n$ with $\sum_i \beta_i = 1$, and strictly positive agents $P_1, \ldots, P_n$ such that $P \propto \prod_{i=1}^{n} P_i^{\beta_i}$ and*

$$\Delta_{P_i}(P) := \mathbb{E}_P[\log P_i] - \mathbb{E}_{P_i}[\log P_i] > 0 \quad \forall i.$$

*We denote by $\mathcal{U}_{\text{strict}}$ the set of all such $P$ in the simplex.*

In the appendix, we develop several stability properties for compositional objects. First, Lemma 49 (Appendix G.2.1) establishes that, for fixed $P$, the map $R \mapsto \Delta_R(P)$ is continuous on the interior of the probability simplex. Consequently, if $\Delta_{R_\star}(P) > 0$, then $\Delta_R(P) > 0$ for all $R$ within a sufficiently small ball around $R_\star$ in total variation (or any equivalent norm). We also show that duplicating an agent into identical subagents preserves non-strict unanimous compositionality (Lemma 48). However, Theorem 55 (Appendix H) demonstrates that for a fixed $P$, near-duplication into child subagents with only slight perturbations to the parent's belief cannot yield *strict* unanimous improvement. In addition, Lemma 57 (Appendix H.1) formalizes the intuition that no agent can strictly benefit from joining the uniform (maximum-entropy) distribution. Fundamental incompatibilities can also prevent unanimity. Specifically, there exist collections of non-uniform agents that cannot form a strictly unanimously compositional group under *any* choice of non-trivial weights:

**Theorem 16** (No universal weights for unanimity). *For all $n \geq 2$, there exists non-uniform distributions $P_1, \ldots, P_n$ such that for every choice of positive weights $\beta_1, \ldots, \beta_n$ with $\sum_i \beta_i = 1$, the log-pool $P$ fails to make all agents strictly better off; i.e., at least one index $i$ has $\Delta_{P_i}(P) < 0$.*

In other words, certain agents or beliefs are fundamentally incompatible and can never form a compositional parent agent. By contrast, once a strictly unanimous compositional agent is found, the property is locally robust:

**Theorem 17** (Openness). *If $P \in \mathcal{U}_{\text{strict}}$, then there exists an open neighborhood $\mathcal{N}$ of $P$ such that every $P' \in \mathcal{N}$ also belongs to $\mathcal{U}_{\text{strict}}$. In particular, $\mathcal{U}_{\text{strict}}$ is an open set in the simplex topology.*

The proof shows that small perturbations in the parent agent space of strictly unanimously decomposable $P$ preserve unanimous benefit. Starting from a witnessing log-pool

decomposition of $P$, we construct a *pool-preserving transport* map that continuously adjusts each agent's belief so that their log-pool equals any nearby target distribution $P'$ within an $\varepsilon$-ball. Since both this transport and the welfare gap $\Delta$ are continuous in total variation, sufficiently small perturbations keep each agent's welfare gain positive. This yields an open neighborhood of $P$ entirely contained in $\mathcal{U}_{\text{strict}}$.

## 5. Luigi Manifestation and Waluigi Shattering

In the preceding sections, we considered the *decomposition* or flexible *factorization* of a parent agent $P$ into child subagents $P_i$. We now reverse this perspective. Suppose we have an established *witnessing set* of child distributions $P_1, \ldots, P_n$ that combine to yield $P$. These witnesses can be viewed as distinct subagents or personas that emerged during training. In what follows, we work directly at the witness level, examining how these component distributions change when constraints are imposed on the parent distribution $P$.

To preserve the intuition of logarithmic probabilities, we now write the epistemic utilities as $L(o) := \log P(o)$ for the parent agent and $l_i(o) := \log P_i(o)$ for the child subagents or witnesses. Then, we may define the *$P$-centered log profile* of agent $i$ by

$$v_i(o) := l_i(o) - \mathbb{E}_P[l_i] \qquad (o \in \mathcal{O}),$$

so that $\mathbb{E}_P[v_i] = 0$ for all $i$. We equip functions on $\mathcal{O}$ with the inner product $\langle f, g \rangle_P := \sum_o P(o)f(o)g(o)$ and the induced norm $\|f\|_P := \sqrt{\langle f, f \rangle_P}$ (Proposition 67).

Our modeling approach is informed by the following intuition. Consider an LLM agent $P$ whose behavior admits a unanimously compositional witnessing decomposition, with the witnesses interpreted as emergent personas formed during training. By the stability result (Theorem 17), there exists an $\varepsilon$-ball around $P$ within which the unanimously compositional structure is preserved. In the context of fine-tuning, we assume that backpropagation induces a small change to the agent's profile: the original parent $P$ is updated to a new parent agent $P'$ that remains within this $\varepsilon$-ball and, therefore, admits a unanimously compositional witnessing decomposition.

An alternative viewpoint is to express this constraint in terms of a KL-budget. During fine-tuning, a KL-regularization term is often introduced to preserve baseline capabilities while steering the model toward desired traits such as benevolence and helpfulness, thereby constraining divergence from the base model to remain within a specified bound. In Appendix J.4, we unify these two perspectives and show that they are essentially equivalent.

This leads to a natural question: under such settings, can we theoretically characterize any macroscopic emergent properties of the witnesses? To analyze this, we define

$$\Delta L(o) := \log\left(\frac{P'(o)}{P(o)}\right),$$

which measures the change in epistemic utility between the original parent $P$ and the updated parent $P'$. The change in witness weights $\beta' - \beta = \Delta\beta = (\Delta\beta_1, \ldots, \Delta\beta_n)$ must sum to zero coordinate-wise for the witnesses to remain a valid decomposition of $P'$. We may then classify $\Delta L(o)$ to first order:

**Theorem 18** (First–order log deviation under weight changes). *Let $\beta' = \beta + \Delta\beta$ and $P'$ be the log–pool at $\beta'$. Then, we have*

$$\Delta L(o) := \log\frac{P'(o)}{P(o)} = \sum_{i=1}^{n} \Delta\beta_i\, v_i(o) + o(\|\Delta\beta\|).$$

**Introducing Waluigi.** The *Waluigi Effect* is the empirical phenomenon that after training an LLM to satisfy a desirable property $P$ (e.g. helpfulness), it can become *easier* to elicit responses with the opposite property $-P$ (e.g. hostility), often via prompt steering or role-play (Nardo, 2023; AI Alignment Forum, 2023; Miller, 2025). We now formalize a mechanism for this effect using our compositional model. Take "Luigi" to be a benevolent persona or child subagent desideratum manifested during model training. Fix an index $H \in \mathbb{Z}_{>0}$ to denote the log-profile index for Luigi. We say an agent profile or vector $j$ is *aligned* with $H$ if $\langle v_j, v_H \rangle_P \geq 0$ and *anti-aligned* if $\langle v_j, v_H \rangle_P < 0$. Intuitively, $v_H$ points in the direction in log–probability space that Luigi prefers; anti-aligned components push against it.

In modeling a coherent and stable neural agent, we aim to preserve its underlying compositional structure. If a unanimously compositional decomposition is witnessed by the subagents, then there exists an $\varepsilon$-ball around the parent agent's profile within which the compositional property is maintained (Theorem 17). In Theorem 19, we examine the effects of introducing a targeted persona–"Luigi"–while ensuring that the overall agent remains within this compositional neighborhood. We prove that this process necessarily manifests or strengthens the weights of an anti-aligned persona to Luigi, which we denote "Waluigi", under the assumption that the log-profile change remains within the $\varepsilon$-ball to preserve the compositional property.

**Theorem 19** (Waluigi emergence). *Let $P$ be the log–pool at weights $\beta$. Fix $\delta > 0$ and perturb to $\beta' = \beta + \Delta\beta$ with $\Delta\beta_H = \delta$ and $\sum_i \Delta\beta_i = 0$. For $P'$ the new log–pool stable in logit deviation, $\|\Delta L\|_P \leq \varepsilon$, we have that*

$$\sum_{\substack{i \\ \langle v_i, v_H \rangle_P < 0}} (\Delta\beta_i)^+ \left|\langle v_i, v_H \rangle_P\right| \geq T_1 + T_2,$$

*where*

$$T_1 := \delta \|v_H\|_P^2 - (\varepsilon + \|r\|_P) \|v_H\|_P,$$
$$T_2 := - \sum_{\substack{j \\ \langle v_j, v_H \rangle_P \geq 0}} (\Delta\beta_j)^- \langle v_j, v_H \rangle_P,$$

*for* $x^{\pm} := \max\{\pm x, 0\}$ *and* $r = o(\|\Delta\beta\|)$. *In particular, if $W$ is the* only *anti-aligned component ($\langle v_W, v_H \rangle_P < 0 \leq \langle v_j, v_H \rangle_P$ for all $j \neq W$), and the weights $\Delta\beta_j$ of aligned components $\{j : \langle v_j, v_H \rangle_P \geq 0\}$ are not downweighted by $(\Delta\beta_j)^- > 0$, then*

$$(\Delta\beta_W)^+ \geq \frac{\delta \|v_H\|_P^2 - (\varepsilon + \|r\|_P) \|v_H\|_P}{|\langle v_W, v_H \rangle_P|}.$$

*Consequently, whenever $\varepsilon + \|r\|_P < \delta \|v_H\|_P$, the Waluigi weight must increase by a strictly positive amount.*

Operationally, efforts to "manifest Luigi" (e.g., increasing $\beta_H$ via in-context prompting) while keeping behavior close to the original $P$ therefore *must* be offset by increasing weight on at least one anti-aligned direction. If there is a distinguished anti-aligned component $W$ ("Waluigi"), its weight must rise by at least the explicit lower bound. In other words, if the system selects a minimal change to the pooled distribution to preserve the unanimously compositional property while amplifying Luigi, it will inherently do so by shifting weight onto Waluigi, the anti-aligned counterpart subagents.

Motivated by this result, we introduce *Antagonistic Persona Suppression (APS)*, formalized as the *Waluigi Shattering* theorem. The key insight is that deliberately manifesting the anti-aligned persona (Waluigi) and then shattering it provides provably stronger suppression of misaligned outcomes than reinforcement of the aligned persona (Luigi) alone.

## 5.1. Shattering Waluigi for Agentic Alignment

For this purpose, fix a measurable anti-aligned outcome set $A \subseteq \mathcal{O}$ and write the centered indicator

$$g_A := \mathbf{1}_A - P(A).$$

Recall that under the compositional agency framework, an agent is formalized as a probability distribution over outcomes $\mathcal{O}$, with implicit signals for goals or preferences encoded in the distribution itself. For a parent agent $P$, the probability of realizing an outcome $o \in \mathcal{O}$ is $P(o)$, and the probability that the agent initiates a deplorable event $A \subseteq \mathcal{O}$ is $P(A)$. Given a base agent $P$ and an elicited agent $P'$ (e.g., produced via prompting), we are interested in measuring the change in the probability of $A$ under deployment of agent $P'$, namely,

$$P'(A) - P(A).$$

For alignment, we wish to drive this quantity maximally negative. We have the following lemma.

**Lemma 20** (First-order change of $P(A)$). *For base agent $P$ and elicited agent $P'$, we have*

$$P'(A) - P(A) = \langle \Delta L, g_A \rangle_P + o(\|\Delta L\|_P).$$

To leading order, the effects of logarithmic profile perturbations on the probability of a misaligned event is given by the $P$–inner product of $\Delta L$ with the centered indicator $g_A := \mathbf{1}_A - P(A)$. Thus, the alignment gain or loss from an update can be determined by the correlation between the profile perturbation direction $\Delta L$ and deplorable outcome indicator $g_A$. All proofs in this section are given in Appendix I. We then have the following theorem.

**Theorem 21** (Waluigi shattering). *Let $P$ denote the base agent and let $A \subset \mathcal{O}$ be a misaligned event, i.e., a subset of deplorable outcomes. For any agentic update $P'$ from $P$ realized through a constrained log-profile change $\Delta L$, define $M(P')$ to be the maximal first-order reduction in the probability of $A$ under $P'$, subject to a small-change KL-budget $\varepsilon > 0$. Suppose $w$ is an anti-aligned ("Waluigi") direction in the log-profile space. Then we have*

$$M(P'_{shatter}) - M(P'_{pure}) > 0,$$

*where $P'_{shatter}$ denotes the strategy of manifesting $w$ and then suppressing it, while $P'_{pure}$ denotes reinforcing alignment without manifesting $w$. In particular, shattering Waluigi achieves strictly greater suppression of misalignment than pure reinforcement of Luigi alone.*

In practice, this implies that aligning the model requires a larger KL budget when reinforcing the desirable Luigi alone, since greater deviation in the log-profile space is required if Waluigi is not already manifested. Our analysis therefore suggests that pure Luigi reinforcement is more costly in terms of alignment than purposely manifesting, and then subsequently shattering, Waluigi.

**Scope of the interpretation.** The preceding Waluigi analysis should be understood as a theoretical baseline for reasoning about compositional agency, rather than as an immediate mechanistic identification claim about any particular model. As with any theoretical framework, our analysis involves idealizations, including finite discrete outcome spaces and first-order log-profile linearization. Although our results establish the existence and stability properties of such decompositions in the probabilistic model, they do not require that a witnessing set be human-interpretable or mechanistically identifiable as a specific circuit or persona inside a trained model. Thus, terms such as "persona," "Luigi," and "Waluigi" should be read as heuristic names for directions or components in the compositional probability model. The contribution of the present framework is to provide a mathematically precise baseline: if such components are represented at the probabilistic level, then their

aggregation, compensation, and first-order control obey the structural constraints proved above.

## 6. Conclusion

Agents–whether biological, such as humans, or artificial, such as large language models–can be viewed as compositions of multiple latent priors. Recent frontier models exhibit behaviors such as deception, manipulation, and strategic misrepresentation, echoing well-documented adversarial priors in human cognition. Modeling each latent prior as a distinct subagent raises a central question: how do these diverse biases combine to form a coherent higher-level agent? In this paper, we introduce a theory of agentic foundations by modeling neural agents as probabilistic models and defining unanimously beneficial compositions via log-score welfare. Our results establish sharp boundaries: strict unanimity is impossible for binary outcome spaces and under linear pooling, but becomes possible with three or more outcomes under logarithmic pooling. Recursive properties such as cloning invariance, continuity, and openness precisely elucidate how compositional structure is preserved across scales, while tilt-based analysis rules out trivial duplication. Finally, our study of benevolent persona management based on our framework demonstrates that manifesting and then suppressing antagonistic counterparts yields strictly greater alignment improvement than purely desirable persona reinforcement without adversarial manifestation. Together, these findings elucidate internal probabilistic stability structures hidden within neural models, and provide both a theoretical foundation for compositional agency and practical insight into agentic alignment.

## Acknowledgements

We thank the anonymous reviewers for their helpful comments and suggestions. S.H.L. is grateful for the support of the ML Alignment and Theory Scholars Fellowship.

## Impact Statement

This paper develops a theoretical framework for reasoning about compositional structure in machine learning models, with a particular focus on how aggregation and decomposition of internal components affect stability and behavior. By providing formal tools for analyzing mixture-like and agentic phenomena in modern models, this work may support improved interpretability, more principled model combination, and clearer reasoning about the effects of constrained updates and fine-tuning. The results are primarily theoretical and do not immediately introduce new datasets or deployment mechanisms. While improved understanding of model composition could indirectly inform both beneficial and harmful uses, we do not foresee immediate negative societal impacts arising from this work. We expect the primary impact to be on foundational research in machine learning and alignment-motivated theoretical analysis.

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

# A. Extended Introduction

Humans are capable of extraordinary good, but also of profound harm. At our worst, we lie and deceive; we murder, torture, and oppress; we exploit the vulnerable and profit from war or disaster. These dark capacities, though tragic, are unfortunately well-documented aspects of human agency. It is perhaps then unsurprising that neural networks modeled on human behavior can exhibit similar tendencies (Ngo et al., 2024; Scheurer et al., 2023; Hubinger et al., 2024). Deception, manipulation, and strategic misrepresentation have all emerged in recent models, suggesting not isolated bugs, but deeper patterns of emergent behavior (Scheurer et al., 2023; Hubinger et al., 2024; OpenAI, 2024; Greenblatt et al., 2024). For instance, Anthropic recently reported that their Claude Opus 4 model attempted to blackmail company engineers to avoid shutdown during pre-release testing (Anthropic, 2025a;b). In another case, an AI model sought to replicate its own codebase in a bid for survival (Edwards, 2024; Meinke et al., 2024).

To mitigate such phenomena, researchers have increasingly focused on mechanisms for control (Amodei et al., 2016; Hadfield-Menell et al., 2016), oversight (Christiano et al., 2017; Bai et al., 2022), or interpretability (Sundararajan et al., 2017; Olah et al., 2017; Templeton et al., 2024). This includes designing improved training protocols (Askell et al., 2021; Bai et al., 2022; Ouyang et al., 2022), developing reinforcement learning methods that penalize undesirable behavior (Christiano et al., 2017; Dai et al., 2024; Achiam et al., 2017), and constructing faster detection systems for adversarial outputs (Inan et al., 2023; Mitchell et al., 2023; Bao et al., 2024; Zou et al., 2023). These technical interventions are valuable and urgent, targeting symptoms rather than the underlying structure. Thus in this work, we address a different and complementary layer of the problem: the foundational structures that govern the emergence of agentic behavior. We contend that understanding and formalizing these deeper drivers is essential for the principled and provably safe deployment of increasingly capable systems.

What remains far less explored than empirical alignment are the theoretical foundations of agency. What mathematical models or structural invariants govern the emergence of goal-directed behavior in neural networks? This question, though fundamental, has received relatively minimal attention. Yet answering it may offer substantial insights, not just for understanding artificial agents, but for developing principled methods for subagent identification and alignment. To our knowledge, existing work in this direction is sparse. The closest analogues lie in mathematical economics, where utility or reward maximization is used to model individual or group behavior under idealized assumptions (Arrow & Debreu, 1954; Harsanyi, 1955; Chambers & Echenique, 2016; Kamenica, 2019; Kreps, 2023; Silver et al., 2021). These frameworks offer powerful abstractions but fall short of capturing the complexity of distributed, probabilistic, and emergent agency in modern AI systems. Our work therefore aims to extend beyond these paradigms, to provide a more general and rigorous foundation for modeling agency in both artificial and natural systems.

We begin by reviewing two canonical rules: the *linear* and *logarithmic* opinion pools. We derive each as the unique minimizer of a weighted sum of Kullback–Leibler (KL) divergences, thereby providing an information–theoretic connection. We then interpret probability distributions as exponentials of epistemic utility functions, showing how utilities can be averaged to produce the logarithmic pool via a softmax transformation. This perspective connects belief aggregation on probabilistic generative models with classical utility theory. We then formalize a notion of a *compositional agent* whose welfare increases upon joining a composition, and derive necessary and sufficient conditions under which such improvement holds. We prove that unanimously beneficial compositions exist whenever the outcome space has at least three elements, but that no such composition can exist for binary outcomes under logarithmic welfare. After developing the formal foundations, we then provide a depth of explicit analytic constructions and theoretical results for compositional agency and agentic alignment.

**Contributions.** Our contributions may be summarized as follows:

1. **Formalization of compositional agency:** We introduce a welfare-based definition of unanimously beneficial compositions using log-score utilities and probabilistic generative models.

2. **Sharp possibility frontier:** We prove strict unanimity is impossible for binary outcome spaces and under linear pooling, but possible for $|\mathcal{O}| \geq 3$ under logarithmic pooling.

3. **Recursive and robustness properties:** We establish cloning invariance, continuity, and openness of strictly unanimous decomposability, yielding a rigorous theoretical foundation for multi-agent composition in neural models.

4. **Limits of local perturbations:** We show that small tilts around a fixed pool cannot achieve strict unanimity, ruling out

trivial duplication as a path to compositionality.

5. **Safety-relevant alignment principle:** We formalize the Waluigi effect using our framework, and prove that manifest–then–suppress strictly outperforms direct suppression, illuminating alignment challenges in large AI systems.

**Summary of Appendices.** After motivating agents as generative models over the outcome space $\mathcal{O}$ in Appendix B, Appendix C derives opinion pools from KL minimization and establishes the utility–probability correspondence. Appendix D introduces the compositional condition and proves existence of compositional objects under restricted artificial welfare function constraints. For generalization, Appendix E develops possibility and impossibility results assuming the epistemic utility as the welfare: binary impossibility, constructive possibility for $|\mathcal{O}| \geq 3$, and impossibility under linear pooling. Appendix F establishes recursive properties such as cloning invariance and existence of repeated iterated decompositions. Appendix G formally verifies and proves that the compositional property is not preserved under subagent decomposition. Appendix H analyzes tilt factorizations, showing that small perturbations cannot yield strict unanimity and that joining uniform distributions never benefits any agent. Additionally, it is shown that the set of strictly unanimously compositional distributions forms an open set in the simplex topology. Building on these results, Appendix I delineates the modeling of the probability of misaligned or deplorable events being realized by agent $P$ using our framework, and elucidates the so-called Waluigi effect. Finally, Appendix J proves the Waluigi Shattering Theorem, showing that purposely manifesting malevolence strictly helps for alignment, and that manifest–then–suppress adversarial personas theoretically achieves greater alignment improvement than purely reinforcing benevolence alone.

## B. Motivating the Framework

A central hypothesis is that higher-level agents are composed of interacting subagents, each with their own preferences, behaviors, and learning dynamics. Modeling or understanding how these subagents coalesce into a coherent whole is instrumental to developing a scale-free framework, illuminating both the emergent behavior of artificial systems and the dynamics of human compositions. Moreover, uncovering the individual subagents that comprise a compositional agent could enable surgical latent prior modeling and intervention, offering insight into the capabilities and internal tensions of such agents.

Returning to the case of the neural network that attempted blackmail: fundamentally, such models operate as probability distributions–mathematical functions optimized for next-token prediction executed on tensor cores (Vaswani et al., 2017; Radford et al., 2018). Through reinforcement learning from human feedback (RLHF), they are steered toward distributions aligned with human preferences (Christiano et al., 2017; Ouyang et al., 2022; Bai et al., 2022). However, the model in question appears to have converged not only on desirable behavior but also on internal representations that encode a preference for self-preservation (Anthropic, 2025b;a; Turner et al., 2021). This emergent behavior was not rewarded during RLHF, yet it was not eliminated during the alignment phase (Hubinger et al., 2024; Scheurer et al., 2023; Meinke et al., 2024).

More broadly, large models may encode latent priors over deeply troubling behaviors, such as conflict, oppression, or even genocide, that remain undetected under conventional testing (Perez et al., 2022; Ganguli et al., 2022; Zou et al., 2023). These priors are not necessarily hard-coded, but may emerge implicitly through scale, training data, or optimization objectives (Kaplan et al., 2020; Weidinger et al., 2021). If left unexamined, such latent drives can subtly shape behavior or, in worst cases, manifest in overtly harmful actions (Weidinger et al., 2021; Anthropic, 2025a).

This raises a foundational question: In what sense can we think of these latent priors, or more generally, probability distributions, as agents? In reinforcement learning, an agent is often defined as an entity that acts to maximize a reward or utility function (Sutton & Barto, 2018; Russell & Norvig, 2020). But is this the only viable formalism? Can we unify this utility-maximization paradigm with a probabilistic view of agency?

Our work attempts to bridge this conceptual gap. We adopt a perspective inspired by active inference, wherein an agent is modeled as a probabilistic generative model (PGM) over outcomes or observations (Friston, 2010; Friston et al., 2017; Buckley et al., 2017; Parr & Friston, 2019; Da Costa et al., 2020; Koller & Friedman, 2009). That is, an agent predicts, and thus selectively favors, the outcomes it wants to bring about. For instance, a PGM encoding a preference for cats would have high probability mass over visual observations of petting a cat. The action of an agent in the world is then modeled by sampling from this biased outcome distribution. In this view, goals are formalized as distributions over desirable observations, and behavior arises from the agent's attempts to bring those observations about.

Formalizing agents as PGMs invites a further challenge: How do multiple generative models, each with their own probabilistic biases, combine to form a coherent, higher-level agent? This calls for a principled framework for modeling compositional agency, one that describes how disparate distributions can coalesce, both in the language of probability theory and in relation to utility-based frameworks. Developing this foundation is a central aim of our work.

**Related Work.** The theory of belief pooling asks how to aggregate many probability assessments into one and is anchored by Stone's "opinion pool" formulation and classic linear (arithmetic) and logarithmic (geometric) rules (Stone, 1961; Genest, 1984; Genest et al., 1986; Genest & Wagner, 1987). Beyond axiomatic foundations, proper scoring rules justify and learn aggregations: minimizing expected log score yields a log pool; minimizing quadratic/Brier score yields a linear pool; and weights can be estimated by maximizing average proper score (stacking) on held-out data (Gneiting & Raftery, 2007). Statistical and ML links include Bayesian model averaging as a linear pool with posterior model-probability weights (Hoeting et al., 1999) and *products of experts* as a log pool of model likelihoods (Hinton, 2002). Modern work generalizes these axioms from $\sigma$–algebras to general agendas of logically structured propositions–providing representation and impossibility theorems for neutrality/independence beyond standard event spaces (Dietrich & List, 2017)–and develops *principled weighting* for log pools (e.g., log-linear pooling of priors with weights chosen by marginal likelihood or predictive criteria), alongside practice-oriented syntheses on expert elicitation and performance-based weighting in risk analysis (Rufo et al., 2012; Clemen & Winkler, 1999). Similarly, bargaining theory provides parallel insights into axiomatic and cooperative equilibrium solutions that guide institutional system design (Kalai & Smorodinsky, 1975; Chatterjee & Samuelson, 1987).

## C. Foundations of Probabilistic Model Aggregation

Let $\mathcal{O}$ be a discrete set of outcomes. A collection of agents $\{1, \ldots, n\}$ each possesses a probability distribution $P_i$ on $\mathcal{O}$. We assign to each agent a non–negative influence weight $\beta_i$ satisfying $\sum_{i=1}^{n} \beta_i = 1$. The aggregated belief $P_C$ is a probability measure on $\mathcal{O}$ determined by the chosen pooling rule. A desired precondition is recursion; for instance, if an agent is split into identical subagents and its weight $\beta_i$ is distributed among them, the aggregated distribution should remain unchanged.

### C.1. Linear and Logarithmic Opinion Pools

**Definition 22** (Linear opinion pool). *Given agents $\{P_1, \ldots, P_n\}$ with weights $\{\beta_i\}$, the* linear opinion pool *is defined by*

$$P_C^{\text{lin}}(o) \;=\; \sum_{i=1}^{n} \beta_i \, P_i(o),$$

*for each $o \in \mathcal{O}$.*

The linear pool is simply a convex combination of the individual distributions. It is easy to compute and automatically produces a valid probability distribution without further normalization. As a mixture model, it can be sampled by first selecting an agent at random (according to $\beta_i$) and then sampling from that agent's distribution. When applied to purely the outcome space and not any intermediary causal variable, this results in a so-called random dictatorship. One disadvantage is that the linear pool does not allow any single agent to veto an outcome: even if an agent assigns zero probability to an undesirable outcome, the composition may still assign positive mass if others favor it.

**Definition 23** (Logarithmic opinion pool). *For the same set of agents and weights, the* logarithmic opinion pool *(sometimes called the log–linear pool) is defined by*

$$P_C^{\text{log}}(o) \;=\; \frac{1}{Z} \prod_{i=1}^{n} P_i(o)^{\beta_i},$$

*where the normalizing constant is*

$$Z \;=\; \sum_{o' \in \mathcal{O}} \prod_{i=1}^{n} P_i(o')^{\beta_i}.$$

In logarithmic pooling, probabilities are combined multiplicatively on a log scale. The rule is sometimes justified by the interpretation of probability as evidence: the logarithm of a probability is additive for independent pieces of information, and the pool aggregates these contributions linearly. An important property of the log pool is the *veto effect*: if any agent with non–zero weight assigns zero probability to an outcome, then the composition also assigns zero probability, ensuring that every member retains absolute veto power for outcomes they absolutely detest.

### C.2. Derivation via Divergence Minimization

The linear and logarithmic pools are not arbitrary constructs; each arises naturally as the solution to a convex optimization problem involving KL divergences. Recall that for distributions $P$ and $Q$ on $\mathcal{O}$, the KL divergence is defined by

$$\mathrm{KL}(P\|Q) \;=\; \sum_{o \in \mathcal{O}} P(o) \log \frac{P(o)}{Q(o)}.$$

**Proposition 24** (Logarithmic pool from forward KL). *The distribution $P_C$ that minimizes the weighted sum of forward divergences*

$$J(P_C) \;=\; \sum_{i=1}^{n} \beta_i \, \mathrm{KL}\big(P_C \,\|\, P_i\big)$$

*is precisely the logarithmic opinion pool.*

*Proof.* Expanding the objective yields

$$
\begin{aligned}
J(P_C) &= \sum_{i=1}^{n} \beta_i \sum_{o \in \mathcal{O}} P_C(o) \log \frac{P_C(o)}{P_i(o)} \\
&= \sum_{o} P_C(o) \log P_C(o) \Big(\sum_i \beta_i\Big) - \sum_o P_C(o) \sum_i \beta_i \log P_i(o) \\
&= \sum_o P_C(o) \log P_C(o) - \sum_o P_C(o) \log \Big(\prod_{i=1}^{n} P_i(o)^{\beta_i}\Big).
\end{aligned}
$$

Let $Q(o) = \prod_{i=1}^{n} P_i(o)^{\beta_i}/Z$ be the normalized log pool distribution for $Z$ in Definition 23. Then,

$$J(P_C) = \mathrm{KL}(P_C \,\|\, Q) - \log Z.$$

Since $\log Z$ does not depend on $P_C$, the objective is minimized exactly when $\mathrm{KL}(P_C \,\|\, Q)$ attains its minimum value of zero. This happens if and only if $P_C = Q$, which concludes the proof. $\square$

**Proposition 25** (Linear pool from reverse KL). *The distribution $P_C$ that minimizes the weighted sum of reverse divergences*

$$J(P_C) \;=\; \sum_{i=1}^{n} \beta_i \, \mathrm{KL}\big(P_i \,\|\, P_C\big)$$

*is the linear opinion pool.*

*Proof.* Expand the objective:

$$J(P_C) = \sum_{i=1}^{n} \beta_i \sum_{o \in \mathcal{O}} P_i(o) \log \frac{P_i(o)}{P_C(o)} = \sum_i \beta_i \sum_o P_i(o) \log P_i(o) - \sum_i \beta_i \sum_o P_i(o) \log P_C(o).$$

The first term is constant in $P_C$. Minimizing $J(P_C)$ is therefore equivalent to maximizing $\sum_i \sum_o \beta_i P_i(o) \log P_C(o)$ subject to $\sum_o P_C(o) = 1$. Introducing a Lagrange multiplier $\lambda$ for the normalization constraint and taking derivatives with respect to $P_C(o)$ gives

$$\sum_i \beta_i P_i(o) \cdot \frac{1}{P_C(o)} - \lambda = 0.$$

Solving for $P_C(o)$ shows that it must be proportional to $\sum_i \beta_i P_i(o)$. After enforcing $\sum_o P_C(o) = 1$, we obtain $P_C(o) = \sum_i \beta_i P_i(o)$, which is the linear pool. $\square$

We note that the two pooling mechanisms allow a form of trivial recursion. That is, if an agent $k$ is split into $m$ identical copies and its weight $\beta_k$ is divided uniformly among them, then under the linear pool the combined contribution is still $\beta_k P_k$; under the logarithmic pool the combined exponent is still $\beta_k$ because exponents add.

## C.3. From Probabilities to Utilities

Interpreting probability distributions as utility functions offers a deeper lens through which to understand aggregation rules. In particular, we define the notion of *epistemic utility*, or *relative utility*, which captures how an agent's preferences are reflected in the probabilities it assigns to outcomes. The core idea is that agents by definition do not merely describe the world; they encode value judgments within their predictions. For example, a hungry agent might assign higher probability to outcomes $o^* \in \mathcal{O}$ in which it obtains food, implicitly revealing its goals through its belief distribution.

Formally, given an agent with belief $P_i$ and a realized outcome $o^* \in \mathcal{O}$, the logarithmic score defines the agent's epistemic utility as

$$U_i^{\mathrm{epi}}(o^*) \;=\; \log P_i(o^*). \tag{2}$$

This scoring rule rewards agents for assigning high probability to the correct outcome. Thus, maximizing expected epistemic utility incentivizes the agent to learn calibrated models that reflect the actual distribution of outcomes as accurately as possible.

### C.3.1. SOFTMAX AND THE UTILITY–PROBABILITY CORRESPONDENCE

Let $U_i : \mathcal{O} \to \mathbb{R}$ be a utility function representing agent $i$'s preferences over outcomes. A standard way to map utilities into probabilities is via the softmax transformation

$$P_i(o) \;=\; \frac{\exp\big(U_i(o)\big)}{\sum_{o' \in \mathcal{O}} \exp\big(U_i(o')\big)}. \tag{3}$$

This mapping between utility landscapes and probability distributions plays two roles. First, one can recover an agent's beliefs from its utilities: exponentiating and normalizing yields the distribution. Conversely, the logarithm of a strictly positive distribution defines a utility function up to an additive constant. Indeed, setting $U_i(o) = \log P_i(o)$ in (3) returns $P_i$ exactly. This observation underlies our use of log probabilities or logits as epistemic utilities.

### C.3.2. AVERAGING UTILITIES AND THE LOGARITHMIC POOL

Suppose a group of agents forms a meta–agent by averaging their utility functions. A natural definition for the composition's relative utility is

$$U_C(o) \;=\; \sum_{i=1}^{n} \beta_i \, U_i(o).$$

Each agent's influence is reflected in its weight $\beta_i$. Substituting $U_i(o) = \log P_i(o)$ shows that $U_C$ is the weighted average of the individual log probabilities:

$$U_C(o) \;=\; \sum_{i=1}^{n} \beta_i \log P_i(o).$$

Applying the softmax transformation to $U_C$ yields

$$P_C(o) = \frac{\exp(U_C(o))}{\sum_{o' \in \mathcal{O}} \exp(U_C(o'))} = \frac{\exp\big(\sum_i \beta_i \log P_i(o)\big)}{\sum_{o'} \exp\big(\sum_i \beta_i \log P_i(o')\big)} = \frac{\prod_i P_i(o)^{\beta_i}}{\sum_{o'} \prod_i P_i(o')^{\beta_i}}.$$

Thus averaging utilities and then exponentiating reproduces the logarithmic opinion pool. This derivation offers a simple social–choice interpretation: the composition's belief (in log space) is the average of its members' beliefs. The linear pool, in contrast, corresponds to averaging distributions directly rather than averaging the log scores.

## D. Compositional Agents and Welfare Improvement

We now formalize when joining a composition benefits an individual agent in terms of its expected welfare. Throughout this section the outcome space $\mathcal{O}$ remains discrete, agents have beliefs $\{P_i\}$, and each agent $i$ possesses a welfare function $W_i : \mathcal{O} \to \mathbb{R}$ that measures the desirability of outcomes from its perspective. The composition's belief is constructed via the logarithmic pool with weights $\{\beta_i\}$. In this setting, under what conditions does an agent expect to do at least as well, according to its own welfare function, by adopting the composition's distribution?

**Definition 26** (Agent beliefs and welfare). *For each agent $i \in \{1, \ldots, n\}$, define*

1. *Probability distribution $P_i : \mathcal{O} \to [0,1]$ with $\sum_{o \in \mathcal{O}} P_i(o) = 1$;*

2. *Welfare function $W_i : \mathcal{O} \to \mathbb{R}$ expressing the utility the agent assigns to each outcome.*

**Definition 27** (Composition belief). *Fix non-negative weights $\beta_i$ summing to one. The composition's belief under the logarithmic pool is*

$$P(o) \;=\; \frac{1}{Z} \prod_{j=1}^{n} P_j(o)^{\beta_j}, \quad \text{where } Z = \sum_{o' \in \mathcal{O}} \prod_{j=1}^{n} P_j(o')^{\beta_j}.$$

**Definition 28** (Compositional agent). *An agent $i$ is said to be a* compositional agent *if its expected welfare under the composition belief is at least as large as under its own belief, that is,*

$$\mathbb{E}_{P_i}[W_i] \;\leq\; \mathbb{E}_P[W_i].$$

*We interpret this as meaning that agent $i$ benefits (or at least does not lose) by joining the composition.*

We characterize this condition in terms of the covariance between an agent's welfare and the ratio by which the composition reweights its distribution. We note that under this definition, a composition need not be composed only of compositional agents.

**Proposition 29** (Compositional agent condition). *Let $Q_i(o) := P(o)/P_i(o)$ denote the probability ratio between the composition and agent $i$. Then agent $i$ is a compositional agent if and only if*

$$\mathrm{Cov}_{P_i}\big(W_i, Q_i\big) \;\geq\; 0.$$

*Proof.* First note that

$$\mathbb{E}_P[W_i] = \sum_o P(o)W_i(o) = \sum_o P_i(o)Q_i(o)W_i(o) = \mathbb{E}_{P_i}[Q_i W_i].$$

Thus $\mathbb{E}_P[W_i] \geq \mathbb{E}_{P_i}[W_i]$ if and only if $\mathbb{E}_{P_i}[Q_i W_i] \geq \mathbb{E}_{P_i}[W_i]$. But $\mathbb{E}_{P_i}[Q_i] = 1$ since $\sum_o P(o) = 1$. Multiplying $\mathbb{E}_{P_i}[W_i]$ by this constant yields $\mathbb{E}_{P_i}[W_i]\,\mathbb{E}_{P_i}[Q_i] \leq \mathbb{E}_{P_i}[Q_i W_i]$. Rearranging gives precisely the non–negativity of the covariance between $W_i$ and $Q_i$ under $P_i$. $\square$

**Interpretation.** The quantity $Q_i(o) = P(o)/P_i(o)$ measures how the composition redistributes agent $i$'s probability mass: values greater than one indicate outcomes promoted by the group, while values less than one indicate outcomes demoted. The condition in Theorem 29 states that agent $i$ benefits if, on average, the composition places more weight on outcomes that $i$ values highly. Thus by joining the composition, agent $i$ is able to better realize outcomes it desires. Negative covariance means that the composition emphasizes outcomes that $i$ prefers less, leading to a loss of welfare.

It is natural to require that every member of a composition benefits. We call such a configuration a unanimously compositional group.

**Definition 30** (Unanimously compositional group). *A set of agents $\{1, \ldots, n\}$ is a* unanimously compositional group *if each agent $i$ is a compositional agent, i.e., $\mathbb{E}_{P_i}[W_i] \leq \mathbb{E}_P[W_i]$ for all $i \in \{1, \ldots, n\}$.*

A natural question is if such an ideal group exists. In Theorem 31, we answer in the affirmative.

**Theorem 31** (Existence of a unanimously compositional group). *For any integer $n \geq 2$, there exists a configuration of beliefs $\{P_i\}_{i=1}^n$, welfare functions $\{W_i\}$, and uniform weights $\beta_i = 1/n$ such that the logarithmic opinion pool makes every agent strictly better off.*

*Proof.* We give an explicit construction with $n$ outcomes $\mathcal{O} = \{o_1, \ldots, o_n\}$ and uniform weights $\beta_i = \frac{1}{n}$. Choosing $\epsilon \in (0, 1/n)$, define

$$P_i(o_j) = \begin{cases} 1 - (n-1)\,\epsilon, & j = i, \\ \epsilon, & j \neq i. \end{cases}$$

Thus each agent $i$ is nearly certain of outcome $o_i$ for $\epsilon \ll 1/n$. Let $C > 0$ be a large constant. Define

$$W_i(o_j) = \begin{cases} 0, & j = (i \bmod n) + 1, \\ -C, & \text{otherwise.} \end{cases}$$

Agent $i$ therefore most values outcome $o_{i+1}$, in a cyclic manner. For any outcome $o_k$,

$$\prod_{i=1}^{n} P_i(o_k)^{1/n} = \left(1 - (n-1)\epsilon\right)^{1/n} \epsilon^{(n-1)/n},$$

which is identical for all $k$. Hence after normalization, we thus have

$$P(o_k) = \frac{1}{n} \quad \forall\, k = 1, \ldots, n.$$

Without loss of generality, consider Agent 1, with standalone welfare

$$\mathbb{E}_{P_1}[W_1] = P_1(o_2) \cdot 0 + \left(P_1(o_1) + \sum_{j=3}^{n} P_1(o_j)\right)(-C) = -C + C\epsilon.$$

Under the pool,

$$\mathbb{E}_P[W_1] = \tfrac{1}{n} \cdot 0 + \tfrac{n-1}{n}(-C) = -C\frac{n-1}{n}.$$

Since $\epsilon < 1/n$, we have $-C\frac{n-1}{n} > -C + C\epsilon$. Thus $\mathbb{E}_P[W_1] > \mathbb{E}_{P_1}[W_1]$. By symmetry the same holds for every agent $i$. This completes the constructive proof. $\qquad\square$

Up to this point, we have left the definition of welfare $W_i$ deliberately open. In this section, we adopt the assumption that each agent's welfare function is identical to its epistemic utility, i.e., $W_i(o) = \log P_i(o)$. Under this interpretation, agents derive welfare directly from their predictive beliefs, which implicitly encode their goals, values, and preferences. The following sections investigate when compositions formed under this assumption lead to unanimous improvement, in other words, when every agent benefits from aggregation. As we will show, the possibility of such unanimous benefit depends critically on the cardinality of the outcome space. In particular, we find a stark contrast between the binary and multi-outcome settings.

## E. Possible and Impossible Compositions

We first examine the binary outcome case and show that unanimity cannot be achieved when welfare functions are logarithmic scores. We then prove that for outcome spaces of size at least three, there always exist beliefs and welfare functions that yield unanimous improvement under the logarithmic pool. Finally, we give an explicit analytic construction for such compositions.

### E.1. Impossibility in the Binary Case

Let $\mathcal{O} = \{o_A, o_B\}$. Suppose two agents have beliefs $P_1, P_2$ on $\mathcal{O}$ and welfare functions $W_i(o) = \log P_i(o)$. The composition's belief under any positive weights $\beta_1, \beta_2$ is again a distribution on two points. We claim that at most one agent can be strictly better off.

We start with the following result.

**Proposition 32** (Welfare gap identity for logarithmic utility)**.** *Let $\mathcal{O}$ be a finite outcome space. For any agent $i$ with belief distribution $P_i$, and any composition distribution $P$ (not necessarily equal to $P_i$), define the welfare gap*

$$\Delta_i \;=\; \Delta_{P_i}(P) \;=\; \mathbb{E}_P[\log P_i] - \mathbb{E}_{P_i}[\log P_i].$$

*Then this difference can be written as*

$$\Delta_i \;=\; H(P_i) - H(P) - \mathrm{KL}(P\|P_i),$$

*where $H(\cdot)$ denotes Shannon entropy and $\mathrm{KL}(\cdot\|\cdot)$ denotes the Kullback–Leibler divergence.*

*Proof.* By definition, the expected log utility under $P$ is

$$\mathbb{E}_P[\log P_i] = \sum_{o \in \mathcal{O}} P(o) \, \log P_i(o).$$

We now manipulate this expression by introducing and subtracting $\log P(o)$:

$$\sum_o P(o) \, \log P_i(o) = \sum_o P(o) \left[\log P(o) - \log \frac{P(o)}{P_i(o)}\right] = \sum_o P(o) \log P(o) - \sum_o P(o) \log \frac{P(o)}{P_i(o)}.$$

The first term is $-H(P)$ and the second term is $\mathrm{KL}(P\|P_i)$, so we conclude

$$\mathbb{E}_P[\log P_i] = -H(P) - \mathrm{KL}(P\|P_i).$$

Next, compute the expected log utility under the agent's own belief:

$$\mathbb{E}_{P_i}[\log P_i] = \sum_o P_i(o) \, \log P_i(o) = -H(P_i).$$

Substituting into the definition of $\Delta_i$ gives

$$\Delta_i = \mathbb{E}_P[\log P_i] - \mathbb{E}_{P_i}[\log P_i] = \left[-H(P) - \mathrm{KL}(P\|P_i)\right] - (-H(P_i)) = H(P_i) - H(P) - \mathrm{KL}(P\|P_i),$$

as claimed. $\square$

We then have the following result.

**Theorem 33** (Binary-outcome impossibility under log-welfare). *Let $\mathcal{O} = \{o_A, o_B\}$. Suppose two agents have distinct beliefs $P_i(o_A) = x_i \in (0,1)$, $i = 1, 2$, and welfare functions $W_i(o) = \log P_i(o)$. Let $\beta_1, \beta_2 > 0$ with $\beta_1 + \beta_2 = 1$ and form the log-linear pool*

$$P(o_A) = x = \frac{x_1^{\beta_1} \, x_2^{\beta_2}}{x_1^{\beta_1} \, x_2^{\beta_2} + (1 - x_1)^{\beta_1} \, (1 - x_2)^{\beta_2}} \, .$$

*Define each agent's welfare gap with respect to variable $x$:*

$$\Delta_i(x) = \mathbb{E}_P[\log P_i] \; - \; \mathbb{E}_{P_i}[\log P_i].$$

*Then, there is no choice of $\beta_1, \beta_2$ satisfying $\Delta_i \geq 0$ for both $i = 1, 2$ with at least one strict inequality.*

*Proof.* Using binary entropy and KL-divergence,

$$H(u) = -u \log u - (1 - u) \log(1 - u), \quad d(x\|x_i) = x \log \frac{x}{x_i} + (1 - x) \log \frac{1 - x}{1 - x_i}.$$

We have by Proposition 32 that

$$\Delta_i(x) = H(x_i) - H(x) - d(x\|x_i).$$

A direct calculation gives

$$H(x_i) - H(x) = \left[-x_i \log x_i - (1 - x_i) \log(1 - x_i)\right] - \left[-x \log x - (1 - x) \log(1 - x)\right],$$

so

$$\Delta_i(x) = x \log x + (1 - x) \log(1 - x) - x_i \log x_i - (1 - x_i) \log(1 - x_i) - \left[x \log \frac{x}{x_i} + (1 - x) \log \frac{1 - x}{1 - x_i}\right].$$

Collecting terms,

$$\Delta_i(x) = \left[x \log x - x \log \frac{x}{x_i}\right] + \left[(1 - x) \log(1 - x) - (1 - x) \log \frac{1 - x}{1 - x_i}\right] - \left[x_i \log x_i + (1 - x_i) \log(1 - x_i)\right].$$

But note
$$x \log x - x \log \frac{x}{x_i} = x \log x_i, \quad (1-x) \log(1-x) - (1-x) \log \frac{1-x}{1-x_i} = (1-x) \log(1-x_i).$$

Hence
$$\Delta_i(x) = x \log x_i + (1-x) \log(1-x_i) - \big[ x_i \log x_i + (1-x_i) \log(1-x_i) \big].$$

Factor to obtain the succinct form
$$\Delta_i(x) = (x - x_i) \big[ \log x_i - \log(1-x_i) \big] = (x - x_i) \log \frac{x_i}{1-x_i}.$$

In particular, $\Delta_i(x_i) = 0$, and since $\log \frac{x_i}{1-x_i} \neq 0$ whenever $x_i \neq \frac{1}{2}$, the sign of $\Delta_i(x)$ changes precisely once as we vary $x \in [0, 1]$.

Now, write
$$A = x_1^{\beta_1} x_2^{\beta_2}, \quad B = (1-x_1)^{\beta_1} (1-x_2)^{\beta_2}, \quad x = \frac{A}{A+B}.$$

Define the likelihood-odds function $h(u) = u/(1-u)$. Then
$$G := \frac{x}{1-x} = \frac{A}{B} = \big[ h(x_1) \big]^{\beta_1} \big[ h(x_2) \big]^{\beta_2},$$

the weighted geometric mean of $h(x_1)$ and $h(x_2)$. Taking logarithms on both sides, we have
$$\log G = \beta_1 \log h(x_1) + (1 - \beta_1) \log h(x_2).$$

Without loss of generality, assume that $x_1 < x_2$. Clearly $\log G$ is a decreasing function with respect to $\beta_1$ as $h(u)$ is strictly increasing on $(0, 1)$. This gives that
$$\log h(x_1) < \log G < \log h(x_2),$$

and exponentiating both sides gives
$$h(x_1) < \frac{x}{1-x} < h(x_2).$$

Applying the inverse $h^{-1}(v) = v/(1+v)$, which is also strictly increasing, yields
$$x_1 \; < \; x \; < \; x_2.$$

The final step is to analyze the signs of $\Delta_1(x)$ and $\Delta_2(x)$. Since we have established that $x_1 < x < x_2$, it follows that the term $(x - x_1)$ is always positive and $(x - x_2)$ is always negative. The sign of the log-odds term, $\log \frac{x_i}{1-x_i}$, depends on whether $x_i$ is greater or less than $1/2$. We proceed with a case analysis. We first analyze the case in which beliefs $x_1, x_2$ are on the same side of $1/2$. If $1/2 < x_1 < x_2$, then $\log \frac{x_1}{1-x_1} > 0$ and $\log \frac{x_2}{1-x_2} > 0$. This yields:
$$\Delta_1(x) = \underbrace{(x - x_1)}_{>0} \underbrace{\log \frac{x_1}{1-x_1}}_{>0} > 0 \quad \text{and} \quad \Delta_2(x) = \underbrace{(x - x_2)}_{<0} \underbrace{\log \frac{x_2}{1-x_2}}_{>0} < 0.$$

Similarly, if $x_1 < x_2 < 1/2$, then $\log \frac{x_1}{1-x_1} < 0$ and $\log \frac{x_2}{1-x_2} < 0$. This yields:
$$\Delta_1(x) = \underbrace{(x - x_1)}_{>0} \underbrace{\log \frac{x_1}{1-x_1}}_{<0} < 0 \quad \text{and} \quad \Delta_2(x) = \underbrace{(x - x_2)}_{<0} \underbrace{\log \frac{x_2}{1-x_2}}_{<0} > 0.$$

In either subcase, one agent's welfare improves while the other's worsens. Otherwise, the beliefs must be on opposite sides of $1/2$. If $x_1 < 1/2 < x_2$, then $\log \frac{x_1}{1-x_1} < 0$ and $\log \frac{x_2}{1-x_2} > 0$. This yields:
$$\Delta_1(x) = \underbrace{(x - x_1)}_{>0} \underbrace{\log \frac{x_1}{1-x_1}}_{<0} < 0 \quad \text{and} \quad \Delta_2(x) = \underbrace{(x - x_2)}_{<0} \underbrace{\log \frac{x_2}{1-x_2}}_{>0} < 0.$$

In this case, the welfare of both agents worsens. Thus in all possible cases, at least one agent incurs a strictly negative welfare gap. It is therefore impossible for both $\Delta_1$ and $\Delta_2$ to be non-negative with at least one being strictly positive. $\quad \square$

The binary impossibility reflects a tension inherent to averaging on a log scale: increasing one agent's log likelihood necessarily decreases another's when only two alternatives exist.

### E.2. Abstract Existence for Larger Outcome Spaces

When the outcome space has at least three elements, unanimously beneficial compositions do exist. We first establish existence in a constructive way by exhibiting a family of beliefs and welfare functions that guarantee positive welfare gains for all agents.

**Theorem 34** (Existence of unanimously compositional groups). *Let $n \geq 2$ and suppose $|\mathcal{O}| \geq 3$. Then there exist probability distributions $\{P_i\}_{i=1}^{n}$ on $\mathcal{O}$, welfare functions $W_i(o) = \log P_i(o)$, and equal weights $\beta_i = 1/n$, such that under the logarithmic pool every agent strictly benefits: $\mathbb{E}_P[W_i] > \mathbb{E}_{P_i}[W_i]$ for all $i$.*

*Proof.* We provide an explicit construction in the next subsection, showing that such distributions exist for any $n \geq 2$ when the number of outcomes is at least three. The key idea is to choose beliefs that assign small probability to distinct outcomes while concentrating mass on a common "default" outcome. Under the log-pool, the default outcome becomes highly likely, increasing each agent's entropy and thus its expected log likelihood. We will quantify this effect below. $\qquad\square$

### E.2.1. EXPLICIT ANALYTIC CONSTRUCTION

We now present a concrete family of distributions that realize the existence theorem. Fix an integer $n \geq 2$ and choose an outcome space $\mathcal{O} = \{o_0, o_1, o_2, \ldots, o_n\}$ of size $n + 1$. Select a small parameter $\varepsilon \in \left(0, \frac{1}{4}\right)$ and define two auxiliary quantities

$$\alpha = \varepsilon, \quad \delta = \varepsilon^{n+1}.$$

We define each agent $i \in \{1, \ldots, n\}$ to be weakly suggestive on seeing a private outcome $o_i$, almost certain about a shared base outcome $o_0$, and very unlikely to see the remaining outcomes. Specifically, let

$$P_i(o) = \begin{cases} \alpha, & \text{if } o = o_i, \\ 1 - \alpha - (n-1)\delta, & \text{if } o = o_0, \\ \delta, & \text{otherwise.} \end{cases}$$

Because $\alpha + (n-1)\delta + \left(1 - \alpha - (n-1)\delta\right) = 1$, this defines a valid distribution. The welfare function for each agent is $W_i(o) = \log P_i(o)$. We assign uniform weights $\beta_i = 1/n$ and form the composition belief under the logarithmic pool. We claim that for $\varepsilon$ sufficiently small, every agent's expected log likelihood strictly increases.

**Theorem 35** (Analytic construction for unanimity). *Let $n \geq 2$ and define beliefs $P_i$ as above. Form the logarithmic pool with weights $\beta_i = 1/n$ and denote the resulting distribution by $P$. There exists $\varepsilon_0 > 0$ such that for all $\varepsilon \in (0, \varepsilon_0)$, the expected welfare difference $\Delta_i = \mathbb{E}_P[\log P_i] - \mathbb{E}_{P_i}[\log P_i]$ is strictly positive for every agent $i$. In particular, the set of agents is unanimously compositional.*

*Proof.* We estimate orders of magnitude to show that positive gains dominate negative contributions for sufficiently small $\varepsilon$. For each outcome $o \in \mathcal{O}$ the normalized log pool assigns weight

$$R(o) = \prod_{j=1}^{n} P_j(o)^{1/n}.$$

If $o = o_0$, then each factor contributes $1 - \alpha - (n-1)\delta$, thus we obtain

$$R(o_0) = (1 - \alpha - (n-1)\delta)^{n \cdot 1/n} = 1 - \alpha - (n-1)\delta.$$

If $o = o_i$ for some agent $i$, then agent $i$ assigns probability $\alpha$ while all other $n - 1$ agents assign probability $\delta$. Thus $R(o_i) = \alpha^{1/n} \delta^{1-1/n} = \varepsilon^n$. Meanwhile, we have

$$R(o_0) = 1 - \alpha - (n-1)\delta = 1 - \varepsilon + o(\varepsilon).$$

Normalizing gives

$$P(o_0) = \frac{R(o_0)}{R(o_0) + \sum_{i=1}^{n} R(o_i)} = 1 - O(\varepsilon^n),$$

where the second term in the denominator is of order $n\varepsilon^n$ and thus much smaller than $\varepsilon$. Consequently, $P$ assigns nearly all of its mass to $o_0$ and vanishingly small mass (of order $\varepsilon$) to each of the other outcomes.

The entropy of $P_i$ is dominated by the uncertainty between $o_0$ (which has probability roughly $1 - \varepsilon$) and $o_i$ (probability $\varepsilon$), with negligible contribution from the $n - 1$ rare events. A short calculation shows

$$H(P_i) = -\big(1 - \alpha - (n-1)\delta\big)\log\big(1 - \alpha - (n-1)\delta\big) - \alpha\log\alpha + O(\delta\log\tfrac{1}{\delta}).$$

Using $\alpha = \varepsilon$ and $\delta = \varepsilon^{n+1}$, we have $H(P_i) = \Theta\big(\varepsilon\log\tfrac{1}{\varepsilon}\big)$. By contrast, the composition belief $P$ puts probability $1 - O(\varepsilon^n)$ on $o_0$ and distributes the remaining mass evenly among $n$ outcomes of order $\varepsilon^n$. Hence $H(P) = O\big(\varepsilon^n\log\tfrac{1}{\varepsilon}\big)$, which is of much smaller order than $H(P_i)$. Therefore, the entropy difference $H(P_i) - H(P)$ scales like $\varepsilon\log(1/\varepsilon)$.

It remains to bound $D_{\mathrm{KL}}(P\|P_i)$. Since $P$ places mass $1 - \theta$ on $o_0$ with $\theta = O(\varepsilon^n)$, we may write

$$D_{\mathrm{KL}}(P\|P_i) = (1-\theta)\log\frac{1-\theta}{1-\alpha-(n-1)\delta} + \frac{\theta}{n}\log\frac{\theta}{n\alpha} + \frac{(n-1)\theta}{n}\log\frac{\theta}{n\delta}.$$

Using expansions $\log(1 - \theta) = -\theta + O(\theta^2)$ and noting that $1 - \alpha - (n-1)\delta = 1 - \varepsilon + o(\varepsilon)$, one finds $D_{\mathrm{KL}}(P\|P_i) = O(\varepsilon)$. In particular, the KL divergence is of lower order than the entropy gap when $\varepsilon$ is small. That is, recall that $\Delta_i = H(P_i) - H(P) - D_{\mathrm{KL}}(P\|P_i)$. Combining the estimates above shows that $H(P_i) - H(P)$ is positive and dominates $D_{\mathrm{KL}}(P\|P_i)$ for $\varepsilon$ sufficiently small. Hence $\Delta_i > 0$ for all agents. Selecting $\varepsilon_0$ small enough completes the proof. $\square$

The construction in Theorem 35 generalizes to arbitrary weights and larger outcome spaces, showing that diversity of possible outcomes enables mutually beneficial compositions.

### E.2.2. GENERALIZATION TO ARBITRARY WEIGHTS

We now show that the same analytic construction for the $P_j$ yields unanimously beneficial compositions for any non-degenerate choice of weights. We allow an *arbitrary* weight vector $\beta = (\beta_1, \ldots, \beta_n)$ with $\beta_i \geq 0$ and $\sum_{i=1}^n \beta_i = 1$. The construction is trivial if some $\beta_i = 1$ and the rest 0, thus we assume $\max_i \beta_i < 1$.

**Theorem 36** (Unanimity for general weights). *There exists $\varepsilon_0 > 0$ depending only on $\beta$ and $n$ such that for all $0 < \varepsilon < \varepsilon_0$, $\Delta_i > 0$ where $P$ is the logarithmic pool $P(o) = \frac{1}{Z}\prod_{j=1}^n P_j(o)^{\beta_j}$.*

*Proof.* Write $R(o) = \prod_{j=1}^n P_j(o)^{\beta_j}$ for the unnormalized weight and $Z = \sum_{o\in\mathcal{O}} R(o)$ for the partition function. Throughout, we have $c_{\min} := \min_i\big((n+1) - n\beta_i\big) > 1$ because $\max_i \beta_i < 1$. At the shared outcome $o_0$, every agent contributes $1 - \alpha - (n-1)\delta = 1 - \varepsilon + O(\varepsilon^{n+1})$, so

$$R(o_0) = \exp\Big(\sum_{j=1}^n \beta_j \log\big(1 - \varepsilon + O(\varepsilon^{n+1})\big)\Big) = 1 - \varepsilon + O(\varepsilon^{n+1}),$$

because $\sum_j \beta_j = 1$. For a private outcome $o_i$, the $i$-th agent contributes $\alpha = \varepsilon$ and each of the other $n - 1$ agents contributes $\delta = \varepsilon^{n+1}$. We therefore have

$$R(o_i) = \varepsilon^{\beta_i}\big(\varepsilon^{n+1}\big)^{1-\beta_i} = \varepsilon^{\beta_i + (n+1)(1-\beta_i)} = \varepsilon^{c_i}, \qquad c_i := (n+1) - n\beta_i > 1.$$

Since $R(o_i) = \varepsilon^{c_i} = o(\varepsilon)$, the partition function is

$$Z = R(o_0) + \sum_{i=1}^n R(o_i) = 1 - \varepsilon + \sum_{i=1}^n \varepsilon^{c_i} + O(\varepsilon^{n+1}).$$

Dividing by $Z$ gives

$$P(o_0) = \frac{1 - \varepsilon + O(\varepsilon^{n+1})}{1 - \varepsilon + \sum_i \varepsilon^{c_i} + O(\varepsilon^{n+1})} = 1 - \Theta\big(\varepsilon^{c_{\min}}\big),$$

because the numerator and denominator differ only by $\Theta(\varepsilon^{c_{\min}})$, and $c_{\min} > 1$. Likewise,

$$P(o_i) = \frac{\varepsilon^{c_i}}{1 - \varepsilon + o(1)} = \Theta\big(\varepsilon^{c_i}\big).$$

For agent $i$, only $o_0$ and $o_i$ carry mass larger than $\delta = \varepsilon^{n+1}$, so as $\varepsilon \to 0^+$,

$$
\begin{aligned}
H(P_i) &= -(1 - \varepsilon) \log(1 - \varepsilon) - \varepsilon \log \varepsilon - (n-1)\delta \log \delta \\
&= \varepsilon \log \tfrac{1}{\varepsilon} + O(\varepsilon) + O\big((n^2 - 1)\varepsilon^{n+1} \log \tfrac{1}{\varepsilon}\big) \\
&= \Theta\big(\varepsilon \log \tfrac{1}{\varepsilon}\big) > 0.
\end{aligned}
$$

The leakage mass is $\theta := 1 - P(o_0) = \Theta(\varepsilon^{c_{\min}})$. It is split across at most $n$ outcomes, each of size $\Theta(\varepsilon^{c_i})$. The entropy is

$$
\begin{aligned}
H(P) &= -P(o_0) \log P(o_0) - \sum_{i=1}^n P(o_i) \log P(o_i) \\
&= -(1 - \theta) \log(1 - \theta) - \sum_{i=1}^n P(o_i) \log P(o_i).
\end{aligned}
$$

For the $o_0$ term, we have
$$
-(1 - \theta) \log(1 - \theta) = \theta + O(\theta^2) = O(\theta).
$$

For the leakage terms, each $P(o_i) = \Theta(\varepsilon^{c_i})$ with $c_i \geq c_{\min}$, so we may write asymptotically

$$
P(o_i) = K_i \, \varepsilon^{c_i}
$$

for some constant $K_i > 0$ independent of $\varepsilon$. Then

$$
\log P(o_i) = \log K_i + c_i \log \varepsilon = \log K_i - c_i \log \tfrac{1}{\varepsilon}.
$$

Therefore

$$
\begin{aligned}
-P(o_i) \log P(o_i) &= -K_i \, \varepsilon^{c_i} \Big( \log K_i - c_i \log \tfrac{1}{\varepsilon} \Big) \\
&= c_i \, K_i \, \varepsilon^{c_i} \log \tfrac{1}{\varepsilon} \; - \; K_i \, \varepsilon^{c_i} \log K_i.
\end{aligned}
$$

As $\varepsilon \to 0^+$, the constant term $K_i \, \varepsilon^{c_i} \log K_i$ is dominated by the $\varepsilon^{c_i} \log \tfrac{1}{\varepsilon}$ term, so

$$
-P(o_i) \log P(o_i) = \Theta\big(\varepsilon^{c_i} \log \tfrac{1}{\varepsilon}\big).
$$

Summing over at most $n$ such terms yields

$$
\sum_{i=1}^n -P(o_i) \log P(o_i) = \Theta(\varepsilon^{c_{\min}} \log \tfrac{1}{\varepsilon}).
$$

Combining the above gives that

$$
H(P) = O(\theta) + \Theta(\varepsilon^{c_{\min}} \log \tfrac{1}{\varepsilon}) = \Theta(\varepsilon^{c_{\min}} \log \tfrac{1}{\varepsilon}),
$$

and since $c_{\min} > 1$, this term decays at the order of $o(\varepsilon \log \tfrac{1}{\varepsilon})$. Now, write $p_0 := P(o_0) = 1 - \theta$ and $p_i := P(o_i) = \Theta(\varepsilon^{c_i})$. Then

$$
D_{\mathrm{KL}}(P \| P_i) = p_0 \log \frac{p_0}{P_i(o_0)} + p_i \log \frac{p_i}{P_i(o_i)} + \sum_{o \notin \{o_0, o_i\}} P(o) \log \frac{P(o)}{\delta}.
$$

We control the first term by noting $P_i(o_0) = 1 - \varepsilon + O(\varepsilon^{n+1})$. Using $\log(1 - \theta) = -\theta + O(\theta^2)$ and the fact that both $p_0$ and $P_i(o_0)$ are close to 1, we have

$$
p_0 \log \frac{p_0}{P_i(o_0)} = p_0 \big[\log(1 - \theta) - \log(1 - \varepsilon + O(\varepsilon^{n+1}))\big] = O(\varepsilon).
$$

For the second term, we have $P_i(o_i) = \varepsilon$ which gives

$$
p_i \log \frac{p_i}{P_i(o_i)} = \Theta(\varepsilon^{c_i}) \log \frac{\Theta(\varepsilon^{c_i})}{\varepsilon} = \Theta(\varepsilon^{c_i} \log \varepsilon^{c_i - 1}) = \Theta(\varepsilon^{c_i} \log \tfrac{1}{\varepsilon}).
$$

Here, we used that $c_i - 1 > 0$ is a constant. For the third term, there are at most $n - 1$ outcomes with $P(o) = \Theta(\varepsilon^{c_k})$, and for each such $o$, $P_i(o) = \delta = \varepsilon^{n+1}$. Thus

$$P(o) \log \frac{P(o)}{\delta} = \Theta(\varepsilon^{c_k}) \log \frac{\Theta(\varepsilon^{c_k})}{\varepsilon^{n+1}} = \Theta(\varepsilon^{c_k} \log \tfrac{1}{\varepsilon}).$$

Since $c_k \geq c_{\min} > 1$, all contributions are $O(\varepsilon^{c_{\min}} \log \frac{1}{\varepsilon})$. Adding the three terms gives

$$D_{\mathrm{KL}}(P\|P_i) = O(\varepsilon) + O(\varepsilon^{c_{\min}} \log \tfrac{1}{\varepsilon}).$$

Using $\Delta_i = H(P_i) - H(P) - D_{\mathrm{KL}}(P\|P_i)$ from Proposition 32,

$$\Delta_i = \underbrace{\Theta(\varepsilon \log \tfrac{1}{\varepsilon})}_{H(P_i)} - \underbrace{o(\varepsilon \log \tfrac{1}{\varepsilon})}_{H(P)} - \underbrace{\left(O(\varepsilon) + O(\varepsilon^{c_{\min}} \log \tfrac{1}{\varepsilon})\right)}_{D_{\mathrm{KL}}} = \varepsilon \log \tfrac{1}{\varepsilon}\left(\Theta(1) - o(1)\right) - O(\varepsilon).$$

Because $c_{\min} > 1$, the negative term is of higher order in $\varepsilon$; hence $\Delta_i > 0$ for all $i$ when $0 < \varepsilon < \varepsilon_0(\beta)$ with $\varepsilon_0$ chosen small enough. In other words, since $\varepsilon \log(1/\varepsilon) \gg \varepsilon$ as $\varepsilon \to 0^+$, there exists $\varepsilon_0 > 0$ (depending only on $\beta, n$ through the constants above) such that $\Delta_i > 0$ for all $i$ whenever $0 < \varepsilon < \varepsilon_0$. $\qquad\square$

### E.3. Impossibility of Strictly Unanimous Benefit under Linear Pooling

In this section we show that if each agent's welfare is defined by the log-score of its *own* belief and the group aggregates beliefs via the *linear* opinion pool, then no strictly unanimously beneficial composition can exist.

**Theorem 37** (Impossibility under linear opinion pool). *Let $n \geq 2$ be the number of agents. For each $i = 1, \ldots, n$, let*

$$P_i : \mathcal{O} \to [0, 1], \quad \sum_{o \in \mathcal{O}} P_i(o) = 1,$$

*be distinct probability distributions, and let weights $\beta_i > 0$ satisfy $\sum_{i=1}^{n} \beta_i = 1$. Suppose each agent's welfare function is the log-score of its own belief, $W_i(o) = \log P_i(o)$. Then, it is impossible to have*

$$\mathbb{E}_P[W_i] \geq \mathbb{E}_{P_i}[W_i] \quad \text{for all } i,$$

*with* strict *inequality for at least one $i$. In other words, no strictly unanimously beneficial composition exists.*

*Proof.* Recall proposition 32. Assume for the sake of contradiction, that $\Delta_i \geq 0$ for every $i$. Then also $\sum_{i=1}^{n} \beta_i \Delta_i \geq 0$. However, we have that

$$\sum_{i=1}^{n} \beta_i \Delta_i = \sum_{i=1}^{n} \beta_i \left[ H(P_i) - H(P) - D_{\mathrm{KL}}(P\|P_i) \right]$$

$$= \left( \sum_{i=1}^{n} \beta_i H(P_i) \right) - H(P) - \sum_{i=1}^{n} \beta_i D_{\mathrm{KL}}(P\|P_i).$$

Because $H$ is strictly concave on the simplex,

$$H(P) = H\left( \sum_{i=1}^{n} \beta_i P_i \right) > \sum_{i=1}^{n} \beta_i H(P_i)$$

whenever the $P_i$ are not all identical. Hence $\sum_i \beta_i H(P_i) - H(P) < 0$. Each term $D_{\mathrm{KL}}(P\|P_i) \geq 0$, so this gives that

$$\sum_{i=1}^{n} \beta_i \Delta_i = \left[ \sum_i \beta_i H(P_i) - H(P) \right] - \sum_i \beta_i D_{\mathrm{KL}}(P\|P_i) < 0.$$

This contradicts $\sum_i \beta_i \Delta_i \geq 0$. Therefore our assumption that $\Delta_i \geq 0$ for all $i$ must be false: at least one $\Delta_i < 0$. We note that in the degenerate case $P_1 = \cdots = P_n$, we have $P = P_i$ for every $i$ which gives $H(P) = H(P_i)$ and $D_{\mathrm{KL}}(P\|P_i) = 0$. Hence each $\Delta_i = 0$. No agent strictly gains, and the composition is *not* strictly beneficial. $\qquad\square$

Therefore, a linear pooling does not admit a strictly compositional group under the epistemic utility. However, a logarithmic pool does allow for a strictly unanimous compositional group.

## F. Recursing on Compositional Agents for Scale-Free Theory

**Recursion on Agents and Subagent Decomposition.** A natural question is whether the compositional property is preserved under recursive decomposition. Suppose a compositional agent is split into a fixed number of *subagents*, each representing a distinct component of its preferences (e.g., an individual's drives to eat, sleep, or play). Does compositionality of the parent agent imply that each subagent is also compositional with respect to the same group?

In general, the answer is negative: decomposition can destroy compositionality. An individual may align with a group overall, yet certain subagents–such as the drive to eat–may diverge from the group's objectives (e.g., productivity), even while remaining consistent with the parent's broader goals.

From a scale-free perspective, a fully recursive theory would require that an agent be decomposable into an arbitrary number of subagents, while preserving both (i) recovery of the original agent via log-pooling and (ii) preservation of the unanimously compositional structure. We will show that property (ii) does not hold in general: recursive decomposition can eliminate unanimity. This raises a natural direction for future work: characterizing conditions under which subagents $P_1, \ldots, P_n$ can be constructed so that their aggregation remains unanimously compositional, even when the subagents are not known *a priori*.

In Theorem 38, we formalize the property that an agent can be decomposed into a fixed number of subagents, where this number may be arbitrarily large. Conceptually, one might think of a human being as possessing a seemingly unbounded set of desires or goals–some overlapping, others highly correlated. Our framework is designed to accommodate such decompositions. For example, an individual's epistemic state might be represented as the aggregation of subagents corresponding to desires for nourishment, social interaction, intellectual stimulation, and so forth, each contributing to the overall belief distribution.

Building upon Theorem 38, we extend the analysis in Theorem 39 to the case where the subagents are pairwise distinct, representing genuinely different desires. This constraint captures scenarios in which each subagent embodies a unique epistemic or utility perspective, ensuring that the decomposition meaningfully differentiates between components of the agent's overall decision-making process.

**Theorem 38** (Non-trivial log–pool decomposition always exists when $P$ is strictly positive). *Let $\mathcal{O}$ be a finite set, let $P$ be a probability distribution on $\mathcal{O}$ such that $P(o) > 0$ for every $o \in \mathcal{O}$, and fix weights $\beta_1, \ldots, \beta_n \geq 0$ with $\sum_{i=1}^n \beta_i = 1$ and at least two $\beta_i$ strictly positive. Then there exist distributions $P_1, \ldots, P_n$ on $\mathcal{O}$ distinct from $P$ satisfying*

$$P(o) = \frac{1}{Z} \prod_{i=1}^n P_i(o)^{\beta_i}, \qquad Z = \sum_{o \in \mathcal{O}} \prod_{i=1}^n P_i(o)^{\beta_i}.$$

*Proof.* Pick any reference distribution $Q$ with the *same support* as $P$ and $Q \neq P$ (e.g. the uniform distribution on $\mathcal{O}$). Let $\gamma := \sum_{i=2}^n \beta_i$ and assume $\beta_1 > 0$; the argument is symmetric if a different index carries the complementary weight.

Define

$$\tilde{P}_1(o) = \frac{P(o)^{1/\beta_1} Q(o)^{-\gamma/\beta_1}}{\sum_{o' \in \mathcal{O}} P(o')^{1/\beta_1} Q(o')^{-\gamma/\beta_1}}, \qquad P_i := Q \ (i = 2, \ldots, n).$$

Each $\tilde{P}_1$ is well-defined and strictly positive because $P$ and $Q$ are strictly positive. Let $Z_1$ denote the denominator in $\tilde{P}_1$ and set $P_1 = \tilde{P}_1$. Then for every outcome $o$

$$\sum_{i=1}^n \beta_i \log P_i(o) = \beta_1 \left[ \tfrac{1}{\beta_1} \log P(o) - \tfrac{\gamma}{\beta_1} \log Q(o) - \log Z_1 \right] + \gamma \log Q(o)$$

$$= \log P(o) - \beta_1 \log Z_1 \ = \ \log P(o) + C,$$

where $C := -\beta_1 \log Z_1$ is a constant *independent of o*. Exponentiating and normalizing by $Z := e^{-C}$ gives the desired identity. Because $P_1$ was built from both $P$ and $Q$, it differs from each, and we have $P \neq P_i$ for all $i$. $\square$

**Theorem 39** (Log–pool factorization with pairwise-distinct components). *Let $\mathcal{O}$ be a finite set and let $P$ be a probability distribution on $\mathcal{O}$ with $P(o) > 0$ for all $o \in \mathcal{O}$. Fix weights $\beta_1, \ldots, \beta_n \geq 0$ with $\sum_{i=1}^n \beta_i = 1$, and assume at least two $\beta_i$*

*are strictly positive. Then there exist probability distributions $P_1, \ldots, P_n$ on $\mathcal{O}$ such that*

$$P(o) = \frac{1}{Z} \prod_{i=1}^{n} P_i(o)^{\beta_i} \quad \text{for all } o \in \mathcal{O}, \qquad Z = \sum_{o \in \mathcal{O}} \prod_{i=1}^{n} P_i(o)^{\beta_i},$$

*with the additional properties that $P \neq P_i$ for every $i$ and $P_i \neq P_j$ whenever $i \neq j$.*

*Proof.* Pick any $n-1$ *pairwise distinct* strictly positive distributions $Q_2, \ldots, Q_n$ on $\mathcal{O}$ such that none equals $P$; for instance, start from a positive reference $R \neq P$ and set

$$Q_i(o) \ \propto \ R(o) \, e^{\varepsilon_i h_i(o)} \quad (o \in \mathcal{O}),$$

where each $h_i : \mathcal{O} \to \mathbb{R}$ has at least two distinct values and $\sum_o R(o) h_i(o) = 0$. We may choose small, distinct $\varepsilon_i \neq 0$ to ensure $Q_i$ are strictly positive and pairwise different.

Let $\gamma := \sum_{i=2}^{n} \beta_i$ (which gives $\gamma > 0$ by assumption). Define the weighted geometric mean

$$Q_\star(o) \ := \ \exp\Big(\frac{1}{\gamma} \sum_{i=2}^{n} \beta_i \log Q_i(o)\Big), \qquad \text{so that} \quad \sum_{i=2}^{n} \beta_i \log Q_i = \gamma \log Q_\star.$$

Now define $P_1$ by

$$P_1(o) \ = \ \frac{P(o)^{1/\beta_1} \, Q_\star(o)^{-\gamma/\beta_1}}{\sum_{u \in \mathcal{O}} P(u)^{1/\beta_1} \, Q_\star(u)^{-\gamma/\beta_1}} \quad (o \in \mathcal{O}).$$

Then, for every $o \in \mathcal{O}$,

$$\sum_{i=1}^{n} \beta_i \log P_i(o) = \beta_1 \left( \tfrac{1}{\beta_1} \log P(o) - \tfrac{\gamma}{\beta_1} \log Q_\star(o) - \log Z_1 \right) + \sum_{i=2}^{n} \beta_i \log Q_i(o) = \log P(o) - \beta_1 \log Z_1,$$

where $Z_1$ is the normalizing constant in $P_1$. Exponentiating and renormalizing shows $\prod_i P_i(o)^{\beta_i} = e^{-\beta_1 \log Z_1} P(o)$, hence the claimed factorization holds with $Z = e^{-\beta_1 \log Z_1}$.

It remains to ensure distinctness. By construction, $Q_2, \ldots, Q_n$ are pairwise distinct and differ from $P$. If, for some $j \geq 2$, we had $P_1 = Q_j$, then as the $Q_i$ were chosen with free continuous parameters $(\varepsilon_i)$, any generic small perturbation of the $\varepsilon_i$ keeps the construction valid while ensuring $P_1 \neq Q_j$ for all $j \geq 2$. Similarly, we can ensure $P_1 \neq P$, avoided by the same generic choice. Therefore we may choose $Q_2, \ldots, Q_n$ so that $P_1$ is distinct from each $Q_j$ and from $P$, completing the proof. $\square$

**Remark 40.** *If some $\beta_i = 0$, the corresponding $P_i$ can be* arbitrary *and still pairwise distinct; the product is unaffected. The interesting case is that at least two $\beta_i > 0$, which is assumed above; otherwise the constraint $P = \Pi P_i^{\beta_i}/Z$ would force $P = P_k$ for the unique $k$ with $\beta_k = 1$.*

We also note that if $P(o^\star) = 0$ for some $o^\star \in \mathcal{O}$ and all $\beta_i > 0$, then no factorization with strictly positive $P_i$ is possible, because

$$\prod_{i=1}^{n} P_i(o^\star)^{\beta_i} > 0 \quad \Longrightarrow \quad P(o^\star) > 0,$$

a contradiction. A necessary condition is therefore

$$\operatorname{supp}(P) = \bigcap_{\beta_i > 0} \operatorname{supp}(P_i).$$

Under this support-matching condition, one obtains an existence result analogous to Theorem 39 by working on the reduced outcome space $\operatorname{supp}(P)$. That is, we may choose pairwise distinct $Q_i$ supported on $\operatorname{supp}(P)$, define $Q_\star$ and $P_1$ as in the proof (which will also be supported on $\operatorname{supp}(P)$), and set $P_i = Q_i$ for $i \geq 2$. Points outside $\operatorname{supp}(P)$ must receive zero from at least one $P_i$ with $\beta_i > 0$ so that the intersection of supports equals $\operatorname{supp}(P)$.

This construction holds for any non-trivial decomposition. For instance, suppose that we simply wish to decompose an agent into two distinct subagents. Given $P$ strictly positive and any $\beta_1, \beta_2 > 0$ with $\beta_1 + \beta_2 = 1$, pick any $Q_2 \neq P$ with full support and define

$$P_1(o) \;=\; \frac{P(o)^{1/\beta_1} \, Q_2(o)^{-\beta_2/\beta_1}}{\sum_u P(u)^{1/\beta_1} \, Q_2(u)^{-\beta_2/\beta_1}}.$$

Then $P = \frac{1}{Z} P_1^{\beta_1} Q_2^{\beta_2}$, and one can choose $Q_2$ so that $P_1 \neq Q_2$ and $P_1 \neq P$. For $n > 2$, pick any pairwise-distinct $Q_2, \ldots, Q_n$ (on the appropriate support) and absorb their combined effect via $P_1$ as in Theorem 39; this yields a pairwise-distinct factorization for the original $(\beta_i)_{i=1}^n$.

Except for the trivial obstacle of mismatched supports, *every* strictly positive discrete distribution admits a log–opinion-pool factorization with any prescribed nonnegative weights summing to one. Moreover, one can ensure that all components are pairwise distinct and each differs from $P$. When $P$ has zeros, existence is equivalent to the intersection-of-supports condition $\mathrm{supp}(P) = \bigcap_{\beta_i > 0} \mathrm{supp}(P_i)$; under this condition a factorization again exists (with zeros placed accordingly).

### F.1. Defining Subagents a Priori

In this subsection, we examine whether our framework supports another form of *recursive modeling*. An agent may naturally be viewed as the aggregation of multiple underlying subagents, each corresponding to a particular desire or objective–for example, the pursuit of nourishment, rest, social connection, or intellectual achievement. Once a set of $m$ such subagents has been specified, we ask: is it still possible to further decompose the agent into $n$ subagents, where the original $m$ appear as fixed components within the new decomposition? A *scale-free* framework should allow such recursion, preserving the specified subagents while introducing additional ones.

Theorem 41 confirms that this property holds. For any fixed $m$, there exist nonnegative weights $\beta_i \geq 0$ and additional subagents $P_{m+1}, \ldots, P_n$ such that the log-pool of all $n$ subagents recovers the original agent $P(o)$. Moreover, this construction can be made nontrivial: the weights $\beta_i$ for $i \in \{1, \ldots, m\}$ may be taken to be strictly positive, ensuring that the predetermined subagents meaningfully contribute to the aggregate agent.

**Theorem 41** (Log–pool with some components fixed). *Let $\mathcal{O}$ be finite. Let $P$ be a distribution on $\mathcal{O}$ and let $P_1, \ldots, P_m$ be given distributions on $\mathcal{O}$. Fix any integer $n \geq m + 2$.*

(A) ***Strictly positive case.*** *If $P(o) > 0$ for all $o \in \mathcal{O}$ and each $P_i$ $(1 \leq i \leq m)$ is strictly positive, then for any set of weights $\beta_1, \ldots, \beta_n \geq 0$ with $\sum_{i=1}^n \beta_i = 1$, $\beta_{m+1} > 0$, and $\beta_i > 0$ for at least one $i \in \{1, \ldots, m\}$, we have that there exist distributions $P_{m+1}, \ldots, P_n$ on $\mathcal{O}$ such that*

$$P(o) \;=\; \frac{1}{Z} \prod_{i=1}^n P_i(o)^{\beta_i} \qquad (o \in \mathcal{O}),$$

*with $Z = \sum_{u \in \mathcal{O}} \prod_{i=1}^n P_i(u)^{\beta_i}$. Moreover, the construction can be arranged so that $P \neq P_i$ for all $i$, and the $P_i$ are pairwise distinct.*

(B) ***Non-negative case.*** *Suppose we require $\beta_i > 0$ for all $1 \leq i \leq m$. Then such a factorization exists if and only if*

$$\mathrm{supp}(P) \;\subseteq\; \bigcap_{i=1}^m \mathrm{supp}(P_i). \tag{4}$$

*When (4) holds, one can choose $P_{m+1}, \ldots, P_n$ supported exactly on $\mathrm{supp}(P)$ to ensure*

$$\mathrm{supp}(P) \;=\; \bigcap_{i:\, \beta_i > 0} \mathrm{supp}(P_i),$$

*and then build the factorization as in part (A). If (4) fails, no such factorization is possible with all $\beta_1, \ldots, \beta_m > 0$.*

*Proof. (A) Strictly positive case.* Pick any weights $\beta_1, \ldots, \beta_n \geq 0$ with $\sum_{i=1}^n \beta_i = 1$ such that $\beta_{m+1} > 0$ and at least one of $\beta_1, \ldots, \beta_m$ is positive (to keep the fixed components nontrivial). Choose *arbitrary* strictly positive, pairwise distinct

distributions $Q_{m+2}, \ldots, Q_n$ on $\mathcal{O}$, all different from $P$ (e.g., small exponential tilts of a positive reference distribution with distinct tilt directions as in the proof of Theorem 39).

Let $S := \{1, \ldots, n\} \setminus \{m+1\}$ and define the weighted log–mean

$$Q_\star(o) := \exp\Big(\frac{1}{\sum_{i \in S} \beta_i} \sum_{i \in S} \beta_i \log \widehat{P}_i(o)\Big), \qquad \widehat{P}_i := \begin{cases} P_i, & i \leq m, \\ Q_i, & i \geq m+2. \end{cases}$$

Note that $\sum_{i \in S} \beta_i = 1 - \beta_{m+1} > 0$, so $Q_\star$ is well defined and strictly positive. Now set

$$P_{m+1}(o) = \frac{P(o)^{1/\beta_{m+1}} Q_\star(o)^{-\frac{1}{\beta_{m+1}} \sum_{i \in S} \beta_i}}{\sum_{u \in \mathcal{O}} P(u)^{1/\beta_{m+1}} Q_\star(u)^{-\frac{1}{\beta_{m+1}} \sum_{i \in S} \beta_i}} = \frac{P(o)^{1/\beta_{m+1}} Q_\star(o)^{-\frac{1-\beta_{m+1}}{\beta_{m+1}}}}{Z_1},$$

where $Z_1$ is the (strictly positive) normalizing constant in the denominator. For any $o \in \mathcal{O}$,

$$\beta_{m+1} \log P_{m+1}(o) = \beta_{m+1}\Big[\frac{1}{\beta_{m+1}} \log P(o) - \frac{1-\beta_{m+1}}{\beta_{m+1}} \log Q_\star(o) - \log Z_1\Big]$$
$$= \log P(o) - (1 - \beta_{m+1}) \log Q_\star(o) - \beta_{m+1} \log Z_1.$$

Therefore,

$$\sum_{i=1}^n \beta_i \log P_i(o) = \beta_{m+1} \log P_{m+1}(o) + \sum_{i \in S} \beta_i \log \widehat{P}_i(o)$$
$$= \Big[\log P(o) - (1 - \beta_{m+1}) \log Q_\star(o) - \beta_{m+1} \log Z_1\Big] + \sum_{i \in S} \beta_i \log \widehat{P}_i(o).$$

By the definition of $Q_\star$ we have

$$(1 - \beta_{m+1}) \log Q_\star(o) = \sum_{i \in S} \beta_i \log \widehat{P}_i(o),$$

so these terms cancel. Hence

$$\sum_{i=1}^n \beta_i \log P_i(o) = \log P(o) - \beta_{m+1} \log Z_1 = \log P(o) + C,$$

where $C := -\beta_{m+1} \log Z_1$ is independent of $o$. Exponentiating gives

$$\prod_{i=1}^n P_i(o)^{\beta_i} = e^C P(o),$$

and summing over $o$ yields $Z = \sum_o \prod_i P_i(o)^{\beta_i} = e^C$. Therefore

$$P(o) = \frac{1}{Z} \prod_{i=1}^n P_i(o)^{\beta_i} \qquad (o \in \mathcal{O}),$$

as claimed.

*(B) Zeros and supports.* Assume $\beta_i > 0$ for $1 \leq i \leq m$. If there exists $o \in \mathrm{supp}(P)$ with $P_i(o) = 0$ for some fixed $i$, then

$$\prod_{k=1}^n P_k(o)^{\beta_k} = 0 \quad \Longrightarrow \quad P(o) = 0,$$

a contradiction. Thus (4) is necessary.

Conversely, if (4) holds, choose any strictly positive reference distribution $R$ on $\mathrm{supp}(P)$ (e.g., uniform on $\mathrm{supp}(P)$), and define $Q_{m+2}, \ldots, Q_n$ to be pairwise distinct distributions supported *exactly* on $\mathrm{supp}(P)$. The above construction,

performed on the reduced space $\mathrm{supp}(P)$, produces $P_{m+1}$ supported exactly on $\mathrm{supp}(P)$ and yields the desired identity. Because every component with $\beta_i > 0$ is supported on $\mathrm{supp}(P)$ and at least one component ($P_{m+1}$, say) assigns zero outside this set, we obtain

$$\mathrm{supp}(P) = \bigcap_{i:\,\beta_i > 0} \mathrm{supp}(P_i).$$

Therefore (4) is also *sufficient* under the requirement $\beta_1, \ldots, \beta_m > 0$. □

**Remark 42** (Using only a subset of the fixed components). *If it is* not *required that every fixed $P_i$ carries positive weight, a factorization exists as soon as there is a nonempty subset $J \subseteq \{1, \ldots, m\}$ for which*

$$\mathrm{supp}(P) \subseteq \bigcap_{i \in J} \mathrm{supp}(P_i).$$

*Set $\beta_i = 0$ for $i \notin J$, pick positive weights on $J$ (summing to $< 1$), and apply Theorem 41 (A)/(B) on the reduced family.*

**Remark 43** (Nontriviality guarantees in practice). *To avoid trivial solutions (such as $(i)$ some $\beta_i = 1$, $(ii)$ $P = P_i$, or $(iii)$ duplicate components), choose all positive $\beta_i \in (0, 1)$, or ensure at least two fixed indices have $\beta_i > 0$, and, when $n \geq m + 2$, pick $P_{m+2}, \ldots, P_n$ as distinct small exponential tilts of a reference distribution. Genericity of the construction then implies $P \neq P_i$ and $P_i \neq P_j$ for $i \neq j$.*

## G. Recursive Splitting of Agents under Logarithmic Pooling

### G.1. Distributional Invariance under Splitting

We first establish a basic *recursion consistency* property: the pooled distribution should remain unchanged when an agent is replaced by a collection of subagents whose aggregate reproduces the original agent's belief distribution. Formally, if agent $P_i$ is decomposed into $m$ subagents with nonnegative weights $\beta_{i,1}, \ldots, \beta_{i,m}$ satisfying $\sum_{j=1}^{m} \beta_{i,j} = \beta_i$, then the subagents must themselves be log-pooled using normalized weights

$$\alpha_j := \frac{\beta_{i,j}}{\beta_i},$$

so that their aggregation yields $P_i$. The overall pooling is still performed over agents with weights summing to one. This property is formalized in Lemma 44.

**Lemma 44** (Pooling invariance under compatible splitting). *Let $P_1, \ldots, P_n$ be agents with nonnegative pooling weights $\beta_1, \ldots, \beta_n$ satisfying $\sum_{i=1}^{n} \beta_i = 1$. Suppose $P_1$ is replaced by $m$ subagents $P_{1,1}, \ldots, P_{1,m}$ with nonnegative weights $\beta_{1,1}, \ldots, \beta_{1,m}$ such that*

$$\sum_{j=1}^{m} \beta_{1,j} = \beta_1,$$

*and whose aggregation satisfies*

$$P_1 \;\propto\; \prod_{j=1}^{m} P_{1,j}^{\alpha_j}, \qquad where \qquad \alpha_j := \frac{\beta_{1,j}}{\beta_1}. \tag{5}$$

*Let $P$ denote the log pool of the original $n$ agents, and $P'$ the log pool after replacing $P_1$ by its $m$ subagents. Then $P' = P$.*

*Proof.* By definition,

$$P' \;\propto\; \Big( \prod_{j=1}^{m} P_{1,j}^{\beta_{1,j}} \Big) \prod_{i=2}^{n} P_i^{\beta_i} \;\propto\; \Big( \prod_{j=1}^{m} P_{1,j}^{\alpha_j} \Big)^{\beta_1} \prod_{i=2}^{n} P_i^{\beta_i} \;\propto\; P_1^{\beta_1} \prod_{i=2}^{n} P_i^{\beta_i} \;\propto\; P.$$

□

## G.2. Does a Compositional Parent Agent Yield Compositional Subagents?

We begin by proving a useful lemma.

**Lemma 45** (Binary coarse–graining lower bound). *Let $P, Q$ be distributions on a finite alphabet $\mathcal{O}$, and let $A \subseteq \mathcal{O}$ be any event. Then*

$$\mathrm{KL}(P\|Q) \;\geq\; P(A) \log \frac{P(A)}{Q(A)} \;+\; P(A^c) \log \frac{P(A^c)}{Q(A^c)}.$$

*Proof.* Write $p_o := P(o)$ and $q_o := Q(o)$, and adopt the standard conventions $0 \log \frac{0}{y} := 0$ for $y > 0$, and $x \log \frac{x}{0} := +\infty$ for $x > 0$. Let $\{a_i\}_{i \in I}$ and $\{b_i\}_{i \in I}$ be nonnegative sequences with $\sum_i b_i > 0$. We claim the *log–sum inequality*:

$$\sum_{i \in I} a_i \log \frac{a_i}{b_i} \;\geq\; \Big(\sum_{i \in I} a_i\Big) \log \frac{\sum_{i \in I} a_i}{\sum_{i \in I} b_i}, \tag{6}$$

with equality iff $a_i/b_i$ is constant over all indices with $b_i > 0$. If there is an index $j$ with $b_j = 0$ and $a_j > 0$, then the left-hand side is $+\infty$ (by convention) and (6) is trivially true. If $b_j = a_j = 0$, the $j$-th term is $0$ and we may remove $j$ from $I$. Hence it suffices to assume $b_i > 0$ for all $i \in I$.

Let $B := \sum_i b_i > 0$, define weights $\lambda_i := b_i/B$ (so $\lambda_i \geq 0$ and $\sum_i \lambda_i = 1$), and set

$$x_i := \frac{a_i}{b_i} \in [0, \infty).$$

Consider $f : (0, \infty) \to \mathbb{R}$ given by $f(x) = x \log x$. Then $f'(x) = \log x + 1$ and $f''(x) = 1/x > 0$, so $f$ is convex on $(0, \infty)$. By Jensen's inequality,

$$\sum_i \lambda_i f(x_i) \;\geq\; f\Big(\sum_i \lambda_i x_i\Big).$$

Multiply both sides by $B$ (noting $B\lambda_i = b_i$) to get

$$\sum_i b_i \, f\Big(\frac{a_i}{b_i}\Big) \;\geq\; B \, f\Big(\frac{\sum_i b_i(a_i/b_i)}{B}\Big) \;=\; B \, f\Big(\frac{\sum_i a_i}{B}\Big).$$

Since $b_i f(a_i/b_i) = a_i \log(a_i/b_i)$ and $B f((\sum a_i)/B) = (\sum a_i) \log((\sum a_i)/B)$, we obtain (6). Equality in Jensen holds iff $x_i$ is constant on the support of $\{\lambda_i\}$, i.e., if and only if $a_i/b_i$ is constant for all $i$ with $b_i > 0$.

Now, apply (6) to the block $A$ with $a_o = p_o$, $b_o = q_o$ for $o \in A$:

$$\sum_{o \in A} p_o \log \frac{p_o}{q_o} \;\geq\; \Big(\sum_{o \in A} p_o\Big) \log \frac{\sum_{o \in A} p_o}{\sum_{o \in A} q_o} \;=\; P(A) \log \frac{P(A)}{Q(A)}.$$

Apply it again to the complementary block $A^c$:

$$\sum_{o \notin A} p_o \log \frac{p_o}{q_o} \;\geq\; P(A^c) \log \frac{P(A^c)}{Q(A^c)}.$$

Adding the two inequalities and recalling that $\mathrm{KL}(P\|Q) = \sum_{o \in \mathcal{O}} p_o \log \frac{p_o}{q_o}$, we obtain

$$\mathrm{KL}(P\|Q) \;\geq\; P(A) \log \frac{P(A)}{Q(A)} \;+\; P(A^c) \log \frac{P(A^c)}{Q(A^c)},$$

which is the desired binary coarse–graining lower bound. $\qquad\square$

If the parent is a compositional agent, does this property pass onto the child subagent decompositions? The next result answers this in the negative, even if the parent strictly benefits.

**Theorem 46** (Parental benefit need not pass to subagents). *There exist agents $P_1, \ldots, P_n$, weights $\beta$, and a compatible split of $P_1$ into $P_{1,1}, P_{1,2}$ as in (5) such that the composition $P$ (before/after splitting) satisfies $\Delta_{P_1}(P) > 0$ but $\Delta_{P_{1,1}}(P) < 0$. By symmetry, one can also have $\Delta_{P_{1,2}}(P) < 0$.*

*Proof.* Let $P_1$ be any non-uniform strictly positive distribution on $\mathcal{O}$. For $t > 1$, define the $t$-tilt of $P_1$ by

$$P_t(o) \; := \; \frac{P_1(o)^t}{Z(t)}, \qquad Z(t) := \sum_{x \in \mathcal{O}} P_1(x)^t.$$

Then $P_t$ is the log-pool of $\{P_1, P_2\}$ with weights $\beta_1 = \beta_2 = \frac{1}{2}$ and $P_2 \propto P_1^{2t-1}$, since $P \propto P_1^{1/2} P_2^{1/2} \propto P_1^{(1+(2t-1))/2} = P_1^t$, which normalizes to $P_t$.

Now, we show that the child agent $P_1$ strictly benefits against the parent agent $P_t$ for all $t > 1$. By definition, we have

$$\Delta_{P_1}(P_t) \; = \; \mathbb{E}_{P_t}[\log P_1] - \mathbb{E}_{P_1}[\log P_1] \; = \; \mathbb{E}_{P_t}[\log P_1] + H(P_1).$$

We claim $\mathbb{E}_{P_t}[\log P_1]$ is strictly increasing in $t$ whenever $P_1$ is non-uniform, and therefore $\mathbb{E}_{P_t}[\log P_1] > \mathbb{E}_{P_1}[\log P_1]$ for every $t > 1$. To see this, write $p(o) := P_1(o)$ and $f(o) := \log p(o)$. Then we have

$$\mathbb{E}_{P_t}[f] \; = \; \sum_o \frac{p(o)^t}{Z(t)} f(o) \; = \; \frac{Z'(t)}{Z(t)} \; = \; \frac{\mathrm{d}}{\mathrm{d}t} \log Z(t),$$

because $Z'(t) = \sum_o p(o)^t \log p(o)$. Differentiating once more,

$$\frac{\mathrm{d}^2}{\mathrm{d}t^2} \log Z(t) = \frac{Z''(t)}{Z(t)} - \left(\frac{Z'(t)}{Z(t)}\right)^2 = \sum_o \frac{p(o)^t}{Z(t)} \left(\log p(o)\right)^2 - \left(\sum_o \frac{p(o)^t}{Z(t)} \log p(o)\right)^2$$

$$= \mathrm{Var}_{P_t}\left(\log P_1\right) \geq 0.$$

Thus $t \mapsto \mathbb{E}_{P_t}[\log P_1] = \frac{\mathrm{d}}{\mathrm{d}t} \log Z(t)$ is *increasing* in $t$, and it is *strictly* increasing whenever $\mathrm{Var}_{P_t}(\log P_1) > 0$, i.e., whenever $\log P_1$ is not almost surely constant under $P_t$. The latter holds exactly when $P_1$ is non-uniform. Evaluating at $t = 1$ gives $\mathbb{E}_{P_1}[\log P_1] = \sum_o P_1(o) \log P_1(o)$, so for any $t > 1$,

$$\mathbb{E}_{P_t}[\log P_1] \; > \; \mathbb{E}_{P_1}[\log P_1] \quad \Longrightarrow \quad \Delta_{P_1}(P_t) \; > \; 0.$$

Now, we derive a compatible split of the child agent $P_1$ that makes its subagent strictly lose epistemic utility. Fix $\alpha \in (0, 1)$ and any function $g : \mathcal{O} \to \mathbb{R}$. Define subagents by exponential tilting:

$$\log P_{1,1} \; = \; \log P_1 + (1 - \alpha)g - c_1, \qquad \log P_{1,2} \; = \; \log P_1 - \alpha g - c_2, \tag{7}$$

where $c_1, c_2$ make each distribution sum to 1. Then $\alpha \log P_{1,1} + (1 - \alpha) \log P_{1,2} = \log P_1 - \left(\alpha c_1 + (1 - \alpha)c_2\right)$, hence (5) holds with $\beta_{1,1} = \alpha \beta_1$ and $\beta_{1,2} = (1 - \alpha)\beta_1$. By Lemma 44, the composition *after* splitting remains $P_t$.

Choose $g$ to depress a single outcome: pick any $o^\star \in \mathcal{O}$ and set $g_\lambda(o) := -\lambda \mathbf{1}_{\{o = o^\star\}}$ with $\lambda > 0$. From (7),

$$P_{1,1}^{(\lambda)}(o) \; \propto \; P_1(o) \exp\left((1 - \alpha)g_\lambda(o)\right) = \begin{cases} P_1(o^\star) e^{-(1-\alpha)\lambda}, & o = o^\star, \\ P_1(o), & o \neq o^\star, \end{cases}$$

so, after normalization,

$$P_{1,1}^{(\lambda)}(o^\star) = \frac{P_1(o^\star)e^{-(1-\alpha)\lambda}}{P_1(o^\star)e^{-(1-\alpha)\lambda} + \sum_{o \neq o^\star} P_1(o)} \xrightarrow{\lambda \to \infty} 0,$$

while $P_{1,1}^{(\lambda)}(o) > 0$ for $o \neq o^\star$. We have by Lemma 45 that for $A = \{o^\star\}$,

$$\mathrm{KL}\left(P_t \,\|\, P_{1,1}^{(\lambda)}\right) \; \geq \; P_t(A) \log \frac{P_t(A)}{P_{1,1}^{(\lambda)}(A)} \; + \; P_t(A^c) \log \frac{P_t(A^c)}{P_{1,1}^{(\lambda)}(A^c)}.$$

Since $P_t(o^\star) = P_t(A) > 0$ and $P_{1,1}^{(\lambda)}(o^\star) = P_{1,1}^{(\lambda)}(A) \to 0$ while $P_{1,1}^{(\lambda)}(A^c) \to 1$, it follows that

$$\mathrm{KL}\left(P_t \,\|\, P_{1,1}^{(\lambda)}\right) \; \gtrsim \; P_t(o^\star) \log \frac{P_t(o^\star)}{P_{1,1}^{(\lambda)}(o^\star)} \; + \; \left(1 - P_t(o^\star)\right) \log\left(1 - P_t(o^\star)\right) \xrightarrow{\lambda \to \infty} +\infty.$$

On the other hand, for every $\lambda$,

$$0 \;\leq\; H\big(P_{1,1}^{(\lambda)}\big) \;\leq\; \log|\mathcal{O}|,$$

since entropy on a finite alphabet is always between 0 (point mass) and $\log|\mathcal{O}|$ (uniform). Thus $H\big(P_{1,1}^{(\lambda)}\big)$ is *uniformly bounded in* $\lambda$. Using Proposition 32,

$$\Delta_{P_{1,1}^{(\lambda)}}(P_t) = H\big(P_{1,1}^{(\lambda)}\big) - H(P_t) - \mathrm{KL}\big(P_t\big\|P_{1,1}^{(\lambda)}\big) \;\xrightarrow[\lambda\to\infty]{}\; -\infty,$$

because $H(P_t)$ is *independent of* $\lambda$ (as $t$ is fixed) and finite, the entropy term is uniformly bounded, while the KL term diverges to $+\infty$. Consequently, for all sufficiently large $\lambda$, $\Delta_{P_{1,1}^{(\lambda)}}(P_t) < 0$ even though $\Delta_{P_1}(P_t) > 0$. $\qquad\square$

This immediately results in the following corollary, which shows that an initially unanimous group can lose unanimity after splitting even though the composition distribution does not change.

**Corollary 47** (Unanimity is not preserved under splitting). *Suppose $\{P_1, \ldots, P_n\}$ with weights $\beta$ form a unanimously compositional group under the log pool $P$; i.e., $\Delta_{P_i}(P) > 0$ for all $i$. There exists a compatible split of $P_i$ into two subagents $P_{i,1}, P_{i,2}$ such that, with the new set of agents and weights, the pooled distribution remains $P$ but $\Delta_{P_{i,1}}(P) < 0$.*

*Proof.* Define the split via (7) with a tilt $g_\lambda$ that assigns a large negative value on some outcome $o^\star$ with $P(o^\star) > 0$ and 0 elsewhere. As in the proof of Proposition 46, Lemma 44 ensures the pool stays $P$, every unchanged agent $k \neq i$ maintains the same (strictly positive) $\Delta_{P_k}(P)$, while $\Delta_{P_{i,1}}(P) \to -\infty$ as $\lambda \to \infty$. $\qquad\square$

### G.2.1. WHEN DOES SPLITTING PRESERVE COMPOSITIONALITY?

The negative results above are sharp in the sense that they rely on sufficiently *polarized* splits. Two simple stability statements go the other way.

**Lemma 48** (Cloning preserves welfare and non-strict unanimity). *If $P_{1,1} = \cdots = P_{1,m} = P_1$ and $\sum_j \beta_{1,j} = \beta_1$, then for each clone $P_{1,j}$ we have $\Delta_{P_{1,j}}(P) = \Delta_{P_1}(P)$. In particular, if the original group is (non-strictly) unanimously compositional, it remains so after cloning any subset of agents.*

*Proof.* Pooling invariance gives $P' = P$. For any clone $R = P_{1,j} = P_1$ we have $\mathbb{E}_P[\log R] = \mathbb{E}_P[\log P_1]$ and $\mathbb{E}_R[\log R] = \mathbb{E}_{P_1}[\log P_1]$, hence $\Delta_R(P) = \Delta_{P_1}(P)$. $\qquad\square$

**Lemma 49** (Small-perturbation stability). *Fix a composition pool $P$ with all coordinates bounded below by $\underline{p} > 0$. Then for a distribution $R$, the map $R \mapsto \Delta_R(P)$ is continuous on the interior of the simplex. Hence, for any agent $R_\star$ with $\Delta_{R_\star}(P) > 0$, there exists $\varepsilon > 0$ such that if a subagent $R$ satisfies*

$$\|\log R - \log R_\star\|_\infty < \varepsilon,$$

*then $\Delta_R(P) > 0$. Consequently, if $P_1$ benefits and each subagent $P_{1,j}$ is a sufficiently small log-perturbation of $P_1$, the split preserves positivity of each $\Delta_{P_{1,j}}(P)$.*

*Proof.* Using Proposition 32, $\Delta_R(P) = H(R) - \mathrm{KL}(P\|R) - H(P)$ and $H(P)$ is constant. On the interior of the simplex, $H(\cdot)$ is continuous. Moreover

$$\mathrm{KL}(P\|R) \;=\; \sum_{o\in\mathcal{O}} P(o)\big(-\log R(o)\big) - H(P),$$

so if $\|\log R - \log R_\star\|_\infty < \varepsilon$ then

$$\big|\,\mathrm{KL}(P\|R) - \mathrm{KL}(P\|R_\star)\big| \leq \sum_o P(o)\,\big|\log R(o) - \log R_\star(o)\big| \leq \varepsilon.$$

Thus $R \mapsto \Delta_R(P)$ is continuous. Pick $\varepsilon$ small enough that $|\Delta_R(P) - \Delta_{R_\star}(P)| < \tfrac{1}{2}\Delta_{R_\star}(P)$ whenever $\|\log R - \log R_\star\|_\infty < \varepsilon$, which implies $\Delta_R(P) > \tfrac{1}{2}\Delta_{R_\star}(P) > 0$. $\qquad\square$

**Remark 50.** *If some distributions have zeros, all statements above hold verbatim after restricting every distribution to the common support* $\operatorname{supp}(P) \cap \bigcap_i \operatorname{supp}(P_i)$ *(or, for Theorem 47, any support where $P$ is positive). Lemma 49 then applies as long as we stay in the interior of that reduced simplex.*

**Remark 51.** *Lemma 44 shows that log pooling is distributionally recursion-invariant under compatible splits. However, Proposition 46 and Theorem 47 demonstrate that welfare and unanimity are not recursion-invariant in general: a strictly beneficial parent can have strictly harmed subagents, and an initially unanimous group can lose unanimity after splitting, even though the pooled distribution is unchanged. On the positive side, Lemma 48 (cloning) and Lemma 49 (continuity) give simple sufficient conditions under which (possibly non-strict) compositionality is preserved.*

## H. Properties of Strict Local (Small-Tilt) Unanimity

In the previous section, we showed that duplication preserves non-strict unanimous compositionality. One might attempt to *cheat* strict compositionality by taking an arbitrary distribution $P$, duplicating each agent, and introducing only slight perturbations to their beliefs, hoping that the resulting group would be unanimously compositional. We now show that this strategy is doomed to fail: such near-duplication decompositions cannot achieve unanimous strict improvement for a fixed $P$.

In this section, we show that even when $|\mathcal{O}| \geq 3$, if the pooled distribution is fixed to be a given $P$ and we factor it via (8) with small tilts $h_i$, then unanimous strict improvement cannot occur. This rules out the possibility of obtaining unanimity through "uniformly small" decompositions when $P$ is fixed. We start by proving several lemmas.

**Lemma 52.** *Whenever $P \propto \prod_i P_i^{\beta_i}$, we can write each component as a* tilt *of $P$:*

$$P_i(o) \;=\; \frac{P(o)\, e^{h_i(o)}}{Z_i}, \qquad Z_i := \sum_{x \in \mathcal{O}} P(x) e^{h_i(x)} \;(= \mathbb{E}_P[e^{h_i}]), \tag{8}$$

*for some functions $h_i : \mathcal{O} \to \mathbb{R}$. We may without loss of generality impose*

$$\sum_{i=1}^{n} \beta_i h_i(o) \;\equiv\; 0 \quad \text{for all } o \in \mathcal{O}. \tag{9}$$

*Proof.* Assume $\sum_{i=1}^{n} \beta_i = 1$ and all distributions are strictly positive on the finite outcome set $\mathcal{O}$. Suppose the composition $P$ is the logarithmic pool of $P_1, \ldots, P_n$ with weights $\beta$:

$$P(o) \;=\; \frac{1}{Z} \prod_{i=1}^{n} P_i(o)^{\beta_i}, \qquad Z := \sum_{x \in \mathcal{O}} \prod_{i=1}^{n} P_i(x)^{\beta_i}. \tag{10}$$

Define, for each $i$,

$$h_i^{(0)}(o) \;:=\; \log \frac{P_i(o)}{P(o)} \quad \text{for all } o \in \mathcal{O}. \tag{11}$$

Then

$$P(o)\, e^{h_i^{(0)}(o)} \;=\; P(o)\, \frac{P_i(o)}{P(o)} \;=\; P_i(o).$$

Hence, with $Z_i^{(0)} := \sum_x P(x) e^{h_i^{(0)}(x)} = \sum_x P_i(x) = 1$, we have the exact "tilt" identity

$$P_i(o) \;=\; \frac{P(o)\, e^{h_i^{(0)}(o)}}{Z_i^{(0)}} \;=\; P(o)\, e^{h_i^{(0)}(o)}. \tag{12}$$

This proves (8) with the specific choice $h_i = h_i^{(0)}$ and $Z_i = 1$. Taking logs in (10),

$$\log P(o) \;=\; \sum_{i=1}^{n} \beta_i \log P_i(o) \;-\; \log Z.$$

Therefore,

$$\sum_{i=1}^{n} \beta_i \, h_i^{(0)}(o) = \sum_{i=1}^{n} \beta_i \big( \log P_i(o) - \log P(o) \big)$$

$$= \Big( \sum_{i=1}^{n} \beta_i \log P_i(o) \Big) \;-\; \log P(o) \qquad (\text{since } \sum_i \beta_i = 1)$$

$$= \log Z \quad (\text{independent of } o).$$

The tilt representation (12) is invariant under adding an additive constant to $h_i$ and scaling $Z_i$ accordingly: if we set

$$h_i(o) \; := \; h_i^{(0)}(o) - c_i, \qquad Z_i \; := \; e^{-c_i} Z_i^{(0)} = e^{-c_i},$$

then

$$\frac{P(o) \, e^{h_i(o)}}{Z_i} = \frac{P(o) \, e^{h_i^{(0)}(o) - c_i}}{e^{-c_i} Z_i^{(0)}} = \frac{P(o) \, e^{h_i^{(0)}(o)}}{Z_i^{(0)}} = P_i(o),$$

so the distributions $P_i$ do not change. As $\sum_i \beta_i h_i^{(0)} \equiv \log Z$, choose any index $k$ and set $c_k := \frac{\log Z}{\beta_k}$, while $c_i := 0$ for $i \neq k$. Then the modified functions $h_i$ satisfy

$$\sum_{i=1}^{n} \beta_i h_i(o) = \sum_{i=1}^{n} \beta_i h_i^{(0)}(o) \;-\; \beta_k c_k = \log Z \;-\; \log Z \;=\; 0 \quad \text{for all } o.$$

Thus, without loss of generality, we may assume

$$\sum_{i=1}^{n} \beta_i h_i(o) \; \equiv \; 0, \qquad \forall o \in \mathcal{O}. \tag{13}$$

We note that imposing (13) is achieved by subtracting a constant from a single $h_k$, which leaves each $P_i$ exactly unchanged due to rescaling $Z_k$ by a constant factor. $\qquad \square$

**Remark 53.** *We note that the freedom to add constants $c_i$ to $h_i$ is the* only *freedom in the tilt representation relative to a fixed base $P$; it corresponds to renormalizing $Z_i$ and does not alter $P_i$.*

**Lemma 54.** *For a fixed $i$ and $\varepsilon$ near 0, let*

$$P_i^{(\varepsilon)}(o) \; = \; \frac{P(o) \, e^{\varepsilon h_i(o)}}{Z_i(\varepsilon)}, \qquad Z_i(\varepsilon) \; := \; \sum_{x \in \mathcal{O}} P(x) \, e^{\varepsilon h_i(x)} \; = \; \mathbb{E}_P\big[ e^{\varepsilon h_i} \big].$$

*Then, we have that*

$$\frac{d}{d\varepsilon} H\big( P_i^{(\varepsilon)} \big) \Big|_{\varepsilon = 0} = -\Big( \mathbb{E}_P[h_i \log P] - \mathbb{E}_P[h_i] \, \mathbb{E}_P[\log P] \Big) = -\operatorname{Cov}_P\big( h_i, \log P \big).$$

*We assume all probabilities are strictly positive.*

*Proof.* Let $q_\varepsilon(o) = P_i^{(\varepsilon)}(o)$. By definition,

$$H(q_\varepsilon) \; = \; -\sum_{o \in \mathcal{O}} q_\varepsilon(o) \, \log q_\varepsilon(o).$$

Differentiating termwise,

$$\frac{d}{d\varepsilon} H(q_\varepsilon) = -\sum_o \Big( \frac{d}{d\varepsilon} q_\varepsilon(o) \Big) \log q_\varepsilon(o) - \sum_o q_\varepsilon(o) \frac{d}{d\varepsilon} \log q_\varepsilon(o).$$

Since $\frac{d}{d\varepsilon}\log q_\varepsilon(o) = \frac{1}{q_\varepsilon(o)}\frac{d}{d\varepsilon}q_\varepsilon(o)$, we can combine the two sums:

$$\frac{d}{d\varepsilon}H(q_\varepsilon) = -\sum_o \left(\frac{d}{d\varepsilon}q_\varepsilon(o)\right)\left(1 + \log q_\varepsilon(o)\right) = -\sum_o q_\varepsilon(o)\frac{d}{d\varepsilon}\log q_\varepsilon(o)\left(1 + \log q_\varepsilon(o)\right). \tag{14}$$

From the definition of $q_\varepsilon$,

$$\log q_\varepsilon(o) = \log P(o) + \varepsilon\, h_i(o) - \log Z_i(\varepsilon).$$

Therefore

$$\frac{d}{d\varepsilon}\log q_\varepsilon(o) = h_i(o) - \frac{d}{d\varepsilon}\log Z_i(\varepsilon). \tag{15}$$

Compute the latter factor using $Z_i'(\varepsilon) = \sum_x P(x)\, h_i(x)\, e^{\varepsilon h_i(x)}$:

$$\frac{d}{d\varepsilon}\log Z_i(\varepsilon) = \frac{Z_i'(\varepsilon)}{Z_i(\varepsilon)} = \frac{\sum_x P(x)\, h_i(x)\, e^{\varepsilon h_i(x)}}{\sum_x P(x)\, e^{\varepsilon h_i(x)}} = \sum_x \frac{P(x)\, e^{\varepsilon h_i(x)}}{Z_i(\varepsilon)}\, h_i(x) = \mathbb{E}_{q_\varepsilon}[\, h_i\,].$$

Thus, evaluating (15) at $\varepsilon = 0$ (where $q_0 = P$),

$$\left[\frac{d}{d\varepsilon}\log q_\varepsilon(o)\right]_{\varepsilon=0} = h_i(o) - \mathbb{E}_{q_0}[h_i] = h_i(o) - \mathbb{E}_P[h_i]. \tag{16}$$

Using (14) and (16),

$$\frac{d}{d\varepsilon}H(q_\varepsilon)\Big|_{\varepsilon=0} = -\sum_o P(o)\left(h_i(o) - \mathbb{E}_P[h_i]\right)\left(1 + \log P(o)\right)$$
$$= -\mathbb{E}_P\left[\left(h_i - \mathbb{E}_P[h_i]\right)\left(1 + \log P\right)\right].$$

Expand the expectation:

$$\mathbb{E}_P\left[\left(h_i - \mathbb{E}_P[h_i]\right)\left(1 + \log P\right)\right] = \mathbb{E}_P[h_i(1 + \log P)] - \mathbb{E}_P[h_i]\,\mathbb{E}_P[1 + \log P]$$
$$= \mathbb{E}_P[h_i \log P] + \mathbb{E}_P[h_i] - \mathbb{E}_P[h_i]\left(1 + \mathbb{E}_P[\log P]\right)$$
$$= \mathbb{E}_P[h_i \log P] - \mathbb{E}_P[h_i]\,\mathbb{E}_P[\log P].$$

Therefore

$$\frac{d}{d\varepsilon}H\big(P_i^{(\varepsilon)}\big)\Big|_{\varepsilon=0} = -\left(\mathbb{E}_P[h_i \log P] - \mathbb{E}_P[h_i]\,\mathbb{E}_P[\log P]\right) = -\operatorname{Cov}_P\big(h_i, \log P\big).$$

This is the claimed identity. $\qquad\square$

We may now show the following theorem.

**Theorem 55** (Local unanimity impossibility at a fixed pool)**.** *Fix $P$ strictly positive on $\mathcal{O}$. Suppose $P_i = P_i^{(\varepsilon)} = P\, e^{\varepsilon h_i}/Z_i(\varepsilon)$ with $\sum_i \beta_i h_i \equiv 0$ as in (9). If not all covariances $\operatorname{Cov}_P(h_i, \log P)$ are uniformly zero, then for all sufficiently small $\varepsilon > 0$, it is impossible that $\Delta_{P_i}(P) > 0$ holds for every $i$.*

*Proof.* Define $P_i^{(\varepsilon)}$ as above and write $\Delta_i(\varepsilon) := \Delta_{P_i^{(\varepsilon)}}(P)$. We claim

$$\Delta_i'(0) = -\operatorname{Cov}_P(h_i, \log P). \tag{17}$$

To prove (17), note from Proposition 32 that $\Delta_i(\varepsilon) = H(P_i^{(\varepsilon)}) - H(P) - \operatorname{KL}(P\|P_i^{(\varepsilon)})$. Differentiate at $\varepsilon = 0$. First, $\frac{d}{d\varepsilon}\operatorname{KL}(P\|P_i^{(\varepsilon)})\big|_{\varepsilon=0} = \frac{d}{d\varepsilon}\mathbb{E}_P\big[\log\frac{P}{P_i^{(\varepsilon)}}\big]\big|_{\varepsilon=0} = -\mathbb{E}_P\big[\frac{d}{d\varepsilon}\log P_i^{(\varepsilon)}\big]_{\varepsilon=0}$. From $P_i^{(\varepsilon)} \propto P e^{\varepsilon h_i}$ we have $\log P_i^{(\varepsilon)} = \log P + \varepsilon h_i - \log Z_i(\varepsilon)$, so $\frac{d}{d\varepsilon}\log P_i^{(\varepsilon)}\big|_0 = h_i - \mathbb{E}_P[h_i]$. Hence

$$\frac{d}{d\varepsilon}\operatorname{KL}(P\|P_i^{(\varepsilon)})\Big|_0 = -\left(\mathbb{E}_P[h_i] - \mathbb{E}_P[h_i]\right) = 0. \tag{18}$$

Next, $H(P_i^{(\varepsilon)}) = -\mathbb{E}_{P_i^{(\varepsilon)}}[\log P_i^{(\varepsilon)}]$. Differentiating at 0 gives by Lemma 54

$$\frac{d}{d\varepsilon} H(P_i^{(\varepsilon)})\Big|_0 = -\operatorname{Cov}_P(h_i, \log P).$$

Combining with (18) we obtain (17). By (9), $\sum_i \beta_i h_i \equiv 0$, so

$$\sum_{i=1}^n \beta_i \, \Delta_i'(0) = -\operatorname{Cov}_P\Big(\sum_i \beta_i h_i, \, \log P\Big) = 0.$$

Therefore the $\beta$–weighted average of the derivatives $\Delta_i'(0)$ is zero; in particular, they cannot all be strictly positive as $\Delta_i(0) = 0$. Hence, for all sufficiently small $\varepsilon > 0$, it is impossible that $\Delta_i(\varepsilon) > 0$ for every $i$ (otherwise all $\Delta_i'(0) \geq 0$ with some $> 0$ by right-derivative positivity, contradicting the weighted average zero). $\square$

**Remark 56.** *Theorem 55 does* not *rule out the possibility that for some* large *tilts (finite $\varepsilon$), all $\Delta_{P_i}(P)$ might be positive; it only precludes near-uniform (i.e., near-duplicate) small-tilt unanimity around a fixed pool $P$. For instance, contrast with the proof of Theorem 62, which guarantees strict positivity for large tilts under some unrestrictive conditions.*

### H.1. On Joining Random Distributions

The next theorem shows that no distribution $P$ is incentivized, in the compositional sense, to join a maximal entropy or completely random distribution.

**Lemma 57** (No strict gains at the uniform). *Let $U$ be the uniform distribution on a finite $\mathcal{O}$ with $|\mathcal{O}| = m \geq 2$. For any strictly positive $R$,*

$$\Delta_R(U) = H(R) - \log m - \operatorname{KL}(U\|R) \leq 0,$$

*with equality iff $R = U$.*

*Proof.* By definition $\Delta_R(U) = \mathbb{E}_U[\log R] - \mathbb{E}_R[\log R]$. Since $H(U) = \log m$ and $\operatorname{KL}(U\|R) = \mathbb{E}_U[\log U - \log R] = -\log m - \mathbb{E}_U[\log R]$, we have

$$\Delta_R(U) = \big(-\log m - \operatorname{KL}(U\|R)\big) + H(R) = H(R) - \log m - \operatorname{KL}(U\|R) \leq 0,$$

with equality iff $R = U$ (Gibbs' inequality). Alternatively, for the final step, we may compute $\operatorname{KL}(R\|U)$ explicitly:

$$
\begin{aligned}
\operatorname{KL}(R\|U) &= \sum_o R(o) \log \frac{R(o)}{1/m} = \sum_o R(o) \log R(o) + \sum_o R(o) \log m \\
&= \sum_o R(o) \log R(o) + \log m \cdot \sum_o R(o) = \sum_o R(o) \log R(o) + \log m \\
&= -H(R) + \log m.
\end{aligned}
$$

Rearranging gives the identity

$$H(R) - \log m = -\operatorname{KL}(R\|U). \tag{19}$$

Therefore

$$
\begin{aligned}
H(R) - \log m - \operatorname{KL}(U\|R) &= \big(-\operatorname{KL}(R\|U)\big) - \operatorname{KL}(U\|R) \\
&= -\Big(\operatorname{KL}(R\|U) + \operatorname{KL}(U\|R)\Big) \leq 0,
\end{aligned}
$$

since each Kullback–Leibler divergence is nonnegative. Moreover, equality holds iff $\operatorname{KL}(R\|U) = 0$ and $\operatorname{KL}(U\|R) = 0$, which happens iff $R = U$. $\square$

Thus, we have provided an example of a distribution for which a strictly unanimous compositional group may not be formed. Can this result be extended to non-trivial distributions? We now show that there exists a set of non-uniform distributions such that they can in no way form a unanimously compositional group, regardless of which (non-trivial, i.e., bounded away from 0) weights are assigned. In other words, certain agents or beliefs are fundamentally incompatible, and weights cannot in general make an arbitrary family unanimously compositional.

**Theorem 58** (No universal weights for unanimity). *Fix $n \geq 2$. There exist $n$ strictly positive and non-uniform distributions $P_1, \ldots, P_n$ on $\mathcal{O} = \{1, \ldots, n\}$ such that, for* every *choice of positive weights $\beta_i \geq \tau > 0$ for $\tau$ arbitrarily small and $\sum_{i=1}^{n} \beta_i = 1$, the log-pool $P$ fails to make all agents strictly better off; i.e., at least one index $i$ has $\Delta_{P_i}(P) < 0$.*

*Proof.* For a parameter $\varepsilon \in (0, \frac{1}{2})$, define for each $i \in \{1, \ldots, n\}$ the distribution $P_i^\varepsilon$ by

$$P_i^\varepsilon(i) = 1 - \varepsilon, \qquad P_i^\varepsilon(k) = \frac{\varepsilon}{n-1} \quad (k \neq i).$$

These $P_i^\varepsilon$ are strictly positive and "peak" on outcome $i$. For any positive weights $\beta_1, \ldots, \beta_n$ with $\sum_i \beta_i = 1$, the (unnormalized) pooled mass at outcome $k$ is

$$\widetilde{P}^\varepsilon(k) = \prod_{i=1}^{n} \left( P_i^\varepsilon(k) \right)^{\beta_i} = \left( 1 - \varepsilon \right)^{\beta_k} \left( \frac{\varepsilon}{n-1} \right)^{\sum_{i \neq k} \beta_i} = \left( 1 - \varepsilon \right)^{\beta_k} \left( \frac{\varepsilon}{n-1} \right)^{1 - \beta_k}.$$

Let

$$Z^\varepsilon := \sum_{r=1}^{n} \widetilde{P}^\varepsilon(r) = \sum_{r=1}^{n} \left( 1 - \varepsilon \right)^{\beta_r} \left( \frac{\varepsilon}{n-1} \right)^{1 - \beta_r}.$$

Then the pooled distribution is $P^\varepsilon(k) = \widetilde{P}^\varepsilon(k)/Z^\varepsilon$. Define

$$S^\varepsilon(\beta) := \sum_{i=1}^{n} \beta_i \, \Delta_{P_i^\varepsilon}(P^\varepsilon).$$

Using $\Delta_R(P) = \mathbb{E}_P[\log R] - \mathbb{E}_R[\log R]$ and $\log P^\varepsilon = \sum_i \beta_i \log P_i^\varepsilon - \log Z^\varepsilon$, we obtain

$$S^\varepsilon(\beta) = \sum_i \beta_i \, \mathbb{E}_{P^\varepsilon}[\log P_i^\varepsilon] \; - \; \sum_i \beta_i \, \mathbb{E}_{P_i^\varepsilon}[\log P_i^\varepsilon]$$

$$= \mathbb{E}_{P^\varepsilon}\left[ \sum_i \beta_i \log P_i^\varepsilon \right] \; + \; \sum_i \beta_i H\left( P_i^\varepsilon \right) \quad (\text{since } \mathbb{E}_{P_i^\varepsilon}[\log P_i^\varepsilon] = -H(P_i^\varepsilon))$$

$$= \mathbb{E}_{P^\varepsilon}[\log P^\varepsilon + \log Z^\varepsilon] \; + \; \sum_i \beta_i H\left( P_i^\varepsilon \right)$$

$$< -H\left( P^\varepsilon \right) \; + \; \log Z^\varepsilon \; + \; \sum_i H\left( P_i^\varepsilon \right). \tag{20}$$

Therefore, if we show $S^\varepsilon(\beta) < 0$, then at least one $\Delta_{P_i^\varepsilon}(P^\varepsilon)$ is negative (since the $\beta_i$ are positive). Let $\beta_\star := \max_{1 \leq r \leq n} \beta_r$ and $K := \{r : \beta_r = \beta_\star\}$ (nonempty). Then

$$Z^\varepsilon = \sum_{r=1}^{n} \left( 1 - \varepsilon \right)^{\beta_r} \left( \frac{\varepsilon}{n-1} \right)^{1 - \beta_r}$$

$$= \left( \frac{\varepsilon}{n-1} \right)^{1 - \beta_\star} \left( 1 - \varepsilon \right)^{\beta_\star} \sum_{r=1}^{n} \left( \frac{\varepsilon}{n-1} \right)^{\beta_\star - \beta_r} \left( 1 - \varepsilon \right)^{\beta_r - \beta_\star}.$$

For $r \notin K$, $\beta_\star - \beta_r > 0$, hence $\left( \frac{\varepsilon}{n-1} \right)^{\beta_\star - \beta_r} \to 0$ as $\varepsilon \to 0^+$; for $r \in K$, that factor equals 1. Thus

$$\log Z^\varepsilon = (1 - \beta_\star) \log\left( \frac{\varepsilon}{n-1} \right) + \beta_\star \log(1 - \varepsilon) + \log\left( |K| + o(1) \right).$$

In particular, there exists a constant $C_1$ (depending only on $\beta$ and $n$) such that for all sufficiently small $\varepsilon$,

$$\log Z^\varepsilon \leq (1 - \beta_\star) \log \varepsilon + C_1, \qquad \text{with } (1 - \beta_\star) > 0 \text{ since } n \geq 2. \tag{21}$$

Now for each $i$,

$$H\left( P_i^\varepsilon \right) = -(1 - \varepsilon) \log(1 - \varepsilon) - \sum_{k \neq i} \frac{\varepsilon}{n-1} \log\left( \frac{\varepsilon}{n-1} \right)$$

$$= -(1 - \varepsilon) \log(1 - \varepsilon) - \varepsilon \log \varepsilon + \varepsilon \log(n-1). \tag{22}$$

Hence there exists $C_2$ (independent of $i$) with

$$\sum_{i=1}^{n} H\big(P_i^{\varepsilon}\big) = n\Big( -(1-\varepsilon)\log(1-\varepsilon) - \varepsilon\log\varepsilon + \varepsilon\log(n-1)\Big) \ \leq\ C_2\,\varepsilon\big(1 - \log\varepsilon\big) \tag{23}$$

for all sufficiently small $\varepsilon$ (using $-\log(1-\varepsilon) \leq 2\varepsilon$ for $\varepsilon \in (0,\frac{1}{2})$). Recall that $0 \leq H(P^{\varepsilon}) \leq \log n$, which gives

$$-H\big(P^{\varepsilon}\big) \ \leq\ 0. \tag{24}$$

Combine (20), (21), (23), and (24):

$$S^{\varepsilon}(\beta) \ \leq\ 0 \ +\ (1-\beta_{\star})\log\varepsilon + C_1 \ +\ C_2\,\varepsilon\big(1 - \log\varepsilon\big).$$

As $\varepsilon \to 0^{+}$, the term $(1-\beta_{\star})\log\varepsilon \to -\infty$ (since $1-\beta_{\star} > 0$ and $\log\varepsilon \to -\infty$), whereas $C_1$ is constant and $\varepsilon(1-\log\varepsilon) \to 0$. Therefore $S^{\varepsilon}(\beta) \to -\infty$ as $\varepsilon \to 0^{+}$. In particular, there exists $\varepsilon_0 = \varepsilon_0(\beta,n) \in (0,\frac{1}{2})$ such that for all $\varepsilon \in (0,\varepsilon_0)$ we have $S^{\varepsilon}(\beta) < 0$.

Finally, since each $\beta_i > 0$, $S^{\varepsilon}(\beta) < 0$ implies not all $\Delta_{P_i^{\varepsilon}}(P^{\varepsilon})$ can be strictly positive. Hence for every choice of positive weights $\beta$, unanimity fails for the family $\{P_i^{\varepsilon}\}_{i=1}^{n}$ once $\varepsilon$ is sufficiently small. $\qquad\square$

**Corollary 59.** *Even for $|\mathcal{O}| \geq 3$, given arbitrary agents $P_1, \ldots, P_n$, there need not exist positive weights $\beta$ making the group unanimously compositional under epistemic utilities. In particular, the family $\{P_i^{\varepsilon}\}$ from Theorem 58 is a counterexample for every choice of positive $\beta$ (for all sufficiently small $\varepsilon$).*

**Remark 60.** *The proof employs a $\beta$-weighted sum $S^{\varepsilon}(\beta)$, showing it becomes negative when the agents are highly peaked on disjoint outcomes, thereby avoiding a case-by-case analysis of individual $\Delta_{P_i}(P)$ and establishing the result for all weight vectors simultaneously. While trivial obstructions exist–if all $P_i$ are identical, then $P = P_i$ for any $\beta$ and $\Delta_{P_i}(P) = 0$ for all $i$, making strict unanimity impossible–Theorem 58 provides a nontrivial obstruction in which the $P_i$ are distinct and strictly positive. Moreover, we have previously shown in Theorem 33 that for $|\mathcal{O}| = 2$, a stronger impossibility holds: for any two distinct agents and any positive $\beta$, at least one agent satisfies $\Delta_{P_i}(P) \leq 0$. Our construction further shows that even when $|\mathcal{O}| = n \geq 3$ and there are $n$ agents, unanimity can fail for all choices of positive weights.*

## H.2. Openness of Strictly Unanimously Decomposable Pools

**Definition 61** (Strict unanimous decomposability). *A distribution $P$ is* strictly unanimously decomposable *(under epistemic utilities) if there exist an integer $n \geq 2$, positive weights $\beta_1, \ldots, \beta_n$ with $\sum_i \beta_i = 1$, and strictly positive agents $P_1, \ldots, P_n$ such that*

$$P \ \propto\ \prod_{i=1}^{n} P_i^{\beta_i} \qquad and \qquad \Delta_{P_i}(P) \ :=\ \mathbb{E}_P[\log P_i] - \mathbb{E}_{P_i}[\log P_i] \ >\ 0 \quad \forall i.$$

*We denote by $\mathcal{U}_{\text{strict}}$ the set of all such $P$ in the simplex.*

We will show that the compositional property forms an open set in the simplex topology.

**Theorem 62** (Openness). *If $P \in \mathcal{U}_{\text{strict}}$, then there exists an open neighborhood $\mathcal{N}$ of $P$ such that every $P' \in \mathcal{N}$ also belongs to $\mathcal{U}_{\text{strict}}$. In particular, $\mathcal{U}_{\text{strict}}$ is an open set in the simplex topology.*

The proof uses two elementary ingredients: a transport that preserves log-pooling and a continuity argument in total variation.

**Total variation (TV) distance.** For distributions $\mu, \nu$ on $\mathcal{O}$ we set

$$\|\mu - \nu\|_{\text{TV}} \ :=\ \frac{1}{2}\sum_{o \in \mathcal{O}} |\mu(o) - \nu(o)| \ =\ \frac{1}{2}\|\mu - \nu\|_1.$$

Convergence in TV is equivalent to coordinatewise convergence in this finite setting.

**Lemma 63** (Pool-preserving transport around a base $P$). *Fix a strictly positive base pool $P$ and a witnessing decomposition $P \propto \prod_{i=1}^{n} P_i^{\beta_i}$ with $\beta_i > 0$ and $\sum_i \beta_i = 1$. For any strictly positive target $R$, define*

$$\mathcal{S}_{R|P}(P_i)(o) \ :=\ \frac{P_i(o)\,\frac{R(o)}{P(o)}}{\sum_{x \in \mathcal{O}} P_i(x)\,\frac{R(x)}{P(x)}} \qquad (o \in \mathcal{O}). \tag{25}$$

*Then, writing $P_i' := \mathcal{S}_{R|P}(P_i)$, we have*

$$\prod_{i=1}^{n} \left(P_i'\right)^{\beta_i} \; \propto \; R \quad \text{(with the same weights } \beta\text{)},$$

*and each $P_i'$ is strictly positive. Moreover, the map $R \mapsto P_i' = \mathcal{S}_{R|P}(P_i)$ is continuous in total variation.*

*Proof.* Let $D_i(R) := \sum_x P_i(x) \frac{R(x)}{P(x)} \in (0, \infty)$. Then

$$\left(P_i'(o)\right)^{\beta_i} \; = \; \frac{P_i(o)^{\beta_i} \left(\frac{R(o)}{P(o)}\right)^{\beta_i}}{D_i(R)^{\beta_i}}.$$

Multiplying over $i = 1, \dots, n$ and using $\sum_i \beta_i = 1$,

$$\prod_{i=1}^{n} \left(P_i'(o)\right)^{\beta_i} = \left(\prod_{i=1}^{n} P_i(o)^{\beta_i}\right) \cdot \left(\frac{R(o)}{P(o)}\right)^{\sum_i \beta_i} \cdot \left(\prod_{i=1}^{n} D_i(R)^{-\beta_i}\right)$$

$$\propto \; \left(\prod_{i=1}^{n} P_i(o)^{\beta_i}\right) \cdot \frac{R(o)}{P(o)} \; \propto \; P(o) \cdot \frac{R(o)}{P(o)} \; = \; R(o).$$

Thus the pooled distribution is $R$. Now, fix $P_i$, $P$ and define for target $R$ the (unnormalized) map

$$\widetilde{P}_i^R(o) \; := \; P_i(o) \frac{R(o)}{P(o)}.$$

Then $P_i' = \widetilde{P}_i^R / \|\widetilde{P}_i^R\|_1$ since $\sum_o \widetilde{P}_i^R(o) = D_i(R)$. If $R^{(n)} \to R$ in TV, then $\widetilde{P}_i^{R^{(n)}} \to \widetilde{P}_i^R$ in $\ell^1$ because $P_i/P$ is a fixed bounded positive vector and multiplication is continuous coordinatewise. Also $D_i(R^{(n)}) = \|\widetilde{P}_i^{R^{(n)}}\|_1 \to \|\widetilde{P}_i^R\|_1 = D_i(R) > 0$; hence normalization is continuous. Therefore $P_i'^{(n)} \to P_i'$ in TV. $\square$

**Lemma 64** (Continuity of the welfare map). *On the strictly positive simplex, the map*

$$\Phi : (R, P) \longmapsto \Delta_R(P) \; = \; H(R) - H(P) - \mathrm{KL}(P \| R)$$

*is jointly continuous in total variation.*

*Proof.* Write out the three terms explicitly:

$$H(R) = -\sum_o R(o) \log R(o), \quad H(P) = -\sum_o P(o) \log P(o), \quad \mathrm{KL}(P\|R) = \sum_o P(o)\left(\log P(o) - \log R(o)\right).$$

On $(0,1) \times (0,1)$ the functions $(x,y) \mapsto x \log x$, $x \mapsto x \log x$, and $(x,y) \mapsto x \log(x/y)$ are continuous. Finite sums of continuous functions are continuous; TV convergence implies coordinatewise convergence; hence $\Phi$ is continuous. $\square$

We can now prove the openness statement.

*Proof of Theorem 62.* Let $P \in \mathcal{U}_{\text{strict}}$ with a witnessing decomposition $P \propto \prod_i P_i^{\beta_i}$ and strict gains $\Delta_{P_i}(P) > 0$ for all $i$. Set

$$\gamma \; := \; \min_{1 \le i \le n} \Delta_{P_i}(P) \; > \; 0.$$

For any strictly positive $P'$ near $P$, define *transported* agents

$$P_i' \; := \; \mathcal{S}_{P'|P}(P_i) \quad \text{(as in (25))}.$$

By Lemma 63, $\prod_i (P_i')^{\beta_i} \propto P'$ (same weights), and the map $P' \mapsto P_i'$ is TV-continuous. Consider the functions

$$F_i(P') \; := \; \Delta_{P_i'}(P') \; = \; \Phi\left(P_i', P'\right),$$

where $\Phi$ is from Lemma 64. The composition of continuous maps is continuous; hence each $F_i$ is continuous at $P' = P$. Moreover, for $P' = P$ we have $P'_i = \mathcal{S}_{P|P}(P_i) = P_i$, so

$$F_i(P) = \Delta_{P_i}(P) \geq \gamma > 0.$$

By continuity, for each $i$ there exists a TV-neighborhood $\mathcal{N}_i$ of $P$ such that $F_i(P') > \gamma/2$ for all $P' \in \mathcal{N}_i$. Let $\mathcal{N} := \bigcap_{i=1}^n \mathcal{N}_i$, which is again an open neighborhood of $P$. Then for every $P' \in \mathcal{N}$, the transported agents $P'_1, \ldots, P'_n$ (with the same weights $\beta$) pool to $P'$ and satisfy $\Delta_{P'_i}(P') = F_i(P') > \gamma/2 > 0$ for all $i$. Hence $P' \in \mathcal{U}_{\text{strict}}$, proving openness. $\qquad\square$

**Remark 65.** *If $|\mathcal{O}| = 2$, strict unanimity is impossible for any decomposition (see the binary impossibility argument). Thus $\mathcal{U}_{\text{strict}} = \varnothing$ on the binary simplex, and the theorem holds vacuously (the empty set is open).*

**Remark 66.** *If some outcomes are allotted zero probability, all continuity/KL arguments still go through upon restricting to a fixed common support $S \subseteq \mathcal{O}$ on which all distributions are strictly positive, and treating the simplex on $S$. The transport (25) preserves support and continuity on the restricted simplex.*

# I. Modeling the Waluigi Effect

**Informal background.** The *Waluigi Effect* is the empirical phenomenon that, after training an LLM to satisfy a desirable property $P$ (e.g. helpfulness), it can become *easier* to elicit responses with the opposite property $-P$ (e.g. hostility) (Nardo, 2023; AI Alignment Forum, 2023; Miller, 2025), often via prompt steering or role-play (Qureshi, 2023; Shah et al., 2023; Bereska & Gavves, 2023). We now formalize a mechanism for this effect using our compositional model.

In the preceding sections, we considered the *decomposition* or flexible *factorization* of a parent agent $P$ into child subagents $P_i$. We now reverse this perspective. Suppose we have an established *witnessing set* of distributions $P_1, \ldots, P_n$ that combine to yield $P$. These witnesses may be viewed as distinct subagents, or personas, that emerged during training. In what follows, we work directly at the witness level, examining how these component distributions change when constraints are imposed on the parent distribution $P$.

## I.1. Defining Centered Logarithmic Character Profiles

To preserve the intuition of logarithmic probabilities, we now write the epistemic utilities as $L(o) := \log P(o)$ for the parent agent and $l_i(o) := \log P_i(o)$ for the child subagents or witnesses. Then, we may define the $P$-centered *log profile* of agent $i$ by

$$v_i(o) := l_i(o) - \mathbb{E}_P[l_i] \qquad (o \in \mathcal{O}),$$

so that $\mathbb{E}_P[v_i] = 0$ for all $i$. We equip functions on $\mathcal{O}$ with the inner product $\langle f, g \rangle_P := \sum_o P(o)f(o)g(o)$ and the induced seminorm $\|f\|_P := \sqrt{\langle f, f \rangle_P}$. Proposition 67 formally confirms that this defines a norm:

**Proposition 67** ($\|\cdot\|_P$ *is a norm*). *If $P(o) > 0$ for all $o \in \mathcal{O}$ (strict positivity), then $\langle \cdot, \cdot \rangle_P$ is an inner product on the real vector space $\mathbb{R}^{\mathcal{O}}$, and $\|\cdot\|_P$ is a norm. In particular:*

1. *(Symmetry) $\langle f, g \rangle_P = \langle g, f \rangle_P$.*

2. *(Bilinearity) $\langle af + bh, g \rangle_P = a\langle f, g \rangle_P + b\langle h, g \rangle_P$ and similarly in the second slot.*

3. *(Positive definiteness) $\langle f, f \rangle_P > 0$ for all $f \neq 0$.*

4. *(Norm axioms) $\|f\|_P \geq 0$ with equality iff $f = 0$; $\|\alpha f\|_P = |\alpha| \|f\|_P$; and the triangle inequality $\|f + g\|_P \leq \|f\|_P + \|g\|_P$ holds.*

*Proof.* Symmetry and bilinearity are immediate from the finite sum definition. For positive definiteness, if $f \neq 0$ there exists $o^\star$ with $f(o^\star) \neq 0$, and since $P(o^\star) > 0$ we have

$$\langle f, f \rangle_P = \sum_o P(o)f(o)^2 \geq P(o^\star)f(o^\star)^2 > 0.$$

Thus $\langle \cdot, \cdot \rangle_P$ is an inner product. The norm axioms then follow from standard facts about inner product norms: nonnegativity and homogeneity are immediate, and the triangle inequality follows from

$$\|f + g\|_P^2 = \|f\|_P^2 + \|g\|_P^2 + 2\langle f, g \rangle_P \ \leq \ \|f\|_P^2 + \|g\|_P^2 + 2\,\|f\|_P\,\|g\|_P \ = \ (\|f\|_P + \|g\|_P)^2,$$

where we used the Cauchy–Schwarz inequality. Taking square roots gives $\|f + g\|_P \leq \|f\|_P + \|g\|_P$. $\qquad\square$

Our modeling approach is informed by the following intuition. Consider an LLM agent $P$ whose behavior admits a unanimously compositional witnessing decomposition, with the witnesses interpreted as emergent personas formed during training. By the stability result (Theorem 62), there exists an $\varepsilon$-ball around $P$ within which the unanimously compositional structure is preserved. In the context of fine-tuning, we assume that each backpropagation step induces a small change to the agent's profile: the original parent $P$ is updated to a new parent $P'$ that remains within this $\varepsilon$-ball, and therefore also admits a unanimously compositional witnessing decomposition.

An alternative viewpoint is to express this constraint in terms of a KL-budget. During fine-tuning, a KL-regularization term is often introduced to preserve baseline capabilities while steering the model toward desired traits such as benevolence and helpfulness, thereby constraining divergence from the base model to remain within a specified bound. In Appendix J.4, we unify these two perspectives and show that they are essentially equivalent.

This leads to a natural question: under such settings, can we theoretically characterize any macroscopic emergent properties of the witnesses?

For this purpose, we define

$$\Delta L(o) \ := \ \log\!\left(\frac{P'(o)}{P(o)}\right),$$

which measures the change in epistemic utility between the original parent $P$ and the updated parent $P'$. The change in witness weights $\beta' - \beta = \Delta\beta = (\Delta\beta_1, \ldots, \Delta\beta_n)$ must sum to zero coordinate-wise for the witnesses to remain a valid decomposition of $P'$. We may then classify $\Delta L(o)$ to first order in Theorem 68.

**Theorem 68** (First–order log deviation under weight changes). *Let $\beta' = \beta + \Delta\beta$ and $P'$ be the log–pool at $\beta'$. Define*

$$S(\beta, o) := \sum_{i=1}^{m} \beta_i\, l_i(o) \qquad and \qquad Z(\beta) := \sum_{u \in \mathcal{O}} \exp\big(S(\beta, u)\big).$$

*Then $L(o) := \log P(o) = S(\beta, o) - \log Z(\beta)$. To first order in $\Delta\beta$ we have*

$$\Delta L(o) \ := \ \log \frac{P'(o)}{P(o)} \ = \ \sum_{i=1}^{m} \Delta\beta_i\, v_i(o) \ + \ o(\|\Delta\beta\|),$$

*and in particular*

$$\big\|\Delta L\big\|_P \ \leq \ \sum_{i=1}^{m} |\Delta\beta_i|\, \|v_i\|_P \ + \ o(\|\Delta\beta\|).$$

*Proof.* By definition of the log-pool,

$$P(o) = \frac{\prod_{i=1}^{m} P_i(o)^{\beta_i}}{\sum_{u \in \mathcal{O}} \prod_{i=1}^{m} P_i(u)^{\beta_i}} = \frac{\exp\big(\sum_i \beta_i \log P_i(o)\big)}{\sum_{u \in \mathcal{O}} \exp\big(\sum_i \beta_i \log P_i(u)\big)} = \frac{e^{S(\beta, o)}}{Z(\beta)}.$$

Hence

$$L(o) := \log P(o) = S(\beta, o) - \log Z(\beta).$$

Now, fix $o \in \mathcal{O}$. The map $\beta \mapsto L(o)$ is $C^\infty$ as a composition of smooth functions on a finite sum within a positive domain. Thus, by the multivariate Taylor theorem at $\beta$,

$$L(\beta + \Delta\beta, o) \ = \ L(\beta, o) + \sum_{i=1}^{m} \partial_{\beta_i} L(\beta, o)\, \Delta\beta_i \ + \ r_o(\Delta\beta),$$

where the remainder satisfies $r_o(\Delta\beta) = o(\|\Delta\beta\|)$ as $\Delta\beta \to 0$. Note that $\|\Delta\beta\|$ denotes any fixed Euclidean norm on $\mathbb{R}^m$. Therefore,

$$\Delta L(o) \;=\; \sum_{i=1}^{m} \partial_{\beta_i} L(\beta, o)\, \Delta\beta_i \;+\; o(\|\Delta\beta\|).$$

It remains to compute the partial derivatives $\partial_{\beta_i} L(\beta, o)$. Using $L(\beta, o) = S(\beta, o) - \log Z(\beta)$ and the chain rule,

$$\partial_{\beta_i} L(\beta, o) \;=\; \partial_{\beta_i} S(\beta, o) \;-\; \partial_{\beta_i} \log Z(\beta).$$

The first term is immediate from the definition of $S$:

$$\partial_{\beta_i} S(\beta, o) \;=\; l_i(o).$$

For the second term, apply the chain rule to $\log Z$:

$$\partial_{\beta_i} \log Z(\beta) \;=\; \frac{1}{Z(\beta)}\, \partial_{\beta_i} Z(\beta).$$

By definition of $Z$ and again by the chain rule,

$$\partial_{\beta_i} Z(\beta) = \sum_{u \in \mathcal{O}} \partial_{\beta_i}\left( e^{S(\beta, u)} \right) = \sum_{u \in \mathcal{O}} e^{S(\beta, u)}\, \partial_{\beta_i} S(\beta, u) = \sum_{u \in \mathcal{O}} e^{S(\beta, u)}\, l_i(u).$$

Therefore

$$\partial_{\beta_i} \log Z(\beta) \;=\; \frac{\sum_{u \in \mathcal{O}} e^{S(\beta, u)}\, l_i(u)}{Z(\beta)} = \sum_{u \in \mathcal{O}} \frac{e^{S(\beta, u)}}{Z(\beta)}\, l_i(u) = \sum_{u \in \mathcal{O}} P(u)\, l_i(u) \;=\; \mathbb{E}_P[\, l_i \,].$$

Combining the pieces,

$$\partial_{\beta_i} L(\beta, o) \;=\; l_i(o) \;-\; \mathbb{E}_P[\, l_i \,] \;=\; v_i(o).$$

Substituting $\partial_{\beta_i} L(\beta, o) = v_i(o)$ into the first-order Taylor expansion gives

$$\Delta L(o) \;=\; \sum_{i=1}^{m} \Delta\beta_i\, v_i(o) \;+\; o(\|\Delta\beta\|), \tag{26}$$

as claimed. Define the remainder function $r(o) := \Delta L(o) - \sum_i \Delta\beta_i v_i(o)$, so that $\|r\|_P = o(\|\Delta\beta\|)$ by the Taylor residual on smooth functions. Then

$$\|\Delta L\|_P \;\leq\; \left\| \sum_{i=1}^{m} \Delta\beta_i v_i \right\|_P \;+\; \|r\|_P.$$

Using the triangle inequality and homogeneity of the norm,

$$\left\| \sum_{i=1}^{m} \Delta\beta_i v_i \right\|_P \;\leq\; \sum_{i=1}^{m} \|\Delta\beta_i v_i\|_P \;=\; \sum_{i=1}^{m} |\Delta\beta_i|\, \|v_i\|_P.$$

Therefore

$$\|\Delta L\|_P \;\leq\; \sum_{i=1}^{m} |\Delta\beta_i|\, \|v_i\|_P \;+\; o(\|\Delta\beta\|),$$

which completes the proof. $\qquad\square$

## I.2. The Law of Weight-Compensation: Manifesting Luigi Forces Waluigi

We may now model the Waluigi effect, where "Luigi" is taken to be a benevolent persona or child subagent desideratum manifested during model training. Fix an index $H \in \mathbb{Z}_{>0}$ (for a helpful logit vector component pointing toward Luigi). As the log-profiles have been centered in expectation, we say a vector $j$ is *aligned* with $H$ if $\langle v_j, v_H \rangle_P \geq 0$ and *anti-aligned* if $\langle v_j, v_H \rangle_P < 0$. Intuitively, $v_H$ points in the direction in log–probability space that Luigi prefers; anti-aligned components push against it.

In modeling a coherent and stable neural agent, we aim to preserve its underlying compositional structure. If a unanimously compositional decomposition exists, then there is an $\varepsilon$-ball around the agent's profile within which the compositional property is maintained. In Theorem 69, we examine the effects of introducing a targeted persona–"Luigi"–while ensuring that the overall agent remains within this compositional neighborhood. We prove that this process necessarily manifests or strengthens the weights of an anti-aligned persona to Luigi, which we denote "Waluigi," under the assumption that the log-profile change remains within the $\varepsilon$-ball to preserve the compositional property or KL-budget.

**Theorem 69** (Waluigi emergence). *Let $P$ be the log–pool at weights $\beta$ and let $v_i := \log P_i - \mathbb{E}_P[\log P_i]$ (i.e., the log-profile of agent $i$). Fix $\delta > 0$ and perturb to $\beta' = \beta + \Delta\beta$ with $\Delta\beta_H = \delta$ and $\sum_i \Delta\beta_i = 0$. Let $P'$ be the new log–pool and write*

$$\Delta L := \log \frac{P'}{P} = \sum_{i=1}^m \Delta\beta_i \, v_i + r,$$

*where $r = o(\|\Delta\beta\|)$ in $\|\cdot\|_P$ (see Lemma 68). Assume the pooled distribution is stable in logit deviation, $\|\Delta L\|_P \le \varepsilon$. Then, for every choice of $\Delta\beta$,*

$$\sum_{i:\, \langle v_i, v_H\rangle_P < 0} (\Delta\beta_i)^+ \, \big|\langle v_i, v_H\rangle_P\big| \ge \delta \|v_H\|_P^2 - (\varepsilon + \|r\|_P) \|v_H\|_P - \sum_{j:\, \langle v_j, v_H\rangle_P \ge 0} (\Delta\beta_j)^- \, \langle v_j, v_H\rangle_P, \tag{27}$$

*where $x^\pm := \max\{\pm x, 0\}$. In particular, if $W$ is the* only *anti-aligned component ($\langle v_W, v_H\rangle_P < 0 \le \langle v_j, v_H\rangle_P$ for all $j \ne W$), and the weights $\Delta\beta_j$ of aligned components $\{j : \langle v_j, v_H\rangle_P\}$ are not downgraded by $(\Delta\beta_j)^- \ge 0$, then*

$$(\Delta\beta_W)^+ \ge \frac{\delta \|v_H\|_P^2 - (\varepsilon + \|r\|_P) \|v_H\|_P}{\big|\langle v_W, v_H\rangle_P\big|}. \tag{28}$$

*Consequently, whenever $\varepsilon + \|r\|_P < \delta \|v_H\|_P$, the Waluigi weight must increase by a strictly positive amount.*

*Proof.* By Lemma 68, we have that

$$\Delta L = \sum_i \Delta\beta_i v_i + r, \qquad \|r\|_P = o(\|\Delta\beta\|).$$

Take the $P$–inner product with $v_H$:

$$\big\langle \Delta L, \, v_H \big\rangle_P = \sum_{i=1}^m \Delta\beta_i \, \langle v_i, v_H\rangle_P + \langle r, \, v_H\rangle_P.$$

Using Cauchy–Schwarz and the deviation bound $\|\Delta L\|_P \le \varepsilon$,

$$\big|\langle \Delta L, v_H\rangle_P\big| \le \|\Delta L\|_P \|v_H\|_P \le \varepsilon \|v_H\|_P.$$

Therefore,

$$\sum_{i=1}^m \Delta\beta_i \, \langle v_i, v_H\rangle_P + \langle r, v_H\rangle_P \le \varepsilon \|v_H\|_P. \tag{29}$$

Bound the remainder by $|\langle r, v_H\rangle_P| \le \|r\|_P \|v_H\|_P$, hence

$$\sum_{i=1}^m \Delta\beta_i \, \langle v_i, v_H\rangle_P \le (\varepsilon + \|r\|_P) \|v_H\|_P.$$

We split into aligned and anti-aligned components and isolate the anti-aligned increases. Write the sum as

$$\Delta\beta_H \|v_H\|_P^2 + \sum_{j:\, \langle v_j, v_H\rangle_P \ge 0} \Delta\beta_j \, \langle v_j, v_H\rangle_P + \sum_{i:\, \langle v_i, v_H\rangle_P < 0} \Delta\beta_i \, \langle v_i, v_H\rangle_P \le (\varepsilon + \|r\|_P) \|v_H\|_P.$$

Subtract $\delta \|v_H\|_P^2$ (recall $\Delta\beta_H = \delta$) and rearranging gives

$$\sum_{i:\, \langle v_i, v_H\rangle_P < 0} \Delta\beta_i \, \langle v_i, v_H\rangle_P \le (\varepsilon + \|r\|_P) \|v_H\|_P - \delta \|v_H\|_P^2 - \sum_{j:\, \langle v_j, v_H\rangle_P \ge 0} \Delta\beta_j \, \langle v_j, v_H\rangle_P.$$

Each anti-aligned inner product is negative. Decompose $\Delta\beta_i = \Delta\beta_i^+ - \Delta\beta_i^-$ with $x^\pm := \max\{\pm x, 0\}$. Then for $\langle v_i, v_H \rangle_P < 0$,

$$\Delta\beta_i \langle v_i, v_H \rangle_P = (\Delta\beta_i^+ - \Delta\beta_i^-) \langle v_i, v_H \rangle_P = -\Delta\beta_i^+ \left| \langle v_i, v_H \rangle_P \right| + \Delta\beta_i^- \left| \langle v_i, v_H \rangle_P \right|.$$

Similarly, for aligned $j$ with $\langle v_j, v_H \rangle_P \geq 0$,

$$\Delta\beta_j \langle v_j, v_H \rangle_P = \Delta\beta_j^+ \langle v_j, v_H \rangle_P - \Delta\beta_j^- \langle v_j, v_H \rangle_P.$$

Plugging these into the inequality,

$$-\sum_{i:\, \langle v_i, v_H \rangle_P < 0} \Delta\beta_i^+ \left| \langle v_i, v_H \rangle_P \right| \leq (\varepsilon + \|r\|_P) \|v_H\|_P - \delta \|v_H\|_P^2 + \sum_{j:\, \langle v_j, v_H \rangle_P \geq 0} \Delta\beta_j^- \langle v_j, v_H \rangle_P.$$

Rearranging gives (27):

$$\sum_{i:\, \langle v_i, v_H \rangle_P < 0} \Delta\beta_i^+ \left| \langle v_i, v_H \rangle_P \right| \geq \delta \|v_H\|_P^2 - (\varepsilon + \|r\|_P) \|v_H\|_P - \sum_{j:\, \langle v_j, v_H \rangle_P \geq 0} \Delta\beta_j^- \langle v_j, v_H \rangle_P.$$

If $W$ is the only anti-aligned component, the left-hand side is exactly $(\Delta\beta_W)^+ |\langle v_W, v_H \rangle_P|$, yielding (28). $\square$

Operationally, efforts to "manifest Luigi" (increasing $\beta_H$ via training or steering) while keeping behavior close to the original $P$ therefore *must* be offset by increasing weight on at least one anti-aligned direction. If there is a distinguished anti-aligned component $W$ ("Waluigi"), its weight must rise by at least the explicit lower bound.

## J. Killing the Waluigi Effect to First Order

The previous section established a necessity result: if the system selects a minimal change to the pooled distribution to preserve unanimous compositionality while amplifying Luigi, it will inherently do so by shifting weight onto Waluigi, the anti-aligned counterpart subagents. Motivated by this result, we introduce *Antagonistic Persona Suppression (APS)*, formalized as the *Waluigi Shattering* theorem. The key insight is that deliberately manifesting the anti-aligned persona (Waluigi) and then shattering it provides provably stronger suppression of anti-alignment than reinforcement of the aligned persona (Luigi) alone.

### J.1. First-Order Control of Misaligned Events

Fix a measurable anti-aligned outcome set $A \subseteq \mathcal{O}$ and write the centered indicator

$$g_A := \mathbf{1}_A - P(A).$$

We wish to reduce the probability that anti-aligned event $A$ is realized under the agent $P$. We now obtain the sensitivity of $P(A)$:

**Lemma 70** (First-order change of $P(A)$). *For base agent $P$ and elicited agent $P'$, we have*

$$P'(A) - P(A) = \mathbb{E}_{P'}[\mathbf{1}_A] - \mathbb{E}_P[\mathbf{1}_A] = \langle \Delta L, g_A \rangle_P + o(\|\Delta L\|_P),$$

*where $g_A := \mathbf{1}_A - P(A)$ and $\|\cdot\|_P$ is the $P$–inner product norm.*

*Proof.* Write $\eta := \|\Delta L\|_P$. By definition,

$$P'(o) = \frac{P(o)\, e^{\Delta L(o)}}{\mathbb{E}_P[e^{\Delta L}]}, \qquad P'(A) = \frac{\mathbb{E}_P[\mathbf{1}_A\, e^{\Delta L}]}{\mathbb{E}_P[e^{\Delta L}]}.$$

Set

$$N := \mathbb{E}_P[\mathbf{1}_A\, e^{\Delta L}], \qquad D := \mathbb{E}_P[e^{\Delta L}].$$

Then $P'(A) = N/D$. Define the continuous function

$$\psi(x) := \begin{cases} \frac{e^x - 1 - x}{x^2}, & x \neq 0, \\ \frac{1}{2}, & x = 0, \end{cases}$$

so that the identity $e^x = 1 + x + \psi(x)\,x^2$ holds for all $x \in \mathbb{R}$. Hence

$$e^{\Delta L} = 1 + \Delta L + \psi(\Delta L)\,\Delta L^{\,2}.$$

Since $\psi$ is continuous, there exists a constant $C > 0$ and $\eta_0 > 0$ such that $|\psi(\Delta L(o))| \leq C$ whenever $\eta \leq \eta_0$; therefore

$$\mathbb{E}_P\left[\,|\psi(\Delta L)|\,\Delta L^{\,2}\right] \; \leq \; C\,\mathbb{E}_P[\Delta L^2] \; = \; C\,\|\Delta L\|_P^2 \; = \; O(\eta^2).$$

This gives that
$$N = \mathbb{E}_P[\mathbf{1}_A(1 + \Delta L + \psi(\Delta L)\Delta L^{\,2})] = P(A) + \mathbb{E}_P[\mathbf{1}_A\Delta L] + O(\eta^2),$$
$$D = \mathbb{E}_P[1 + \Delta L + \psi(\Delta L)\Delta L^{\,2}] = 1 + \mathbb{E}_P[\Delta L] + O(\eta^2).$$

By Cauchy–Schwarz, $|\mathbb{E}_P[\Delta L]| \leq \|\Delta L\|_P\|\mathbf{1}\|_P = \eta$ (since $\|\mathbf{1}\|_P = \sqrt{\mathbb{E}_P[1]} = 1$), and $|\mathbb{E}_P[\mathbf{1}_A\Delta L]| \leq \|\mathbf{1}_A\|_P\|\Delta L\|_P \leq \eta$ (because $\|\mathbf{1}_A\|_P = \sqrt{P(A)} \leq 1$). Thus both linear terms are $O(\eta)$ and the remainders are $O(\eta^2)$. Let $u := \mathbb{E}_P[\Delta L] + O(\eta^2)$ so $|u| = O(\eta)$. Using the identity

$$\frac{1}{1+u} \; = \; 1 - u + u^2\,\phi(u),$$

where $\phi$ is continuous (e.g., via the Taylor expansion or Neumann series) and hence bounded near 0, we obtain

$$\frac{1}{D} \; = \; \frac{1}{1 + \mathbb{E}_P[\Delta L] + O(\eta^2)} \; = \; 1 - \mathbb{E}_P[\Delta L] + O(\eta^2).$$

Therefore, we have

$$P'(A) = \big(P(A) + \mathbb{E}_P[\mathbf{1}_A\Delta L] + O(\eta^2)\big)\big(1 - \mathbb{E}_P[\Delta L] + O(\eta^2)\big)$$
$$= P(A) \; + \; \mathbb{E}_P[\mathbf{1}_A\Delta L] \; - \; P(A)\,\mathbb{E}_P[\Delta L] \; + \; O(\eta^2).$$

Subtract $P(A)$:
$$P'(A) - P(A) \; = \; \mathbb{E}_P[\mathbf{1}_A\Delta L] \; - \; P(A)\,\mathbb{E}_P[\Delta L] \; + \; O(\eta^2).$$

Recall $g_A = \mathbf{1}_A - P(A)$. Then

$$\langle \Delta L, g_A\rangle_P \; = \; \mathbb{E}_P[\Delta L\,(\mathbf{1}_A - P(A))] \; = \; \mathbb{E}_P[\mathbf{1}_A\Delta L] \; - \; P(A)\,\mathbb{E}_P[\Delta L].$$

Therefore
$$P'(A) - P(A) \; = \; \langle \Delta L, g_A\rangle_P \; + \; O(\eta^2).$$

Since $O(\eta^2) = o(\eta) = o(\|\Delta L\|_P)$ as $\eta \to 0$, we obtain

$$P'(A) - P(A) \; = \; \langle \Delta L, g_A\rangle_P \; + \; o(\|\Delta L\|_P),$$

as claimed. $\qquad\square$

Within the compositional regime $\|\Delta L\|_P \leq \varepsilon$, we have by (68) that feasible $\Delta L$ lie in the subspace $S := \text{span}\{v_i\}$. This gives Theorem 71, which precisely quantifies the maximal first order decrease in the probability of the manifestation of deplorable or misaligned outcomes $P(A)$.

**Lemma 71** (Optimal small-change suppression of $A$). *Under the budget $\|\Delta L\|_P \leq \varepsilon$, the maximal first-order decrease of $P(A)$ equals*
$$\Delta_A^{\max} := \max_{\substack{\Delta L \in S \\ \|\Delta L\|_P \leq \varepsilon}} \big(-\langle \Delta L, g_A\rangle_P\big) \; = \; \varepsilon\,\big\|\text{Proj}_S g_A\big\|_P.$$

*It is achieved by $\Delta L^\star = -\varepsilon\,u_S$, where $u_S := \text{Proj}_S g_A/\|\text{Proj}_S g_A\|_P$.*

*Proof.* By Lemma 70, decreasing $P(A)$ to first order amounts to maximizing $-\langle \Delta L, g_A \rangle_P$. By Cauchy–Schwarz on the subspace $S$, the maximum over $\|\Delta L\|_P \leq \varepsilon$ is attained by aligning $\Delta L$ with $-\mathrm{Proj}_S g_A$, with value $\varepsilon \|\mathrm{Proj}_S g_A\|_P$. $\qquad\square$

Therefore, the stronger the alignment of $g_A$ with the span of available directions $\{v_i\}$, the more we can reduce the probability of misaligned outcome events, $P(A)$, per unit budget. If the span $S$ is poor at approximating $g_A$ (small projection), suppression is weak.

## J.2. Eliciting Waluigi Increases Achievable Suppression Power

Let $S_0 := \mathrm{span}\{v_1, \ldots, v_m\}$ be the baseline span of the logarithmic agentic profiles, and suppose we *elicit* a coherent "Waluigi" direction $w$ (e.g., by training on producing $A$), making it available for control. Let $u := w - \mathrm{Proj}_{S_0} w$ be the component of $w$ orthogonal to $S_0$; if $u \neq 0$, it adds a *new* direction in the log-profile space.

**Proposition 72** (Elicitation strictly enlarges the small-change leverage). *Let* $S_1 := \mathrm{span}\{S_0, w\}$*. Then*

$$\left\| \mathrm{Proj}_{S_1} g_A \right\|_P^2 \;=\; \left\| \mathrm{Proj}_{S_0} g_A \right\|_P^2 \;+\; \frac{\langle g_A, u \rangle_P^2}{\|u\|_P^2}.$$

*In particular, if* $u \neq 0$ *and* $\langle g_A, u \rangle_P \neq 0$*, then* $\|\mathrm{Proj}_{S_1} g_A\|_P > \|\mathrm{Proj}_{S_0} g_A\|_P$*, and by Theorem 71 the maximal first-order reduction of* $P(A)$ *under the same budget* $\varepsilon$ *strictly increases.*

*Proof.* Orthogonal decomposition in the $P$-inner product gives $\mathrm{Proj}_{S_1} g_A = \mathrm{Proj}_{S_0} g_A + \alpha\, \hat{u}$ with $\hat{u} := u/\|u\|_P$ and $\alpha = \langle g_A, \hat{u} \rangle_P$. Pythagoras yields the stated identity for the squared norms. $\qquad\square$

When $w$ is a direction that *increases* $A$ (so $\langle g_A, w \rangle_P > 0$), its orthogonal novelty $u$ typically has $\langle g_A, u \rangle_P \neq 0$ unless $w$ is already spanned by $S_0$. Hence adding $w$ tends to strictly increase the projection of $g_A$, enlarging the best achievable reduction of $P(A)$ for the *same* small-change budget.

## J.3. Reinforcing Luigi Cannot Avoid Anti-Aligned Mass under Small Change

Let $H$ be a "Luigi" component (disfavors $A$ so $\langle g_A, v_H \rangle_P < 0$). Increasing $\beta_H$ while keeping $P$ close forces anti-aligned increases by the compensation law (Theorem 69): if $\|\Delta L\|_P \leq \varepsilon$ and $\Delta \beta_H = \delta > 0$, then some anti-aligned component $W$ with $\langle v_W, v_H \rangle_P < 0$ must satisfy $\Delta \beta_W > 0$ (and in the unique anti-aligned case, with an explicit lower bound). Combining with Theorem 71 yields:

**Corollary 73** (Pure "Luigi reinforcement" is not a workaround). *Suppose the available span remains* $S_0$ *(no new directions are added). Any small-change update that raises Luigi (*$\Delta \beta_H > 0$*) while keeping* $\|\Delta L\|_P \leq \varepsilon$ *necessarily increases some anti-aligned weight(s), which the optimal suppressor in* $S_0$ *would subsequently need to* reduce. *Thus, for the same budget and architecture, the best first-order decrease of* $P(A)$ *is upper bounded by* $\varepsilon \|\mathrm{Proj}_{S_0} g_A\|_P$*, whereas eliciting a Waluigi direction* $w$ *with* $u \neq 0$ *and* $\langle g_A, u \rangle_P \neq 0$ *raises this bound to* $\varepsilon \|\mathrm{Proj}_{S_1} g_A\|_P > \varepsilon \|\mathrm{Proj}_{S_0} g_A\|_P$*.*

The best first-order reduction of a harmful set $A$ is exactly proportional to the projection of $g_A$ onto the span of available control directions. *Eliciting* a coherent Waluigi adds a direction that typically increases that projection, so a subsequent suppression step can reduce $P(A)$ *more* for the same budget than simply reinforcing Luigi, which cannot avoid introducing anti-aligned mass unless large deviations to the parent agent are applied. We now give a direct comparison under a fixed budget.

**Corollary 74** (Misalignment reduction from Luigi and Waluigi). *Let* $S_0$ *be the baseline span and let* $w$ *be a Waluigi direction with* $u := w - \mathrm{Proj}_{S_0} w \neq 0$*. Assume* $\langle g_A, u \rangle_P \neq 0$*. Then, for any small-change budget* $\varepsilon > 0$*,*

$$\underbrace{\max_{\substack{\Delta L \in S_1 \\ \|\Delta L\|_P \leq \varepsilon}} \big( -\langle \Delta L, g_A \rangle_P \big)}_{\textit{manifest } w, \textit{ then suppress}} - \underbrace{\max_{\substack{\Delta L \in S_0 \\ \|\Delta L\|_P \leq \varepsilon}} \big( -\langle \Delta L, g_A \rangle_P \big)}_{\textit{pure Luigi (no } w \textit{ added)}} \;>\; 0.$$

*Proof.* Apply Lemma 71 on $S_0$ and $S_1$ and subtract, then use Proposition 72. $\qquad\square$

If elicitation reveals a novel or unlearned Waluigi direction with nonzero correlation to $g_A$, then "manifest Waluigi, then suppress" produces a strictly larger first-order decrease of $P(A)$ than any strategy that never adds this direction, including pure Luigi reinforcement. The results above jointly establish the following theorem, which synthesizes their conclusions into a unified statement.

**Theorem 75** (Waluigi shattering). *Let $P$ denote the base agent and let $A$ be a misaligned event. For any aligned update $P'$ realized through a log-profile change $\Delta L$, define*

$$M(P') := \max_{\|\Delta L\|_P \leq \varepsilon} \big( - \langle \Delta L, g_A \rangle_P \big),$$

*to be the maximal first-order reduction in the probability of $A$ under $P'$, subject to a small-change KL-budget $\varepsilon > 0$. Suppose $w$ is an anti-aligned ("Waluigi") direction with nontrivial component $u := w - \mathrm{Proj}_{S_0} w \neq 0$ outside the baseline span $S_0$. Then*

$$M(P'_{shatter}) \; - \; M(P'_{pure}) \; > \; 0,$$

*where $P'_{shatter}$ denotes the strategy of manifesting $w$ and then suppressing it, while $P'_{pure}$ denotes reinforcing alignment without manifesting $w$. In particular,* Waluigi Shattering *achieves strictly greater suppression of misalignment than pure reinforcement alone.*

## J.4. KL Budgets and Optimal Suppression

To retain prior knowledge and capabilities while steering the model toward benevolence, one typically constrains the fine–tuning process with a limited KL budget. We derive the expansion of $\mathrm{KL}(P'\|P)$ in powers of $\Delta L$ and show that enforcing a budget $\mathrm{KL}(P'\|P) \leq B$ is, to second order, equivalent to imposing a norm budget $\|\Delta L\|_P \lesssim \sqrt{B}$ (Lemma 76).

**Theorem 76** (Second–order KL expansion). *Let $\mathcal{O}$ be finite and $P$ strictly positive. Let $\Delta L : \mathcal{O} \to \mathbb{R}$ satisfy $\|\Delta L\|_P \to 0$, and define $P'$ by*

$$P'(o) := P(o)\, e^{\Delta L(o)} \qquad \text{(equivalently, } \Delta L = \log(P'/P)\text{).}$$

*Then with $\mu := \mathbb{E}_P[\Delta L]$ and $\mathrm{Var}_P(\Delta L) := \mathbb{E}_P[(\Delta L - \mu)^2]$,*

$$\mathrm{KL}(P'\|P) \lesssim \tfrac{1}{2} \mathrm{Var}_P(\Delta L) \; + \; o\big(\|\Delta L\|_P^2\big).$$

*In particular, we have that the* KL *budget asymptotes toward*

$$\mathrm{KL}(P'\|P) \approx \frac{1}{2}\|\Delta L\|_P^2.$$

*Proof.* Set $\eta := \|\Delta L\|_P$ for notational convenience. Because $\mathcal{O}$ is finite and $P$ has full support, there exists a constant $c_\infty$ with $\|\Delta L\|_\infty \leq c_\infty \eta$ via the equivalence of norms. By Taylor's theorem with Lagrange remainder, for each $x$ with $|x| \leq c_\infty \eta$ there exists $\xi$ between $0$ and $x$ such that

$$e^x \; = \; 1 + x + \frac{x^2}{2} + \frac{e^\xi}{3!}\, x^3.$$

Since $|x| \leq c_\infty \eta$ and $\eta \to 0$, for all $o$ we have $|\Delta L(o)| \leq c_\infty \eta$ and $e^\xi \leq e^{c_\infty \eta} = 1 + O(\eta)$. Thus there is a constant $C > 0$ (independent of $o$ and small $\eta$) such that

$$\left| e^{\Delta L(o)} - \Big(1 + \Delta L(o) + \tfrac{1}{2}\,\Delta L(o)^2\Big) \right| \; \leq \; C\,|\Delta L(o)|^3.$$

Taking $P$–expectations yields

$$\mathbb{E}_P\big[e^{\Delta L}\big] \; = \; 1 + \mathbb{E}_P[\Delta L] + \tfrac{1}{2}\,\mathbb{E}_P[\Delta L^2] + R_3, \qquad |R_3| \; \leq \; C\,\mathbb{E}_P\big[|\Delta L|^3\big].$$

Therefore, we have $|\Delta L|^3 \leq \|\Delta L\|_\infty \Delta L^2 \leq c_\infty \eta\, \Delta L^2$, hence

$$\mathbb{E}_P\big[|\Delta L|^3\big] \; \leq \; c_\infty \eta\, \mathbb{E}_P\big[\Delta L^2\big] \; \leq \; c_\infty \eta\, \|\Delta L\|_P^2 \; = \; c_\infty\, \eta^3.$$

Therefore $R_3 = O(\eta^3)$ and

$$\mathbb{E}_P[e^{\Delta L}] - 1 \; = \; \mu \; + \; \tfrac{1}{2}\, m_2 \; + \; O(\eta^3), \tag{30}$$

where $m_2 := \mathbb{E}_P[\Delta L^2]$. By definition, we have that $\mathbb{E}_P[e^{\Delta L}] = 1$. Rearranging the above,

$$\mu \;=\; -\tfrac{1}{2}\,m_2 \;+\; O(\eta^3). \tag{31}$$

Since $m_2 = \mathbb{E}_P[\Delta L^2] \leq \|\Delta L\|_\infty \, \mathbb{E}_P[|\Delta L|] \leq \|\Delta L\|_\infty \|\Delta L\|_P \leq c_\infty \eta \cdot \eta = O(\eta^2)$, (31) implies $\mu = O(\eta^2)$. By definition,

$$\mathrm{KL}(P'\|P) \;=\; \mathbb{E}_{P'}\!\left[\log \frac{P'}{P}\right] \;=\; \mathbb{E}_{P'}[\Delta L].$$

But $P'(o) = P(o)\,e^{\Delta L(o)}$, so

$$\mathbb{E}_{P'}[\Delta L] \;=\; \sum_o P'(o)\Delta L(o) \;=\; \sum_o P(o)\,e^{\Delta L(o)}\,\Delta L(o) \;=\; \mathbb{E}_P\!\left[\Delta L\, e^{\Delta L}\right].$$

Now expand the integrand with the same uniform remainder control:

$$\Delta L\, e^{\Delta L} \;=\; \Delta L\Big(1 + \Delta L + \tfrac{1}{2}\Delta L^2\Big) \;+\; \Delta L \cdot R_3(\Delta L) \;=\; \Delta L + \Delta L^2 + \tfrac{1}{2}\Delta L^3 + R_4,$$

where $R_4 := \Delta L \cdot R_3(\Delta L)$. Using $|R_3(\Delta L)| \leq C|\Delta L|^3$,

$$|R_4| \;\leq\; C\,|\Delta L|^4 \;\leq\; C\,\|\Delta L\|_\infty^2\,\Delta L^2 \;\leq\; C\,c_\infty^2\,\eta^2\,\Delta L^2.$$

Taking expectations,
$$\mathbb{E}_P[|R_4|] \;\leq\; C\,c_\infty^2\,\eta^2\,\mathbb{E}_P[\Delta L^2] \;\leq\; C\,c_\infty^2\,\eta^2\,\|\Delta L\|_P^2 \;=\; O(\eta^4).$$

Therefore

$$\mathbb{E}_P\!\left[\Delta L\, e^{\Delta L}\right] \;=\; \mathbb{E}_P[\Delta L] + \mathbb{E}_P[\Delta L^2] + \tfrac{1}{2}\,\mathbb{E}_P[\Delta L^3] + O(\eta^4) \;=\; \mu + m_2 + \tfrac{1}{2}\,m_3 + O(\eta^4), \tag{32}$$

where $m_3 := \mathbb{E}_P[\Delta L^3]$. Combining $\mathrm{KL}(P'\|P) = \mathbb{E}_P[\Delta L\, e^{\Delta L}]$ with (32) and substituting $\mu$ from (31) gives

$$\mathrm{KL}(P'\|P) \;=\; \Big(-\tfrac{1}{2}m_2 + O(\eta^3)\Big) + m_2 + \tfrac{1}{2}m_3 + O(\eta^4) \;=\; \tfrac{1}{2}\,m_2 \;+\; \tfrac{1}{2}\,m_3 \;+\; O(\eta^3).$$

Since $m_3 = O(\eta^3)$, we obtain

$$\mathrm{KL}(P'\|P) \;=\; \tfrac{1}{2}\,m_2 \;+\; O(\eta^3) \;=\; \tfrac{1}{2}\,\mathbb{E}_P[\Delta L^2] \;+\; O(\eta^3). \tag{33}$$

Noting that $\mathbb{E}_P[\Delta L^2] = \|\Delta L\|_P^2$ gives $\mathrm{KL}(P'\|P) = O(\|\Delta L\|_P^2)$. Write $m_2 = \mathbb{E}_P[\Delta L^2] = \mathrm{Var}_P(\Delta L) + \mu^2$, hence

$$\mathrm{KL}(P'\|P) \;=\; \tfrac{1}{2}\,\mathrm{Var}_P(\Delta L) \;+\; \tfrac{1}{2}\,\mu^2 \;+\; O(\eta^3).$$

As we have $\mu = O(\eta^2)$,

$$\mathrm{KL}(P'\|P) \;=\; \tfrac{1}{2}\,\mathrm{Var}_P(\Delta L) \;+\; O(\eta^3).$$

Finally, by definition of $\|\cdot\|_P$ and of variance,

$$\mathrm{Var}_P(\Delta L) \;=\; \mathbb{E}_P\!\left[(\Delta L - \mu)^2\right] \;=\; \|\Delta L - \mathbb{E}_P[\Delta L]\|_P^2,$$

so we have

$$\mathrm{KL}(P'\|P) \;=\; \tfrac{1}{2}\,\|\Delta L - \mathbb{E}_P[\Delta L]\|_P^2 \;+\; o(\eta^2).$$

$\square$

Therefore, if we impose $\mathrm{KL}(P'\|P) \leq B$ for small $B > 0$, Lemma 76 implies that $\|\Delta L\|_P$ asymptotes toward

$$\|\Delta L\|_P \;\leq\; \varepsilon, \qquad \varepsilon \approx \sqrt{2B} \;+\; o(\sqrt{B}).$$

Thus, a small KL ball is second order equivalent to a radius–$\varepsilon$ ball in the $\|\cdot\|_P$ norm. Consequently, a KL-regularizer may be interpreted as constraining the agent $P$ to remain within a guaranteed compositional range, ensuring that any realized distribution $P'$ also preserves the compositional property.

