# OpenReview forum: "Probabilistic Modeling of Latent Agentic Substructures in Deep Neural Networks"
_ICML.cc/2026/Conference — ICML 2026 regular_

### Official Review · Reviewer_uc9Z · 2026-03-04

**Soundness:** 3
**Presentation:** 2
**Significance:** 1
**Originality:** 2
**Overall Recommendation:** 3
**Confidence:** 3

**Summary:**

The paper proposes a framework for describing agents as distributions over outcomes with welfare defined as the log-probability assigned to outcomes. Agents are aggregated via logarithm opinion pooling. The authors then analyze structural consequences of this decomposition. The work provides a narrative about latent "subagents" and analyze the emergence of "Luigi/Waluigi" phenomena.

**Compliance With Llm Reviewing Policy:**

Affirmed.

**Final Justification:**

My score remains 3 because, while the paper proves that positive distributions admit logarithmic-pool factorizations, the manuscript still presents a neural network as a collection of interacting subagents and in Section 5 explicitly assumes an "established witnessing set" whose members are interpreted as emergent personas.

The rebuttal clarified that the strongest results are better understood as geometric statements in log-probability space, and that no privileged or human-interpretable witnessing set is identified. However, since the manuscript itself still strongly invites the stronger LLM-facing interpretation, I do not think the current version adequately supports claims about identifiable latent subagents/personas in real language models.

**Key Questions For Authors:**

N/A

**Limitations:**

The most important implications are not addressed, such as the non-identifiability of the decomposition. Therefore, the interpretation of $P_i$ as subagents is underdetermined.

**Strengths And Weaknesses:**

The paper formulates a consistent probabilistic model of i) agents as probabilistic models, ii) welfare as log score, iii) aggregation as log opinion pooling.

However there are weaknesses with this submission:
- The central modelling assumption, that a language model can be interpreted as a logarithmic pool of latent subagent distributions, remains unvalidated.
  - The appendix explicitly argues that any strictly positive distribution admits such a factorization. But the factorization is non-unique, and there are infinitely many choices of $\{P_i, \beta_i\}$ with $P \propto \prod_i P_i^{\beta_i}$. Thus, without any additional constraints, there is no indication which $P_i$ are the real subagents, and "Luigi/Waluigi" can be redistributed across components without changing $P$.
  - The paper implicitly hopes that, even though many factorizations exist mathematically, the training dynamics or architecture might induce a natural decomposition. While logit decompositions do arise in some explicitly modular architectures, the authors don't show that a large language model admits such a stable decomposition. Ways that future work might validate this hypothesis could be:
    - mechanistically identifying circuits or subnetworks
    - if the decomposition could predict how the model will change after finetuning
    - if there was some uniquely salient decomposition?
    - But the paper establishes none of these.

Many of the results the authors prove can be understood as as statements about the geometry of the log-probability space, rather than being about interactions among specific subagents. Because the witness decomposition is highly non-identifiable, any conclusions about particular subagents gaining or losing weight (e.g., Luigi/Waluigi dynamics) depend on the chosen witness set.

The paper suggests the Luigi/Waluigi emergence may arise in language models. But without evidence that the model contains identifiable components, the alignment claims are speculative.

Overall the paper's mathematical results appear sound but the impact and implications on alignment are greatly limited by the vast non-identifiability of witness set mentioned.

---

> ### Author Rebuttal · Authors · 2026-03-30
>
> Thank you for your review. We believe your feedback reflects places where our presentation of the results could be clearer, and we provide detailed clarifications below.
>
> **[Decomposition as modeling assumption?]** The reviewer notes that the subagent decomposition is a "central modelling assumption" that "remains unvalidated". However, we respectfully clarify that this is *not* an assumption, and is a mathematical property that we prove. For instance, Theorems 11 and 12 (Section 4) and Theorems 39 and 41 (Appendix F) establish precisely for this purpose that any positive distribution admits a log-pool factorization into pairwise-distinct components for non-degenerate weights (e.g., see the exponential tilting arguments in lines 1537-1601).
>
> The actual modeling assumptions of the paper are minimal and grounded directly in neural network architecture (e.g., discussion in Sections 1-2, Appendix C):
>
> - interpreting the softmax/linear final layer of a neural network as defining a distribution over outcomes, and
>
> - treating cross-entropy training objective as utility maximization (with $U_i(o) = \log P_i(o)$).
>
> Everything else, including the existence of subagent decompositions, follows via mathematical analysis motivated by economics and game theory, applied in the neural-network setting. Because many modern architectures employ precisely a linear layer followed by softmax, these assumptions are not restrictive but rather descriptive of the models the framework targets. Further discussions are provided in lines 038-098, Section 1.
>
> **[Non-unique decomposition]** The reviewer notes as a central weakness that the subagent "factorization is non-unique", asking which $P_i$ are the "real" subagents. We respectfully clarify that the non-uniqueness of the subagent decomposition is a strictly necessary property, as discussed in Section 4, lines 226-237 and Appendix F, lines 1385-1614, etc. If the framework admits only a single decomposition, it would be invalid and too rigid to model agents that plausibly harbor multiple overlapping subdrives, which is precisely the setting we aim to capture. We note that the theory in Section 3.1 and Appendix E is developed specifically to clearly and rigorously articulate this in mathematical terms. The non-uniqueness reflects the genuine degrees of freedom in how an agent can be understood as composed of interacting subagents.
>
> An appropriate analogy is the basis of a vector space. A vector space admits infinitely many bases, and no single basis is "the real one." Yet theorems proved using *any* valid basis–about dimension, projections, or rank–remain valid and meaningful. Similarly, our results characterize structural properties of the log-probability space (e.g., anti-aligned compensation, subspace-dependent suppression bounds, etc.,) that hold for all valid decompositions simultaneously.
>
> The question of whether a given neural network admits a *strictly unanimously compositional* witnessing set is a distinct and interesting open problem that we propose using our framework, which provides a principled theoretical vocabulary to formalize and ask such questions. We view identifying conditions under which such decompositions arise, whether via mechanistic interpretability, architectural constraints, or training dynamics as the reviewer notes, as a promising direction for follow-up empirical work.
>
> **[Section 5 & Compositionality]** The reviewer states that "any conclusions about particular subagents gaining or losing weight (e.g., Luigi/Waluigi dynamics) depend on the chosen witness set." We respectfully clarify that this is not the case. The calculations in Section 5–first-order directional calculus in probability space and the bounds on misaligned-event probability–rely on local directional analysis and general properties of *any* arbitrarily fixed decomposition basis only, and not on a specific choice of witnesses (c.f., the vector space/basis analogy). The surrounding discussion was included purely to provide intuition and contextual motivation, softening what would otherwise be a highly technical section. We will revise the exposition in Section 5 to further clarify this point.
>
> **[Logit geometry and subagent interaction]** Linking the geometry of the logit space with interactions among specific subagents or subdistributions comprising probability distributions is precisely one of the novel contributions of our work. That the geometry admits a natural interpretation in terms of subagent interactions enriches rather than undermines our results, and we do not view this as a weakness.
>
> To incorporate the reviewer's feedback, we will revise the presentation to further clarify these points. Please let us know if any concerns remain. We are very happy to discuss further.

---

> > ### Author Rebuttal · Reviewer_uc9Z · 2026-04-01
> >
> > Thanks to the authors for their response.
> >
> > Thank you for correcting me in my review. Yes, it is clearly proven in the manuscript that any positive distribution can be factorized into logarithmic pool of pairwise-distinct components.
> >
> > My main concern remains the stronger interpretive claim. The paper presents a neural network as a collection of interacting subagents and in Section 5 explicitly assumes an *"established witnessing set"* whose members are interpreted as emergent personas. While I appreciate the clarification that the strongest calculations in Section 5 can be read as local geometric statements in log-probability space, I do not think the paper establishes that these witnesses correspond to identifiable subagents in a real language model, rather than one admissible interpretation of the same underlying geometry.
> >
> > To be clear my concern is not that the framework should admit only a unique decomposition.  But once the paper interprets a chosen witnessing set as specific personas, I do not see what identifies that witnessing set as more than one valid decomposition among many. I remain thus somewhat unresolved about the LLM-facing interpretations.

---

> > > ### Author Response · Authors · 2026-04-01
> > >
> > > Absolutely, thank you for the follow-up.
> > >
> > > Section 5 does indeed explicitly take a subagent decomposition basis (i.e., an “established witnessing set”), and we introduce the heuristic motivation that these may be thought of as “personas” when following the mathematical analysis. On this point, we fully agree that we do not provide a method for constructing a privileged witnessing set or basis. We do not show that a witnessing decomposition (whose existence we do establish for LLMs), must be human-interpretable, nor do we identify a property that would distinguish one such decomposition from the many others that may exist. More broadly, formalizing human-interpretability itself is quite difficult within the information-theoretic framework used in our work, and we do not attempt to resolve that problem here. In fact, Reviewers Tz1Z and 1eNB (both with accept scores 5) also enquired about this as well, and raised the question of whether one can construct a privileged witnessing set or manipulate particular decompositions, to obtain greater empirical utility. We agree that this is a fascinating and important question that our framework provides the machinery to formalize cleanly. However, we do not solve that problem here, and doing so lies outside the scope of the paper.
> > >
> > > In writing the paper, we made a concerted effort to present the mathematics in a way that preserves the intuitive motivations behind it, so that the reader never loses sight of the ideas from which the formalism is derived. This is why we use heuristic language such as “personas” alongside the technical development. Our intent was to aid intuition, not to claim that an arbitrary witnessing decomposition can thereby be completely identified with a human-interpretable goal or persona, such as deception. If the current presentation gives the impression that the theory cleanly establishes such an identification, then we will revise the discussion to make this distinction more explicit.
> > >
> > > To complement the response above, the key point here is abstraction. In our framework, a “trait” is represented by a probability distribution $P$ over the outcome space (e.g., discussion in Section 5). Alignment is then evaluated relative to that trait via the centered $P$-inner product introduced in our paper (Section 5, lines 336-348, or Appendices I.1-I.2, lines 2229-2364 for a more detailed explanation). Strictly speaking, this $P$ is simply an abstract probability distribution; it does not need to correspond to a human-interpretable goal or concrete persona. In this sense, the theory does not depend on having a privileged or anthropomorphic decomposition. Rather, it provides a formal notion of alignment to an abstract trait distribution $P$, and studies the corresponding emergence and shattering effects in that space.
> > >
> > > The reviewer writes that they “do not see what identifies that witnessing set as more than one valid decomposition among many.” Absolutely. The surprising result is that, for the emergence and shattering theorems, no privileged witnessing set is required. We prove that if the relevant $P$-inner-product alignment holds with the $P$-centered logarithmic character profiles $v_i(o)$ (Section 5, line 343), then the emergence and shattering theorems apply against that particular trait distribution $P$. Aside from the main text, we expand this out fully in Appendices I and J. This is how a desirable "trait" or property is modeled mathematically in the paper along with subagent log profiles, and it is part of the novelty of the framework in conjunction with the proofs. Although the interpretation is heuristically intuitive, the fact that the results go through cleanly at this level of abstraction is highly nontrivial.
> > >
> > > Finally, to summarize this discussion, we would like to clarify a formalization point. In our framework, a subagent is defined as a probability distribution $P_i$ over the outcome space that participates in a parent decomposition (see the modeling setup in Section 2, Definitions 1-3 in Section 3, or Appendices C-D where we expand this in detail). Section 5 requires only the existence of such a decomposition, which we formally prove exists. It does *not* require that the corresponding witnesses be mechanistically identifiable circuits or human-interpretable personas. The reviewer’s comment appears to address this stronger interpretation, which is not a claim we make and is outside the scope of this theoretical contribution. We will add an explicit discussion stating that 'witnessing set' and 'persona' are used as heuristic motivation and do not imply mechanistic identifiability in any specific model.
> > >
> > > We thank the reviewer for this discussion, which has identified places where the presentation can be clarified. We will revise the presentation to make this distinction more explicit. Please let us know if any concerns remain; we would be happy to discuss them further. Thank you.

---

### Official Review · Reviewer_8Kay · 2026-03-13

**Soundness:** 2
**Presentation:** 3
**Significance:** 2
**Originality:** 2
**Overall Recommendation:** 4
**Confidence:** 2

**Summary:**

This paper provides a theoretical framework for modeling neural networks as composition agents. It models LLMs as log pools of latent subagents optimizing for utility. The authors establish mathematical boundaries for compositional groups, and formalize the Waluigi effect as an agent with an antagonistic anti-aligned persona to an aligned persona (Luigi). The agents introduce the *Waluigi Shattering* theorem and show that it can provide stronger suppression of misaligned outcomes than reinforcing the aligned persona alone.

**Compliance With Llm Reviewing Policy:**

Affirmed.

**Key Questions For Authors:**

Commentary and questions above.

**Limitations:**

Limitations are developed in the impact statement.

**Strengths And Weaknesses:**

### Strengths
* The reasoning and motivation for more rigor in alignment are clear.
* Mathematically formalizing empirical community concepts (such as the Waluigi effect) is helpful for the alignment field.
* The proofs provide an interesting theoretical bound on ensemble dynamics, showing that strict unanimity is impossible under linear pooling.

### Weaknesses

My primary concern is not that this is a strictly theoretical work, but rather that the core theorems used idealized assumptions that may not translate to the applied, empirical spaces that the paper claims to impact (e.g. RLHF alignment, persona emergence). Consider restructuring the framework to position it more as more of a theoretical baseline, or provide active grounding to show how these theories hold up under real-world training mechanics.

* The claim that "manifest-then-suppress strictly outperforms direct suppression" could benefit from empirical or simulated demonstration on a toy model. In particular, the static log pool and ridigt $\epsilon$-ball constraints may not be often found in practice and in traditional alignment training; how might empiricists apply these to the targeted fields of safety and alignment in practice, when ensembling and boundaries may not be so rigid?

* The `Waluigi Emergence theorem` states that probabilities of `Waluigi` must go up to balance any increases in `Luigi` probabilities within rigid boundaries. However, in practice, model updates using SGD are non-linear; SGD removes old representations and may not perfectly balance weights.

---

> ### Author Rebuttal · Authors · 2026-03-30
>
> We thank the reviewer for their feedback. We are pleased that the reviewer found the motivation and reasoning clear and the presentation good. We also agree that putting empirical alignment concepts on firmer mathematical footing is both important and valuable. Below, we incorporate the reviewer's feedback.
>
> **[Positioning as theoretical baseline]** Given that our work is a pure theory paper, we agree with the reviewer's suggestion that the paper could be framed as a theoretical baseline, and will make this positioning more explicit in Section 1. Our goal is not to claim that the present framework fully captures the complexity of modern alignment training, but rather to provide the first rigorous mathematical vocabulary for reasoning about subagent structure, internal stability, and compositional behavior in neural networks, analogous to how expected utility theory (e.g., [1-3]) provides an axiomatic foundation for decision theory without itself being an empirical claim about any particular agent. We view the contribution as foundational: it isolates a tractable setting in which precise theorems can be proved, and is intended to support future empirical as well as theoretical work.
>
> **[Idealized assumptions]** Regarding the mathematical modeling assumptions, the framework is not meant as a literal description of every SGD update or every RLHF-style training run. Instead, it studies a clean regime in distribution/logit space, where rigorous analysis is possible. The two main modeling assumptions from which our results are derived are well-motivated by modern neural architectures:
>
> - The log-pool structure arises because modern neural networks compute logits via a linear final layer and normalize via softmax–this is the dominant output architecture across language models, vision classifiers, and multimodal systems.
>
> - The epistemic utility assumption $W_i(o) = \log P_i(o)$ corresponds to cross-entropy training, the standard objective for autoregressive language models as well as classification models.
>
> That said, as with any theoretical framework, some idealizations are present (e.g., first-order linearization, discrete outcome spaces). We will add further discussions that position it as a theoretical baseline that establishes structural results, providing a foundation on which future theoretical or empirical work can build.
>
> For instance, the (Waluigi Emergence-) Shattering Theorem gives a precise sense in which suppressing a manifested misaligned component yields tighter bounds than reinforcing an aligned component alone. The rigid-ball constraint in Theorem 19 corresponds to the widely used KL-divergence regularization (Appendix J.4, Theorem 76), and the log-pool structure, in turn, arises naturally in any architecture whose final layer is a linear projection followed by softmax–which includes the vast majority of modern language models.
>
> **[Representation removal and subagent composition]** The reviewer raises an important point: SGD updates are non-linear and may not preserve internal representations. Additionally, we note that it is difficult to combine the actions of different subagents at the representation level, especially if they are parameterized by differing architectures. We agree that reasoning directly in this space is difficult, and this is precisely why our framework operates in the space of probability distributions rather than parameters. This is because if we only think on the representation or model weight level, the intrinsic coordinates of each subagent may differ even among the subagents of the same parameterized distribution. In other words, the ambient parent parameterization need not transfer its shape to the effective subagent tensors in the information space. While the underlying parameterization may change in complex ways, the induced distribution over outputs is the object our theorems address. Logarithmic pooling at the logit level is robust to reparameterization; different weight configurations that induce the same output distribution are equivalent from the framework's perspective. This is a deliberate design choice that substantially broadens the applicability of our results, since the theorems hold regardless so long as the output distribution is defined.
>
> We thank the reviewer for their review, which helped us better position the paper and clarify its contributions. Please let us know if any concerns remain. We are very happy to discuss further.
>
> [1] Karni, Edi. Axiomatic Foundations of Expected Utility and Subjective Probability. In Handbook of the Economics of Risk and Uncertainty, Vol. 1, Elsevier, 2014.
>
> [2] Grant, Simon, and Timothy Van Zandt. Expected Utility Theory. In The Handbook of Rational and Social Choice, Oxford University Press, 2009.
>
> [3] von Neumann, John, and Oskar Morgenstern. Theory of Games and Economic Behavior. Princeton University Press, 1947.

---

> > ### Author Rebuttal · Reviewer_8Kay · 2026-04-01
> >
> > Thanks for the rebuttal. I agree that softening the language around applied alignment and safety is needed to minimize overclaiming. I'll modify my score to reflect that this work and rebuttal are better positioned as a pure theoretical paper.
> >
> > My concerns about the probabilistic assumptions and idealized environment stand - providing a toy simulation or, to my original review, a more thorough discussion of the limitations of the stylized setup will better help to ground the piece.

---

> > > ### Author Response · Authors · 2026-04-01
> > >
> > > Thank you for the feedback and for raising your score. We appreciate the constructive dialogue throughout the review process. In the revision, we will add a dedicated paragraph discussing the stylized setup more explicitly, including its limitations and the positioning of the paper as a theoretical contribution. Your feedback has meaningfully improved how we present and communicate the work. Thank you.

---

### Official Review · Reviewer_Tz1Z · 2026-03-17

**Soundness:** 3
**Presentation:** 4
**Significance:** 3
**Originality:** 4
**Overall Recommendation:** 5
**Confidence:** 4

**Summary:**

This paper introduces a new modeling of general deep neural network as composition of probability generative models (PGM), a system that compose the output of several generative agents into one final output. The composition is treated as the choice of an f-divergence "barycenter" agent of the set of existing PGM agnets.

The main analysis is done under the setup that the composition is logarithmic pooling rules, which corresponds to DNNs that ends with a linear logits and softmax layer producing the outputs which is widely used in modern models. The log-pooling composition could be derived from KL-divergence (one special case of f-divergence), and is natural from utility aggregation and corresponds to the standard softmax output layer of modern classifiers.

There are 3 main contributions:
1. The paper derives sharp possibility/impossibility results for unanimously beneficial compositions. A conclusion is, under log-score utilities, strict unanimity is impossible for binary outcome spaces but achievable when outcome set is of cardinality $\ge3$.
2. The paper proves recursive decomposition properties (cloning invariance, openness, and continuity), demonstrating that any positive distribution can be factorized into distinct subagents via log-pooling, and that the set of strictly unanimously decomposable distributions is open.
3. This paper provides a formal treatment of the Waluigi effect, showing that increasing the weight of a benevolent subagent (Luigi) while keeping the overall distribution close (e.g., under a KL budget) necessarily strengthens an antagonistic subagent (Waluigi). This leads to a theoretical comparison of alignment strategies, proving that manifest-then-suppress of the adversarial persona yields strictly greater reduction in undesirable outcomes than reinforcing the benevolent persona alone.

This paper is purely solid theoretical, every term has a definition and every claim in terms of a theorem / proposition has mathematical proof. This work builds a principled foundation for understanding internal conflicts in AI systems and for designing theoretically grounded alignment interventions.

**Compliance With Llm Reviewing Policy:**

Affirmed.

**Key Questions For Authors:**

1. In real world alignment, how to construct an appropriate or a "good" waluigi-agent that helps alignment in an efficient way?
2. The improvement on misalignment using waluigi-shattering seems to be continuous on vairous dependencies. Could the conclusion be continuiated to the non-negative measure boundaries? As mentioned in [W2], the real world LLM alignment may not guarantee a nonzero probability on every output.

**Limitations:**

Yes

**Strengths And Weaknesses:**

### **Strengths**
1. The paper originally builds a connection between DNN models to game theory / economics, introducing the concepts such as epistemic utility / Pareto improvements into DNN analysis. It provides novel insights about models and model improvement behaviors.
2. This paper is purely solid theoretical, every term has a definition and every claim in terms of a theorem / proposition has mathematical proof.
3. The model applies to a wide variety of DNN models, the only requirement on model structure is that the ending layer is a linear layer of logits and softmax output. Such a structure is widely used in current AI area.
4. The paper provides a mathematical proof of the Waluigi effect which is observed in model alignment, theoretically guarantees its existence.
5. It provides a clear guidance on model alignment. Theorem 21 guarantees that shattering strategy is better in the alignment.

### **Weaknesses**

1. The subagent decomposition seems to work in arbitrary amount, could there be a suggestion of a most appropriate decomposition in various downstream applications?
2. A lot of results depend on the positivity of the probability measure. However, in real LLMs, the scale of outputs and quantization may make some outputs bearing a zero probability. There could be another analysis on when / how the results fail to work in this extreme situation. Furthermore, in real world alignment tasks, whether the misalignment in Theorem 21 has certain continuity at boundary of nonnegative measures so that the improvement still exists?
3. In real world alignment, the real need is the construction of an efficient waluigi-agent to minimize the misalignment of the system.
4. This paper is purely theoretical, providing a mathematical explanation of the Waluigi effect and possible treatment of it. However, I still think a simple simulation / experiment / example could help general audience to understand the theory, and also acts as a basic example that show the assumptions of the theory are valid in a concrete case.

---

> ### Author Rebuttal · Authors · 2026-03-30
>
> We thank the reviewer for the thoughtful and detailed review. We are glad the reviewer found the presentation excellent and recognized the paper’s contributions on developing a principled theoretical foundation for reasoning about internal stability, compositional substructures, and alignment in modern neural networks. We also appreciate the reviewer’s observation that the framework naturally connects contemporary DNNs with ideas from game theory and economics, including epistemic utility and Pareto improvement.
>
> **[Constructing targeted subagents]** This is an important question and a natural next step for the framework. In general, decompositions are not unique, just as a vector can be expressed in many different bases. One impactful question is which decompositions are most useful under downstream criteria such as interpretability, sparsity, orthogonality, or steerability. Our paper provides the formal machinery to state this question precisely, but it does not yet develop a canonical construction algorithm. Likewise, our results theoretically analyze interventions using candidate subagent decompositions which we *prove* always exists, and leaves the separate algorithmic problem of discovering the most "useful Waluigi-agent” for practitioners as an open problem. A promising future direction is therefore to study decompositions aligned with empirically identified steering directions or persona-like directions in representation space, which may help make such subagents both more interpretable and useful for downstream alignment.
>
> **[Theoretical analysis in degenerate settings]** We agree that in practical systems, effective zero-probability outcomes can arise. Appendix F already addresses this issue for subagent factorizations (e.g., Theorem 41): a degenerate target distribution admits a log-pool factorization exactly when the positively weighted fixed factors are support-compatible, i.e., when the target support is contained in the intersection of their supports. Necessity follows because zeros propagate under logarithmic pooling. Sufficiency is constructive: one performs the factorization on the reduced support and assigns zero mass outside it, thereby recovering the degenerate distribution exactly. Extending the full theory, including boundary-continuity questions relevant to Theorem 21, would require substantially more measure-theoretic machinery. Such extensions would in essence be direct generalizations of the current proofs, but with far more notation that would almost certainly obscure the intuition and clear motivations of the present work. We therefore chose not to include that extension in the current version for readability, but agree it is a possible direction for future work.
>
> **[Simple toy examples for accessibility]** We agree that a simple illustrative example could help make the paper more accessible. Because this is a pure theory paper, such experiments would primarily instantiate already-proven claims rather than adjudicate between competing empirical hypotheses. For that reason, we did not include experiments in the current submission. That said, we agree that a small toy simulation or worked example could help readers build intuition, and we will consider adding one in the revision. More broadly, we will add to the introductory framing to make clearer that the goal of the paper is to provide a rigorous mathematical apparatus for subagent structure, internal stability, and compositional behavior in neural probabilistic models: a theoretical foundation on which future theoretical or empirical work can build.
>
> We appreciate the reviewer's suggestions. Thank you for your detailed review.

---

### Official Review · Reviewer_1eNB · 2026-03-18

**Soundness:** 3
**Presentation:** 3
**Significance:** 3
**Originality:** 4
**Overall Recommendation:** 5
**Confidence:** 3

**Summary:**

This paper introduces a new theoretical framework where neural networks are treated as collections of subagents each with their own utility, beliefs, and action distributions. The overall network is then characterized as a pooling of these subagents. The authors prove that strict unanimity is impossible in binary action settings with epistemic utility, but feasible for larger action spaces. They also demonstrate that strict unanimity is impossible under linear pooling, justifying the assumption of log-pooling used throughout much of the paper. The authors then prove several properties of such compositions: that decompositions exist for any agent, can be extended, and that subagents can be further decomposed. These results are then applied to explain the Waluigi Effect: when the weight on one subagent is increased within an $\epsilon$-KL ball in the agent's distribution, this naturally leads to an increased weight placed on an anti-aligned subagent.

**Compliance With Llm Reviewing Policy:**

Affirmed.

**Final Justification:**

The rebuttal was clarifying and maintained my positive score.

**Key Questions For Authors:**

1. Is your framework's utility in the broad insights offered (manifest-then-suppress over direct elicitation) alone or would realizing a decomposition into subagents provide further benefits?
2. If the latter, how would this be realized in practice?

**Limitations:**

Yes

**Strengths And Weaknesses:**

## Strengths
1. The paper addresses a timely problem of broad interest and applicability. Within this scope, it aims high, hoping to provide a foundational framework to guide work in alignment and AI agency.
2. The technical details of the work appear to be sound and constitute a strong contribution
3. Theorems are interpreted and assumptions (e.g., the use of epistemic utility) are discussed and justified reasonably.
4. The results of Section 5 (Waluigi emergence and shattering) are particularly interesting in the context of alignment.

## Weaknesses
1. While the writing of the paper is engaging, for many I think the Waluigi results would be the main payoff. The paper might benefit from teasing the nature of questions that their framework will help answer (e.g. "what happens to a compositional agent when I upweight a particular subagent?"). While I agree that foundational frameworks like the one offered are useful and needed, establishing this necessity up front would help motivate a reader through a (sometimes dense) mathematical framework.
2. While the Waluigi results nicely show the explanatory power of the framework, they rest on the assumption of "an established witnessing set" of subagents. It is not clear how one might come about such a set or if realizing the set itself would actually provide any utility. More broadly, I think the paper could do a better job of explaining how practitioners should go about utilizing their framework. Is its utility in the broad insights offered (manifest-then-suppress over direct elicitation) alone or would realizing a decomposition into subagents provide further benefits?

---

> ### Author Rebuttal · Authors · 2026-03-30
>
> We thank the reviewer for their review. We are pleased that they found the technical development to be a strong contribution and the paper to address a timely problem of broad interest. We also appreciate that the reviewer found the theorems well interpreted and the assumptions, including the use of epistemic utility, reasonably discussed and justified. It is encouraging that the reviewer has found our paper to be engagingly written. Below, we incorporate the reviewer's feedback.
>
> **[Utilitarian Witnessing Sets]** The results in Section 5 take as their decomposition basis an established witnessing set of subagents, whose existence we prove constructively in Theorems 11 and 12 of Section 4. Importantly, the Section 5 theorems do not require this witnessing set to be strictly unanimously compositional in the sense of Definition 15; thus, a valid decomposition basis is always available. More broadly, we agree with the reviewer that identifying a particularly useful witnessing set is an interesting open problem. Our contribution is to show that analyzing interventions through candidate subagent decompositions, which we prove always exist, already yields novel insights. At the same time, selecting a decomposition that is especially useful for a fixed downstream objective is nontrivial. We view one contribution of the paper as providing the formal toolkit needed to pose this question precisely and mathematically. The development of a canonical procedure for extracting an optimal or otherwise task-specific substructure is an intriguing direction for future work.
>
> **[What is the utility of the framework?]** The utility of the framework is broader than the specific insight of manifest-then-suppress over direct elicitation, though that is one concrete implication. The paper’s intuition is to notice two natural assumptions motivated by probabilistic neural models and then carefully study their consequences through an abstract formalism that connects DNNs with economics and game theory. In particular, it models subagents as probability distributions over outcomes and, motivated by canonical training objectives, defines welfare in terms of log-probability and asks when the resultant parent agent makes each constituent better off according to a well-motivated welfare metric.
>
> From there, the paper provides sharp theoretical guarantees including possibility, impossibility, recursion, and local-geometry results, and then uses that machinery to build a formal version of empirical alignment phenomena as a concrete example. It also defines a harmful event set, studies how its probability changes under small perturbations via decompositionality, and proves that adding a novel anti-aligned direction can increase the best achievable first-order suppression of that harmful set.
>
> **[Summary]** More broadly, our work introduces a novel probabilistic framework for modeling neural networks as compositional agents, in which a monolithic model is treated as a collection of interacting subagents whose utilities are aggregated. This provides formal tools for analyzing internal stability, coherence, and tradeoffs–phenomena central to alignment that previously lacked a unified mathematical foundation. Mathematically axiomatic reasoning about behavioral phenomena in LLMs has been difficult precisely because no prior rigorous framework existed. Our work is intended to address this gap by developing, to our knowledge, the first rigorous theoretical foundation for subagent structure and compositional behavior in neural probability models. We therefore view the framework as establishing a rigorous foundation: it isolates a tractable setting in which precise theorems can be proved, and is intended to support future empirical as well as theoretical work.
>
> To incorporate the reviewer's comments, we will revise the introduction to further tease out the nature of the questions that the framework helps answer, as the reviewer suggests. Thank you for your feedback.

---

> > ### Author Rebuttal · Reviewer_1eNB · 2026-04-02
> >
> > The authors adequately responded to my comments. I maintain my score.
> >
> > One note: I think the paper could benefit from a toy example worked out in greater detail. This would make it easier for the reader to wrap their head around the concepts introduced. If needed for space, I think some of the technical results could be abridged or relegated to the appendix without taking away from the paper significantly.

---

> > > ### Author Response · Authors · 2026-04-02
> > >
> > > Thank you for the helpful suggestion. We agree that a worked toy example could improve accessibility, and we appreciate the practical advice on making space for it by moving some technical results to the appendix. We will take this into account in the revision.

---

### Decision · Program_Chairs · 2026-04-30

**Decision:**

Accept (regular)

**Comment:**

The paper presents a theory of neural models as a collection of subagents. Technically, each subagent is a probabilistic generative model, whose output is composed as a barycenter via logarithmic pooling. Theoretical analyses are presented, and the paper concludes with an analysis of alignment problems.

The reviewers note a number of strengths of the paper. First, the theoretical and technical approach was noted as very strong. Second, the theoretical results themselves were noted as particularly interesting. Third, the usefulness of the theory in understanding problems in alignment was commended. Notably, the reviewer indicating weak reject is willing to increase their rating, after the discussion, to weak accept contingent on modest edits to the paper. I agree that the paper is novel and interesting.